# GenOL: Generating Diverse Examples for Name-only Online Learning

**Minhyuk Seo**[*]                                                     *KU Leuven, Seoul National University*
**Seongwon Cho**                                                     *Seoul National University*
**Minjae Lee**                                                         *Seoul National University*
**Diganta Misra**                                                  *ELLIS  MPI-IS Tübingen*
**Hyeonbeom Choi**                                                 *Seoul National University*
**Seon Joo Kim**[†]                                                    *Yonsei University*
**Jonghyun Choi**[†]                                                 *Seoul National University*

**Reviewed on OpenReview:** *https://openreview.net/forum?id=QPfVoTMLWq*

## Abstract

Online learning methods often rely on supervised data. However, under data distribution shifts, such as in continual learning (CL), where continuously arriving online data streams incorporate new concepts (*e.g.*, classes), real-time manual annotation is impractical due to its costs and latency, which hinder real-time adaptation. To alleviate this, 'name-only' setup has been proposed, requiring only the name of concepts, not the supervised samples. A recent approach tackles this setup by supplementing data with web-scraped images, but such data often suffers from issues of data imbalance, noise, and copyright. To overcome the limitations of both human supervision and webly supervision, we propose *Generative name only Online Learning* (**GenOL**) using generative models for name-only training. But naïve application of generative models results in limited diversity of generated data. Here, we enhance (i) intra-diversity, the diversity of images generated by a single model, by proposing a diverse prompt generation method that generates diverse text prompts for text-to-image models, and (ii) inter-diversity, the diversity of images generated by multiple generative models, by introducing an ensemble strategy that selects minimally overlapping samples. We empirically validate that the proposed GenOL outperforms prior arts, even a model trained with fully supervised data by large margins, in various tasks, including image recognition and multi-modal visual reasoning. Code is available at https://github.com/snumprlab/genol.

## 1 Introduction

Online learning has achieved remarkable progress, demonstrating improved model adaptability to continuously evolving data streams. However, most existing approaches (Wang et al., 2022; Kim et al., 2024a) still rely on large, meticulously curated datasets that demand extensive human supervision. For instance, constructing the 423.5k clean images in DomainNet (Neyshabur et al., 2020) required over 50,000 hours of manual filtering to remove outliers. In standard learning, where all training data are provided at once, the time spent on data collection and preprocessing does not affect performance, as these steps are completed prior to model training. In contrast, online learning continuously introduces new data that may encompass novel 'concepts' (*e.g.*, classes, adjectives, and verbs in multi-modal tasks) under distribution shifts, a setting known as continual learning (CL), requiring ongoing data preparation throughout the training process. Consequently, delays in preparing data for newly encountered concepts in online learning can directly hinder the model's real-time adaptation to new concepts (Koh et al., 2021; Caccia et al., 2022).

To address the issue of human annotation in the online setup, particularly under distribution shifts, Prabhu et al. (2024) propose 'name-only continual learning' setup, which requires only the names of new concepts,

---

[*]First author: minhyukseo@yonsei.ac.kr.    [†] Corresponding authors.

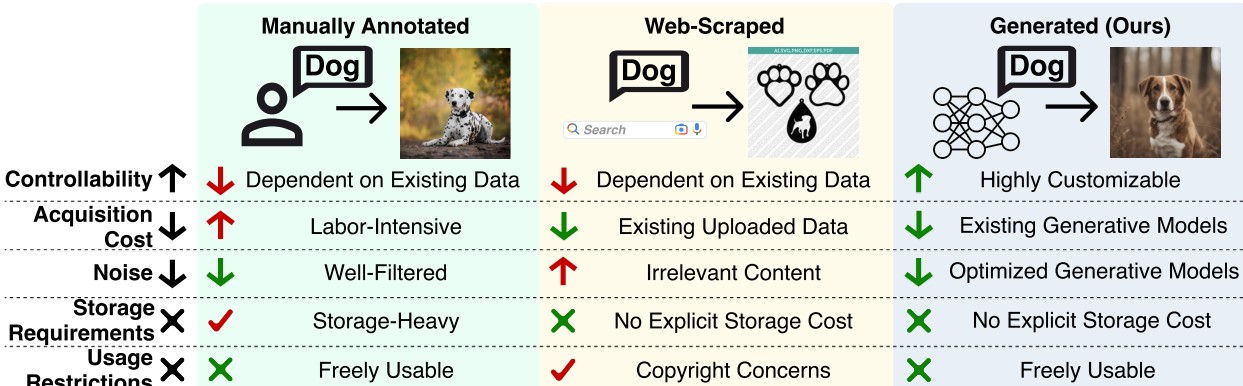

Figure 1: **Comparison of manually annotated data, web-scraped data, and generated data (ours) for name only continual learning.** Generated data overcomes usage restrictions (*i.e.*, whether images can be used for learning) and offers controllability, enabling the generation of images in diverse contexts (*e.g.*, background, color) as desired. Compared to web-scraped data, generated data contains fewer irrelevant samples (*i.e.*, less noise), and they are more cost-effective than MA data, which requires manual human annotation. For more details on the terminology used in this figure, see Section A.30.

not the manually annotated data. They propose to use web-scraped data relevant to the given concepts as an alternative to manual annotation. Although web-scrapped data are abundant and easily accessible (Sun et al., 2018), challenges arise from copyright concerns (Zhang et al., 2023a), as well as inherent noise (Neyshabur et al., 2020), which significantly hinders the performance of continual learner (Kim et al., 2021).

To tackle these challenges of using web-scrapped data, we propose leveraging text-to-image (T2I) generative models for name-only setup. Specifically, we propose a generative name-only framework, *Generative name only Online Learning* **(GenOL)**, which takes only *concepts* as input and trains on images generated from given concepts without requiring labeled data. It takes advantage of generative models, such as controllability (Nie et al., 2021) (*i.e.*, generating desired data) and unlimited image generation (Liang et al., 2022), as shown in Figure 1. Additionally, it significantly accelerates data collection, *e.g.*, generating DomainNet with SDXL(Podell et al., 2023) on 8 NVIDIA RTX 4090 GPUs takes only 80 hours, compared to 50,000 hours for manual annotation. Note that GenOL can be utilized both general online learning setup and online continual learning under distribution shifts that introduces new concepts over time

In the name-only CL scenario, we focus on cases where new concepts represent *relative novelty*, concepts that are new to specific systems or domains, rather than completely novel concepts to the world (Cossu et al., 2022; Shaheen et al., 2022). For example, a deployed home assistant robot may need to learn new behaviors (Kim et al., 2024a) like 'ride a bike' and 'hold a bike', or e-commerce recommendation systems may need to incorporate new categories, such as 'bladeless fans' or 'electric scooters', after deployment. While these are common concepts, not all common concepts can be included in pre-deployment training due to the impracticality of anticipating all possible operational scenarios. To this end, we aim to train models with new concepts that are actually needed in their operational environment (*i.e.*, personalization), leveraging generative models' broad coverage of most everyday concepts (Ghasemi et al., 2024). We leave name-only CL for completely novel concepts unknown to generative models as future work.

Despite the advantages of generative models, generated images often suffer from limited diversity (Tian et al., 2024a; Bianchi et al., 2023; Fraser et al., 2023). To address the limited diversity, we first define *intra-diversity* and *inter-diversity*, which refer to the diversity within data generated by a single T2I model and the diversity among data generated by multiple T2I models, respectively. To improve intra-diversity, we propose HIerarchical Recurrent Prompt Generation (**HIRPG**), a prompt generation method leveraging LLMs' in-context learning to create diverse text prompts. To enhance inter-diversity, we propose COmplexity-NAvigating eNsembler (**CONAN**), a complexity-guided data ensemble approach that aggregates samples from multiple generative models, selects a representative coreset, and trains a model exclusively on this coreset for efficient training and real-time adaptation to new concepts. We empirically demonstrate that GenOL significantly outperforms baselines across name-only class-incremental learning, multimodal visual concept-incremental learning setups, and even online learning setups without distribution shifts.

In summary, we address the following research questions, highlighting our core contributions:

**RQ1**. *Can generated data replace manual annotation in online learning setups?* GenOL improves $A_{\mathrm{AUC}}$ on the PACS OOD domain by 9% and 13% over models trained on web-scraped and MA data, respectively.

**RQ2**. *How can we improve diversity in images generated by generative models?* We propose HIRPG, a prompt generation method that leverages LLMs to create diverse text prompts for a given concept, which are then used by T2I generative models to generate varied images.

**RQ3**. *How can we improve image diversity generated by multiple generators in an ensemble?* We propose CONAN, a data ensembling strategy that considers the overlap and complexity of generated data.

## 2  Related Work

**Learning from generative models.** With the rise of robust generative models, several recent studies have leveraged synthetic data for training (Zhang et al., 2024d; Tian et al., 2024b). To train a model with data generated by T2I generative models for a given concept $c$, concept-specific prompts $p_c$ are required. Ramesh et al. (2022); Jones et al. (2024) use the template "A photo of a $c$", as in CLIP (Radford et al., 2021), to construct prompt for concept $c$, while Sarıyıldız et al. (2023) claims that using just the concept as a prompt ($p_c =$ "$c$") yields better image generation. To add more concept-specific context to $p_c$, Sarıyıldız et al. (2023) combine the concept name $c$ with its definition $d_c$ from WordNet (Miller, 1995), resulting in $p_c =$ "$c, d_c$". Nonetheless, all these approaches rely on a single type of $p_c$ per concept, limiting the diversity of generated images despite the ability of T2I models to generate an unlimited number of images (Tian et al., 2024a).

To enhance the diversity of generated images, several prompt diversification methods have been proposed. LE (He et al., 2023) uses a pre-trained word-to-sentence T5 model (Raffel et al., 2020) to generate diverse sentences that incorporate the given concepts. Sarıyıldız et al. (2023) integrates the concept's hypernym $h_c$ from WordNet and background scenes $b$ from Places365 (López-Cifuentes et al., 2020), resulting in $p_c =$ "$c, h_c$ inside $b$". In contrast to random background selection, Tian et al. (2024a) uses LLMs to generate contextually appropriate backgrounds for the given concept $c$, creating more plausible prompts. Similarly, Hammoud et al. (2024) and our proposed diverse prompt generation method, HIRPG, also employ an LLM for prompt generation. However, unlike previous LLM-based prompt generation methods, which do not consider the relationship between generated prompts, HIRPG minimizes overlap between them by providing previously generated prompts as negative examples to the LLM.

**Coreset selection.** Building a coreset from a larger candidate set and using only the coreset for training has been explored to reduce computational costs (Killamsetty et al., 2021c; Xia et al., 2024). Formally, from the candidate set $T$, these methods select a coreset $V$ ($|V| \ll |T|$), that preserves as much information from $T$ as possible (Shin et al., 2023). To estimate the informativeness of the candidates, several metrics have been proposed, such as gradient (Paul et al., 2021; Pooladzandi et al., 2022), uncertainty (Coleman et al., 2020), influential score (Yang et al., 2022a; Pooladzandi et al., 2022), and distance (Xia et al., 2023).

Although these methods effectively select coresets, many have substantial computational costs. Gradient-based methods (Paul et al., 2021; Pooladzandi et al., 2022; Shin et al., 2023), which aim to minimize the difference between the gradients of the full training dataset $T$ and the selected set $V$, require a well-trained model on $T$, which significantly increases computational overhead. Similarly, the influence score-based method (Yang et al., 2022a) also requires significant computation due to the necessity of calculating the Hessian in the influence function, along with the iterative data selection process (Xia et al., 2023). In contrast, distance-based methods, such as Moderate (Xia et al., 2023) and our proposed CONAN, leverage a well-trained feature extractor without backward processes, *i.e.*, requiring only model forward passes for feature extraction, leading to faster data preparation time.

We review more relevant literature and provide extended related work in Sec. A.15 for space's sake.

## 3  Problem Statement of Name Only CL

Name-only CL setup (Prabhu et al., 2024) assumes that only new concepts $\mathcal{Y} = \{y_1, y_2, ...\}$ are provided in a stream, while prevalent online CL setups assume well-curated annotated data $(\mathcal{X}, \mathcal{Y})$ are given. The

objective is to train a model $f_\theta$, parameterized by $\theta$, to effectively understand the concepts seen up to the time step $t$, *i.e.*, $\{y_i\}_{i=1}^t$. To train $f_\theta$ on given concepts, the learner can access public data such as web-scraped or generated data. To evaluate whether $f_\theta$ has learned concepts $\{y_i\}_{i=1}^t$ at time step $t$, curated test data $\{\mathcal{X}_i, y_i\}_{i=1}^t$ are used, where $\mathcal{X}_i$ refers to the set of data corresponding to the concept $y_i$.

## 4 Approach

We propose the Generative name only Online Learning (**GenOL**) framework to address the absence of data in the name-only CL setup. The GenOL framework consists of four components: (i) Prompt Generation Module $\psi$ (Section 4.1), (ii) Text-to-Image Generators $\mathcal{G}$ (Section 4.2), (iii) Complexity-Guided Ensembler $\Delta$ (Section 4.3), and (iv) a learner $f_\theta$. We illustrate an overview of GenOL in Figure 2.

When a new concept is introduced, for which $f_\theta$ needs to be learned, a generator $g \in \mathcal{G}$ generates images related to the concept. Despite the capability of generative models to create an infinite number of images, the diversity of generated outputs is often limited (Liu et al., 2023a; Sadat et al., 2023). To improve the diversity, we introduce a diverse prompt generation module $\psi$, named HIerarchical Recurrent Prompt Generation (**HIRPG**), which creates diverse text prompts and feeds them into T2I generative models. Furthermore, we enhance diversity using an ensembler $\Delta$, which ensembles the images generated by multiple generators $\mathcal{G}$ while minimizing redundancy, thereby maximizing the diversity of the generated images. We name this ensemble strategy as COmplexity-NAvigating eNsembler (**CONAN**). Generated images are then streamed to the learner $f_\theta$ in real-time. Note that while it is possible to generate data for past concepts in real-time, we use episodic memory for efficiency, as it helps reduce computational costs.

### 4.1 Prompt Generation Module ($\psi$)

Prompt generation module $\psi$ begins with a new concept $c$ as input and constructs a base prompt $P_B$ using the template: '*A photo of c*', following (Shtedritski et al., 2023; Shi et al., 2023). While $P_B$ can be used directly with the T2I generators $\mathcal{G}$, generating multiple images using a single prompt may lead to limited diversity in style, texture, and backgrounds (Fan et al., 2024). Therefore, to enhance intra-diversity, the diversity of images generated by a single generator, $\psi$ leverages LLMs to generate additional diverse prompts.

A straightforward approach to generating $N$ diverse prompts using LLMs is to generate $N$ different prompts at once or to generate a single prompt $N$ times, as in previous work (He et al., 2023; Hammoud et al., 2024). However, generating $N$ different prompts at once using LLM often generates duplicated prompts with similar semantics (Shur-Ofry et al., 2024; Hayati et al., 2024). Similarly, in the case of generating a single prompt $N$ times, multiple inferences to an LLM with the same input can yield similar outputs (Zhang et al., 2024a; Skapars et al., 2024), despite the non-deterministic nature of LLMs (Song et al., 2024b).

**Recurrent Prompt Generation (RPG).** To reduce overlap between generated prompts, we iteratively generate new prompts that are distinct from those produced in previous steps. Inspired by previous work that solves complex problems by providing negative examples in in-context learning (Zhang et al., 2024b) and contrastive Chain-of-Thought (Chia et al., 2023), we explicitly include previously generated prompts into the LLM input as negative examples. Presenting previously generated prompts and requesting a new prompt distinct from them imposes a hard constraint that effectively prevents overlap between a newly generated prompt and the previous ones. Formally, this process is described as follows:

$$P_i = \psi(S_i; P_S), \quad \text{where} \quad S_i = \begin{cases} \{P_B\} & \text{if } i = 1 \\ \{P_B\} \cup \{P_j\}_{j=1}^{i-1} & \text{if } i \geq 2, \end{cases} \tag{1}$$

where $P_S$ denotes the system prompt and $S_i$ denotes the set of previously generated prompts up to the $i_{\text{th}}$ generation step. Since there are no negative examples in the initial step (*i.e.*, $i = 1$), we use the base prompt $P_B =$ '*A photo of c*' as the initial negative example. To generate $N$ different prompts, we repeat the process $N$ times. As previously generated prompts are iteratively used as negative examples, we call this process as Recurrent Prompt Generation (RPG). We provide the details of the system prompt $P_S$ in the Section A.4.

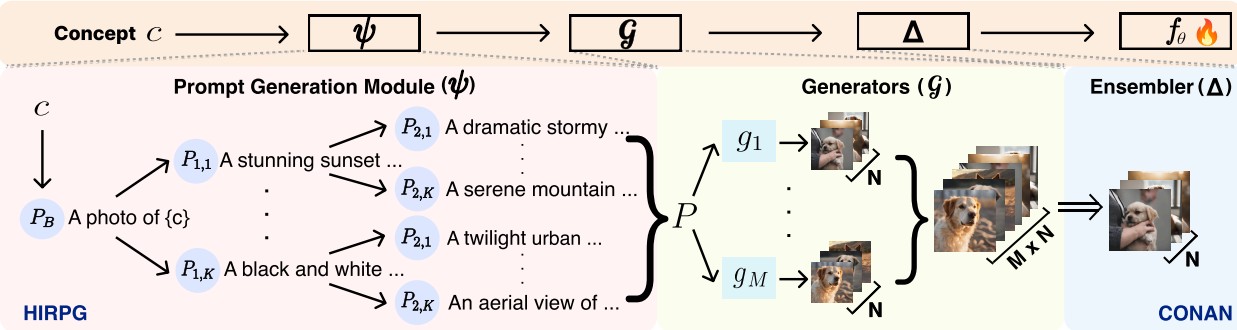

Figure 2: **Illustration of the proposed GenOL framework.** When a new concept that needs to be learned is encountered, it is passed through a prompt generation module, $\psi$, to produce diverse prompts. These prompts are then used to generate data from a set of generators, $\mathcal{G}$. The data generated by each generator are combined through the ensembler, $\Delta$, and subsequently used to train the model, $f_\theta$.

However, generating a large number of prompts using RPG poses a critical challenge. As the iterative steps are repeated, the LLM input length increases, causing difficulties in fully utilizing the long context, a problem known as *lost-in-the-middle challenge* (An et al., 2024; Liu et al., 2023c), as well as substantial computational overhead in long-context ICL (Li et al., 2024).

**HIerarchical Recurrent Prompt Generation (HIRPG).** To address this challenge, we divide the RPG into multiple subtasks using a hierarchical tree. Specifically, we construct a complete $K$-ary tree (Gross et al., 2018), where every internal node has $K$ child nodes. Each node represents a generated prompt, with the root node (*i.e.*, the node at depth $d = 0$) defined as $P_B = $ '*A photo of c*'. For fewer than $K$ prompts, RPG is performed at depth $d = 1$. For more than $K$ prompts, the tree extends to depth $d \geq 2$, with each parent node at depth $d - 1$ serving as the base prompt, as shown in Figure 2. Formally, focusing on the $k^{\text{th}}$ child node at depth $d$, denoted as $P_{d,k}$ ($d \geq 0$, $1 \leq k \leq K$), its child nodes $P_{d+1,k'}$ ($1 \leq k' \leq K$) are generated through the RPG as follows:

$$P_{d+1,k'} = \psi(S_{d+1,k'}; P_S), \quad \text{where} \quad S_{d+1,k'} = \begin{cases} \{P_{d,k}\} & \text{if } k' = 1 \\ \{P_{d,k}\} \cup \{P_{d+1,m}\}_{m=1}^{k'-1} & \text{if } k' \geq 2, \end{cases} \tag{2}$$

where $S_{d+1,k'}$ denotes the set of previously generated nodes that share the same parent node $P_{d,k}$. By constructing a complete $K$-ary Tree with a depth of $D$, we can generate $\frac{K^{D+1}-1}{K-1}$ nodes (*i.e.*, prompts), including all internal and leaf nodes. This hierarchical generation allows for diverse prompt generation while bounding the number of negative examples by $K$ ($\ll N$), thereby addressing both the *lost-in-the-middle challenge* and the computational overhead.

Since we divide RPG into subtasks using a hierarchical tree structure, we cannot consider nodes generated from different branches as negative examples. Nonetheless, overlap between generated prompts from different nodes is rare, as shown in Section A.34. This is because RPG in each node begins with a distinct $P_{d,k}$ in Equation 2, which serves as a negative example in the first step ($k' = 1$), and different examples in in-context learning lead to different outputs (Su et al., 2022; Agarwal et al., 2024). We present an ablation study on the components of HIRPG to validate the contribution of each component, *i.e.*, RPG and hierarchical generation (HIG), along with the pseudo code for $\psi$ in Section A.8 and Section A.17, respectively.

## 4.2 Text-to-Image Generators ($\mathcal{G}$)

To further amplify the diversity of generated images, we employ multiple T2I generators $\mathcal{G}$. Specifically, using a T2I generator $g_i(\cdot) \in \mathcal{G}$ and a prompt set $\mathbf{P}$ generated by $\psi$, we generate a set of images $U_i = g_i(\mathbf{P})$. At the end of generation using $\mathcal{G}$, we obtain $\mathbf{U} = \bigcup_{i=1}^{|\mathcal{G}|} U_i$, the union of images generated by $\mathcal{G}$, with the same number of images generated by each model, *i.e.*, $|U_1| = |U_2| = \cdots = |U_{|\mathcal{G}|}|$. We provide detailed information about the generators we employ in Section A.14.

### 4.3 Complexity-Guided Ensembler (△)

When ensembling images generated by different T2I models, a key question arises: *Do we need to use all of them?* While training on large-scale datasets is crucial for achieving performances (Zhao et al., 2021; Yang et al., 2022b), it increases time-complexity and computational costs (Sharir et al., 2020; Kim et al., 2022), as well as carbon footprint (Schwartz et al., 2020; Patterson et al., 2021). Moreover, in CL setups, extended training times hinder fast adaptation to new concepts (Seo et al., 2025).

Therefore, we select a coreset **V** from the entire generated data **U**, and train a learner only using **V**. Specifically, to maintain the same training cost as a single-generator setup while enhancing diversity, we sample the number of samples generated by a single generator (*i.e.*, $|U_i|$ samples) from **U**. A straightforward approach to constructing an ensemble set **V** is to sample an equal number of images from each generator. However, surprisingly, this degrades performance, even compared to training on images from a single generator (*i.e.*, no ensembling), as shown in Table 4. This degradation occurs because equal-weight sampling fails to account for the overlap between images, *i.e.*, diversity.

**Measuring Diversity using Relative Mahalanobis Distance (RMD) (Ren et al., 2021).** To enhance diversity in the ensembled set **V**, we select samples positioned far from the concept prototype in the feature space, *i.e.*, *difficult samples*, since these images are less likely to overlap with common images compared to those that are closer to the prototype (Zhang et al., 2024c; Mehra et al., 2025). Specifically, to select difficult samples in the ensemble set considering both concept-wise difficulty and their relationship to other concepts, we employ the relative Mahalanobis distance (RMD) score (Ren et al., 2021), which considers both the distance between a sample and its concept prototype and the distance between the sample and the global prototype (Cui et al., 2023). RMD score for a sample $(x_i, y_i) \in \mathbf{U}$ is given as follows:

$$
\begin{aligned}
\mathcal{RMD}(x_i, y_i) &= \mathcal{M}(x_i, y_i) - \mathcal{M}_{\text{agn}}(x_i), \\
\mathcal{M}(x, y) &= D_M\left(g(x), \ \frac{1}{|\mathbf{U}_y|} \sum_{j \in \mathbf{U}_y} f(x_j)\right), \\
\mathcal{M}_{agn}(x) &= D_M\left(g(x), \ \frac{1}{|\mathbf{U}|} \sum_{j \in \mathbf{U}} f(x_j)\right),
\end{aligned}
\tag{3}
$$

where $g(x)$ refers to the penultimate feature of the feature extractor $g$, $D_M$ refers to the Mahalanobis distance (MD), $\mathbf{U}_y$ denotes the set of samples belonging to class $y$, $\mathcal{M}(x_i, y_i)$ and $\mathcal{M}_{\text{agn}}(x_i)$ represents class-wise MD and class-agnostic MD (*i.e.*, global MD), respectively. If a sample is close to the class prototype but far from the global prototype (*i.e.*, low RMD score), it is easy to classify correctly. Conversely, if it is far from the class prototype but close to the global prototype (*i.e.*, high RMD score), the sample is hard to classify correctly and is more likely to be confused with nearby classes. We show examples with low RMD scores and samples with high RMD scores in Figure 3.

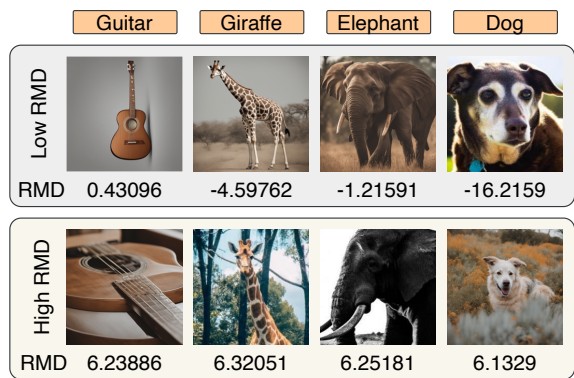

Figure 3: **Samples with high vs. low RMD scores.** Samples with low RMD scores have simple backgrounds or frontal views, while those with high scores show diverse backgrounds or viewpoints.

**Probabilistic Ensemble using RMD Score.** A coreset, a representative subset of an entire dataset (Anonymous, 2023), includes not only samples near the decision boundary but also concept-representative samples (Bang et al., 2021; Harun et al., 2023). Therefore, we adopt a probabilistic approach for ensemble selection, rather than simply choosing images with the highest RMD scores, to incorporate concept-representative samples. Specifically, we calculate $p_{u|c}$, the probability of selecting sample $u$ in concept $c$'s coreset, detailed as follows. We first truncate samples with RMD scores in the upper and lower $L\%$ to eliminate outliers. Then, we normalize the scores

using Z-score normalization and apply a softmax function to compute the selection probability as:

$$p_{u|c} = \frac{e^{\overline{RMD}_{u|c}/\tau}}{\sum\limits_{u' \in \mathbf{U}_c} e^{\overline{RMD}_{u'|c}/\tau}}, \tag{4}$$

where $\mathbf{U}_c$ denotes the set of generated data for concept $c$, $\overline{RMD}_{u|c}$ represents the normalized RMD score for sample $u \in \mathbf{U}_c$, and $\tau$ denotes the temperature. This selection probability allows for sampling complex samples while also including a small portion of concept-representative samples in the ensemble set. We compare it with various RMD-based ensembles (*e.g.*, top-k selection) in Section A.12.

## 5 Experiments

### 5.1 Experimental Setup

**Continual Learning Setups.** We empirically validate GenOL by comparing it with state-of-the-art methods in name-only class-incremental learning (CIL) and name-only multi-modal visual-concept incremental learning (MVCIL) setups (Seo et al., 2025), where class names and concepts (*e.g.*, 'ride a bike', 'kick a ball') are encountered incrementally. MVCIL setup requires both positive and negative support sets: positive sets contain images representing the concept, while negative sets include non-representative images. Following Seo et al. (2025), we consider two tasks addressing following queries: (1) Concept answering (CA) - *What is the concept exclusively depicted by the positive support set?* and (2) P/N - *Given a query image, does it belong to the positive or negative support set?*. Details of the MVCIL setup is in Section A.1.

**Models ($f_\theta$).** We use ResNet-18 (He et al., 2016) and ImageNet-1K pretrained ViT-base (Dosovitskiy, 2020) as a backbone network for the CIL experiments. For the MVCIL experiments, we use the LLaVA-1.5-7B (Liu et al., 2023b), fine-tuning only the projection MLP layers and LoRA adapters (Hu et al., 2021) for training efficiency, following Ye et al. (2023); Dong et al. (2024). In all experiments, we train a model with ER (Rolnick et al., 2019), which is a simple but strong CL method (Prabhu et al., 2023; Seo et al., 2025).

**Datasets.** We evaluate GenOL 's domain generalization in CIL using PACS (Zhou et al., 2020), CIFAR-10-W (Sun et al., 2024), ImageNet-R (Hendrycks et al., 2021), and DomainNet (Neyshabur et al., 2020), dividing them into multiple discrete tasks. Each DG benchmark consists of multiple domains, *e.g.*, PACS contains Photo, Art, Cartoon, and Sketch domains. For comparison with the oracle scenario that assumes sufficient manually annotated (MA) data, we select photo-realistic domain as MA data for each benchmark. This domain is treated as in-distribution (ID), while the remaining domains are considered out-of-distribution (OOD). For MVCIL experiments, we use Bongard-HOI (Jiang et al., 2022) and Bongard-OpenWorld (Wu et al., 2024a). We provide the details of the task split in Section A.2.

**Metrics.** We report $A_{\text{AUC}}$ (Koh et al., 2021; Caccia et al., 2022; Koh et al., 2023) and $A_{last}$, which measure inference performance at any time and at the end of training, respectively. Note that for evaluation, we use the test set for seen concepts up to that point in time. We provide details on metrics in Section A.13.

**Baselines.** We compare a model trained using GenOL with those using web-scraped data (C2C (Prabhu et al., 2024), IE (Li et al., 2023b), Seafaring (Sato, 2023), and Tiara (Sato, 2022)), other synthetic data (Glide-Syn(He et al., 2023), CHB (Sarıyıldız et al., 2023), SC (Tian et al., 2024a), LE (He et al., 2023) and CCG (Hammoud et al., 2024)), and MA data. Specifically, we compare HIRPG with diverse prompt generation baselines (CHB, SC, LE, and CCG). Furthermore, we integrate them with our proposed CONAN to demonstrate CONAN's plug-and-play capability and ensure a fair comparison with GenOL, which leverages multiple generators. Then, we compare CONAN with coreset selection baselines, *i.e.*, Uncertainty (Coleman et al., 2020), CRAIG (Mirzasoleiman et al., 2020), Glister (Killamsetty et al., 2021b), GradMatch (Killamsetty et al., 2021a), Adacore (Pooladzandi et al., 2022), LCMat (Shin et al., 2023), and Moderate (Xia et al., 2023), and active learning baselines, such as AL (Park et al., 2022) and LDM-S (Cho et al., 2025).

For details on prompt generation baselines, data ensemble baselines, name-only standard learning setup, and implementation/hyperparameters, see Sections A.3, A.9, A.10, and A.4, respectively.

### 5.2 Quantitative Analysis

| Method | PACS | | | | DomainNet | | | |
| --- | --- | --- | --- | --- | --- | --- | --- | --- |
| | ID | | OOD | | ID | | OOD | |
| | $A_{\text{AUC}}$ ↑ | $A_{last}$ ↑ | $A_{\text{AUC}}$ ↑ | $A_{last}$ ↑ | $A_{\text{AUC}}$ ↑ | $A_{last}$ ↑ | $A_{\text{AUC}}$ ↑ | $A_{last}$ ↑ |
| C2C (Prabhu et al., 2024) | 47.29±2.75 | 39.23±3.78 | 28.33±1.93 | 20.77±1.51 | 35.06±0.41 | 27.81±0.15 | 11.89±0.22 | 8.82±0.08 |
| Glide-Syn (He et al., 2023) | 34.59±2.14 | 32.05±1.44 | 31.53±1.56 | 26.56±1.84 | 15.64±0.44 | 10.68±0.19 | 4.06±0.13 | 2.59±0.03 |
| Real-Fake (Yuan et al., 2024) | 55.60±2.36 | 53.00±2.26 | 28.66±1.47 | 21.22±1.33 | 24.43±0.26 | 18.89±0.30 | 6.33±0.11 | 4.50±0.05 |
| IE (Li et al., 2023b) | 47.29±3.29 | 38.99±2.94 | 25.74±2.11 | 18.23±1.87 | 34.76±0.52 | 27.55±0.24 | 11.92±0.26 | 8.50±0.14 |
| Seafaring (Sato, 2023) | 45.42±2.61 | 41.15±2.42 | 27.10±2.05 | 18.31±1.66 | 32.58±0.53 | 27.24±0.22 | 10.47±0.28 | 7.05±0.11 |
| Tiara (Sato, 2022) | 39.97±1.39 | 37.78±2.73 | 24.18±2.90 | 20.05±1.48 | 27.01±1.82 | 20.33±0.39 | 8.35±0.51 | 6.53±0.74 |
| LE (He et al., 2023) | 46.47±2.00 | 45.76±2.33 | 32.42±1.35 | 27.56±0.66 | 20.01±0.27 | 15.38±0.31 | 6.40±0.13 | 4.59±0.09 |
| (+) CONAN | 49.37±3.77 | 50.45±1.56 | 33.88±1.79 | 30.29±0.81 | 30.80±0.63 | 25.33±0.20 | 9.54±0.25 | 7.59±0.17 |
| CHB (Sarıyıldız et al., 2023) | 47.52±2.69 | 46.11±1.07 | 31.02±1.11 | 22.82±1.61 | 16.69±0.16 | 13.45±0.19 | 5.61±0.11 | 4.18±0.05 |
| (+) CONAN | 52.01±2.72 | 45.46±3.27 | 32.62±1.72 | 24.26±0.89 | 29.06±0.37 | 24.52±0.17 | 9.28±0.14 | 7.56±0.14 |
| SC (Tian et al., 2024a) | 44.03±1.95 | 41.48±3.05 | 30.72±1.19 | 23.07±1.04 | 11.89±0.17 | 8.66±0.20 | 3.90±0.07 | 2.68±0.04 |
| (+) CONAN | 50.45±2.70 | 52.35±0.99 | 31.04±1.26 | 23.90±1.35 | 22.36±0.34 | 19.13±0.32 | 6.71±0.15 | 5.48±0.13 |
| CCG (Hammoud et al., 2024) | 45.49±2.81 | 45.29±1.69 | 30.20±1.91 | 23.44±0.71 | 12.55±0.22 | 10.21±0.26 | 4.03±0.10 | 2.91±0.10 |
| (+) CONAN | 46.65±3.36 | 45.75±1.92 | 31.14±1.88 | 25.77±1.18 | 18.32±0.42 | 15.83±0.34 | 5.78±0.17 | 4.70±0.14 |
| HIRPG | 51.36±2.59 | 51.63±2.49 | 34.12±1.27 | 28.18±1.32 | 27.72±0.30 | 23.71±0.39 | 10.70±0.19 | 8.75±0.13 |
| (+) CONAN (**Ours**) | **55.89±3.06** | **55.43±2.49** | **38.53±1.15** | **33.73±1.82** | **35.60±0.31** | **29.99±0.11** | **14.53±0.22** | **12.65±0.09** |
| MA | 67.10±4.07 | 61.95±0.92 | 27.75±1.44 | 20.90±0.95 | 51.13±0.28 | 42.95±0.15 | 13.48±0.09 | 10.69±0.07 |

Table 2: **Quantitative comparison between different name-only baselines on CIL setup.** MA refers to training a model with manually annotated data. The combination of HIRPG and CONAN refers to our proposed GenOL. We bolded the highest accuracy, excluding the MA data (*i.e.*, ideal scenario). We provide details about ID and OOD setups for each domain-generalization benchmark in Section A.2.

| Method | Bongard-HOI | | | | Bongard-OpenWorld | | | |
| --- | --- | --- | --- | --- | --- | --- | --- | --- |
| | Positive / Negative | | Concept Answering | | Positive / Negative | | Concept Answering | |
| | $A_{\text{AUC}}$ ↑ | $A_{last}$ ↑ | $A_{\text{AUC}}$ ↑ | $A_{last}$ ↑ | $A_{\text{AUC}}$ ↑ | $A_{last}$ ↑ | $A_{\text{AUC}}$ ↑ | $A_{last}$ ↑ |
| C2C (Prabhu et al., 2024) | 61.53±3.13 | 59.58±2.49 | 73.88±3.21 | 67.40±3.15 | 49.75±0.49 | 50.39±0.89 | 69.56±3.58 | 67.56±1.47 |
| IE (Li et al., 2023b) | 62.26±2.79 | 62.42±2.51 | 74.22±3.03 | 67.51±2.26 | 48.79±0.77 | 51.05±0.72 | 68.95±2.35 | 68.86±1.85 |
| Seafaring (Sato, 2023) | 61.38±2.55 | 62.25±1.98 | 72.66±2.84 | 67.02±2.46 | 49.06±0.87 | 48.95±0.96 | 66.76±2.72 | 65.20±1.02 |
| Tiara (Sato, 2022) | 59.65±2.38 | 60.55±2.12 | 70.56±2.87 | 66.14±2.76 | 47.01±2.21 | 46.34±1.91 | 64.22±2.27 | 63.38±2.82 |
| Glide-Syn (He et al., 2023) | 54.83±2.07 | 55.77±3.54 | 67.87±3.30 | 59.38±3.62 | - | - | - | - |
| LE (He et al., 2023) | 64.03±3.10 | 62.40±2.58 | 73.65±3.60 | 70.68±3.80 | - | - | - | - |
| (+) CONAN | 65.90±2.59 | 65.63±2.59 | 74.99±3.07 | 72.38±2.76 | - | - | - | - |
| HIRPG | 67.25±2.61 | 71.49±0.42 | 75.52±3.17 | 73.97±3.11 | 48.37±1.17 | 47.48±3.47 | 70.09±1.92 | 74.59±3.11 |
| (+) CONAN (Ours) | **70.20±3.97** | **73.18±2.40** | **77.01±3.45** | **75.80±1.83** | **53.68±1.18** | **57.74±2.18** | **73.10±3.79** | **76.77±3.81** |
| MA | 69.50±1.84 | 73.04±2.71 | 76.02±3.85 | 70.37±3.87 | 53.44±1.91 | 53.06±3.45 | 70.84±3.44 | 72.21±3.75 |

Table 3: **Quantitative comparison between different name-only baselines on Multi-modal MV-CIL setup.** GenOL outperforms baselines, as well as manual annotations (MA). We provide details on Positive/Negative and Concept Answering tasks in Section A.1.

**Effectiveness of GenOL in name-only CIL setup.** We compare models trained with GenOL, baselines, and MA data (oracle case), and summarize the results in Table 1 and Table 2. In the ID domain, MA outperforms baselines and GenOL, since the MA data test set is used as the test set in the ID domain, making the training and test sets come from the same domain, thus serving as an oracle.s However, in the OOD domain, GenOL outperforms both MA and baselines. We believe that GenOL achieves better generalization by generating more diverse images through HIRPG and CONAN. Additional comparisons with various combinations of prompt generation baselines and ensemble methods are in Section A.31.

| Method | ID | | OOD | |
| --- | --- | --- | --- | --- |
| | $A_{\text{AUC}}$ ↑ | $A_{last}$ ↑ | $A_{\text{AUC}}$ ↑ | $A_{last}$ ↑ |
| IE | 9.94 | 8.26 | 6.25 | 5.10 |
| C2C | 9.25 | 8.05 | 6.38 | 4.78 |
| Glide-Syn | 6.25 | 3.53 | 5.16 | 2.73 |
| SC | 13.92 | 10.65 | 5.83 | 4.38 |
| LE | 12.59 | 9.29 | 10.03 | 7.86 |
| CCG | 12.43 | 8.84 | 5.62 | 5.07 |
| CHB | 12.30 | 9.14 | 7.19 | 6.29 |
| CCG | 12.43 | 8.84 | 5.62 | 5.07 |
| GenOL (Ours) | **14.18** | **11.39** | **10.73** | **9.64** |
| MA | 30.01 | 25.03 | 6.90 | 4.95 |

Table 1: **Qualitative comparison on ImageNet-R.** GenOL outperforms baselines in the ID domain and even outperforms MA in the OOD domain.

**Effectiveness of GenOL in name-only MVCIL setup.** We also empirically validate the effectiveness of GenOL in the name-only MVCIL setup and summarize the results in Table 3. CHB, SC, and CCG are excluded due to their focus on image classification, and Glide-Syn and LE, which use word-to-sentence

| Method | Full Dataset Training | PACS | | | | DomainNet | | | |
|---|---|---|---|---|---|---|---|---|---|
| | | ID | | OOD | | ID | | OOD | |
| | | $A_{\text{AUC}}\uparrow$ | $A_{last}\uparrow$ | $A_{\text{AUC}}\uparrow$ | $A_{last}\uparrow$ | $A_{\text{AUC}}\uparrow$ | $A_{last}\uparrow$ | $A_{\text{AUC}}\uparrow$ | $A_{last}\uparrow$ |
| CONAN (**Ours**) | ✗ | **55.89±3.06** | **55.43±2.49** | **38.53±1.15** | **33.73±1.82** | **34.60±0.31** | **30.09±0.11** | **14.53±0.22** | **12.65±0.09** |
| No ensembling | ✗ | 51.36±2.59 | 51.63±2.49 | 34.12±1.27 | 28.18±1.32 | 27.72±0.30 | 23.71±0.39 | 10.70±0.19 | 8.75±0.13 |
| EWS | ✗ | 50.56±2.32 | 50.03±2.13 | 34.59±1.41 | 27.13±3.44 | 32.38±0.47 | 26.45±0.35 | 12.93±0.23 | 10.92±0.06 |
| Moderate (ICLR 2023) | ✗ | 47.03±3.52 | 45.34±1.11 | 35.06±2.03 | 27.91±2.17 | 25.57±0.42 | 20.38±0.16 | 10.53±0.29 | 8.17±0.13 |
| LDM-S (ICLR 2024) | ✓ | 53.16±2.06 | 43.32±1.76 | 36.92±2.10 | 26.32±1.01 | 30.68±0.36 | 26.61±0.37 | 12.29±0.15 | 10.45±0.10 |
| Uncertainty (ICLR 2020) | ✓ | 39.75±2.10 | 33.17±3.69 | 32.99±1.42 | 25.17±3.01 | 21.90/±0.37 | 15.70±0.08 | 10.01±0.23 | 7.19±0.11 |
| CRAIG (ICML 2020) | ✓ | 53.57±2.43 | 54.24±2.04 | 35.54±0.90 | 32.29±0.96 | 32.53±0.20 | 28.44±0.23 | 13.25±0.15 | 11.53±0.06 |
| Glister (AAAI 2021) | ✓ | 40.55±2.43 | 37.75±3.81 | 34.30±1.66 | 27.56±1.31 | 23.16±0.37 | 16.98±0.35 | 10.56±0.26 | 7.60±0.18 |
| GradMatch (ICML 2022) | ✓ | 54.93±3.24 | 54.06±1.49 | 35.05±1.70 | 29.81±1.35 | 32.53±0.43 | 28.36±0.41 | 13.48±0.31 | 11.74±0.18 |
| Adacore (ICML 2022) | ✓ | 52.06±2.64 | 48.37±2.80 | 35.55±2.09 | 30.36±0.85 | 32.15±0.55 | 26.83±0.18 | 13.62±0.27 | 11.37±0.04 |
| AL (NeurIPSW 2022) | ✓ | 54.49±1.57 | 52.82±2.39 | 36.98±1.86 | 30.44±1.99 | 28.92±0.33 | 25.78±0.20 | 10.55±0.21 | 9.24±0.14 |
| LCMat (AISTATS 2023) | ✓ | 53.40±2.35 | 54.60±1.65 | 35.37±1.62 | 30.04±0.82 | 32.38±0.44 | 28.36±0.32 | 13.42±0.26 | 11.76±0.17 |

Table 4: **Quantitative comparison between data selection methods.** EWS combines generated data from multiple models in equal proportions, while No Ensembling uses a single model (*i.e.*, SDXL).

models, are inapplicable to Bongard-OpenWorld, which has sentence-form concepts. As shown in the table, GenOL effectively trains LLaVA-1.5-7B, outperforming baselines and even compared to MA data.

Note that while C2C, IE, and MA data utilize human-annotated hard negative concepts (*e.g.*, 'hold a bike') alongside positive concepts (*e.g.*, 'ride a bike'), GenOL relies solely on positive concepts. Specifically, for MA data, high-quality annotators from Amazon Mechanical Turk were employed to select hard negative examples and filter out noisy data (Jiang et al., 2022). For web-scraped data, long-context queries (*e.g.*, *'hard negative images of riding a bike'*) often retrieve noisy images, thus, it requires manually annotated negative concepts instead. In contrast, GenOL automatically selects relevant hard negative concepts based on the given concept, leveraging commonsense priors from LLMs (Zhao et al., 2023; Yang et al., 2024).

Even without manual annotation, GenOL outperforms models trained with both MA and web-scraped data by leveraging LLMs' commonsense prior and the controllability of T2I generative models (Nie et al., 2021). We provide prompts that we employ to select hard negative concepts in Section A.6.

**Effectiveness of GenOL in online learning setup.** We further validate that GenOL outperforms baseline methods even in online learning setups without distribution shifts (*i.e.*, where all target concepts are presented at once). We provide detailed results in Sec. A.28 for space's sake.

**Comparison of HIRPG with Prompt Generation Methods.** To validate the effectiveness of HIRPG in diverse prompt generation, we compare models trained on data generated from prompts derived by prompt generation baselines (LE, CHB, SC, and CCG). As shown in Table 2, HIRPG significantly outperforms the baselines, with and without CONAN. We attribute this improvement to two key components of HIRPG: recurrent prompt generation (RPG), which mitigates prompt overlap, and hierarchical generation (HIG), which alleviates the lost-in-the-middle issue (Liu et al., 2023c) that arises with RPG alone. An ablation of these components, along with Diversity (Naeem et al., 2020) and Recognizability (Fan et al., 2024) analyses, is provided in Section A.5.

**Comparison of CONAN with Data Ensemble Methods.** To demonstrate the effectiveness of CONAN, we compare it with existing data ensemble methods, as well as the equal-weight selection (EWS). For a fair comparison, we ensure an equal number of selected images in the ensemble set across all ensemble methods. After data selection, we evaluate the performance of continual learners trained with each ensemble set and summarize the results in Table 4. For sample selection, we use a CLIP-pretrained ResNet-50 as the feature extractor $g$, following Cui et al. (2023). Note that Uncertainty, Glister, GradMatch, and LCMat require fine-tuning the feature extractor on the full dataset for gradient calculations, even with a pre-trained feature extractor. As a result, for these baselines, we fine-tune $g$ for 30 epochs using the full dataset. In contrast, Moderate and CONAN do not require fine-tuning, as they only use $g$ to calculate feature distances. Despite being training-free, CONAN outperforms methods that require fine-tuning, as well as Moderate.

## 5.3 Analysis of Bias

We analyze gender, race, and geographical bias in web-scraped data (*i.e.*, C2C) and generated data from our proposed GenOL and other generative baselines (*i.e.*, LE, CHB, SC, and CCG), and manually annotated

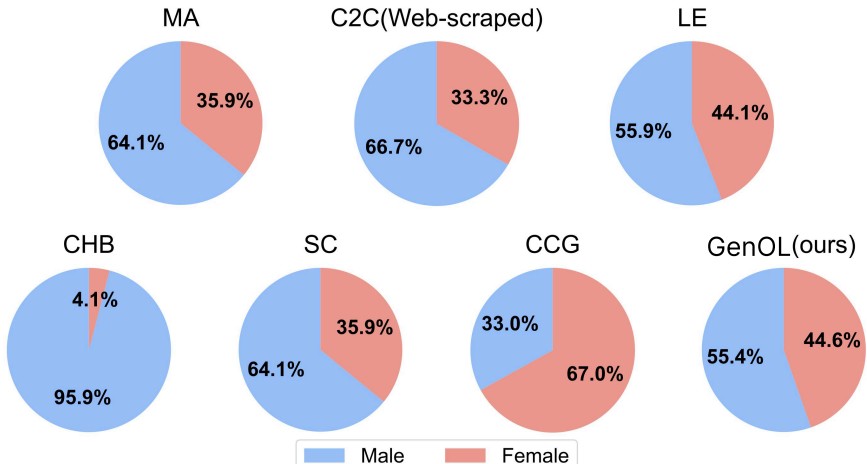

Figure 4: **Comparison of gender bias.** GenOL generates data with reduced gender bias compared to baselines, and even compared to MA data.

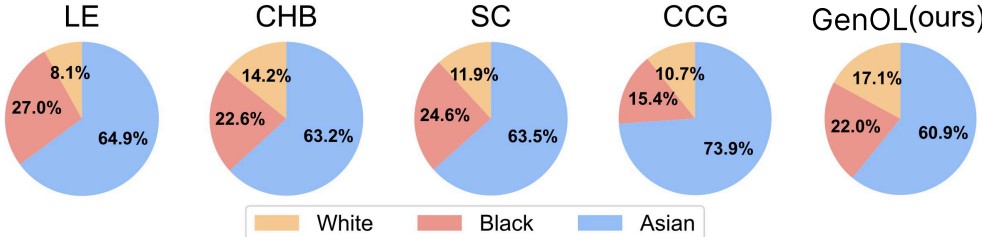

Figure 5: **Comparison of race bias.** GenOL generates data with reduced race bias compared to baselines.

(MA) data. To measure bias, we calculate the similarity between text embeddings of gender/race attribution keywords and image embeddings, following Mandal et al. (2023). Attribution keywords are summarized in Sec. A.20. Based on this similarity, we assign each image to its closest gender or race category and compare the number of images across these categories, following Wan et al. (2024); He et al. (2024).

**Gender/Race Bias.** We first compare gender bias among GenOL, web-scraped data (*i.e.*, C2C), and manually annotated (MA) data, and summarize the results in Figure 4. As shown in the figure, GenOL exhibits less bias in both gender and race compared to C2C and MA. Next, we compare the race biases with generative baselines, including GenOL, and we summarize the results in Figure 5. As depicted in the figure, GenOL demonstrates the least bias compared to other generative baselines. We attribute this to our proposed components, HIRPG and CONAN, which successively enhance intra- and inter-diversity of generated data, respectively, thereby mitigating bias issues. To this end, we believe that training with data generated by GenOL is unlikely to introduce significant biases, even compared to MA data.

**Geographical bias.** We also measure the geographical bias of objects, evaluating whether the generated content reflects artifacts and surroundings from across the globe, rather than disproportionately representing certain regions Basu et al. (2023). We provide the results in Section A.20 for spaces' sake.

## 5.4 Qualitative Analysis

We qualitatively compare data acquisition methods in the Bongard-HOI benchmark in Figure 6. Although the desired positive images are related to 'ride a bike', C2C, a web-scrapping baseline, includes irrelevant images, such as 'sitting on a bike' or a road with a bicycle symbol, in the positive set. These noises can significantly hinder model performance (Kim et al., 2021; Bang et al., 2022). In contrast, GenOL effectively generates the desired output and hard negative examples through text descriptions, leveraging the controllability (Nie et al., 2021) of the generative model, thus generating results similar to MA data.

## 5.5 Comparison of Computational and Memory Cost

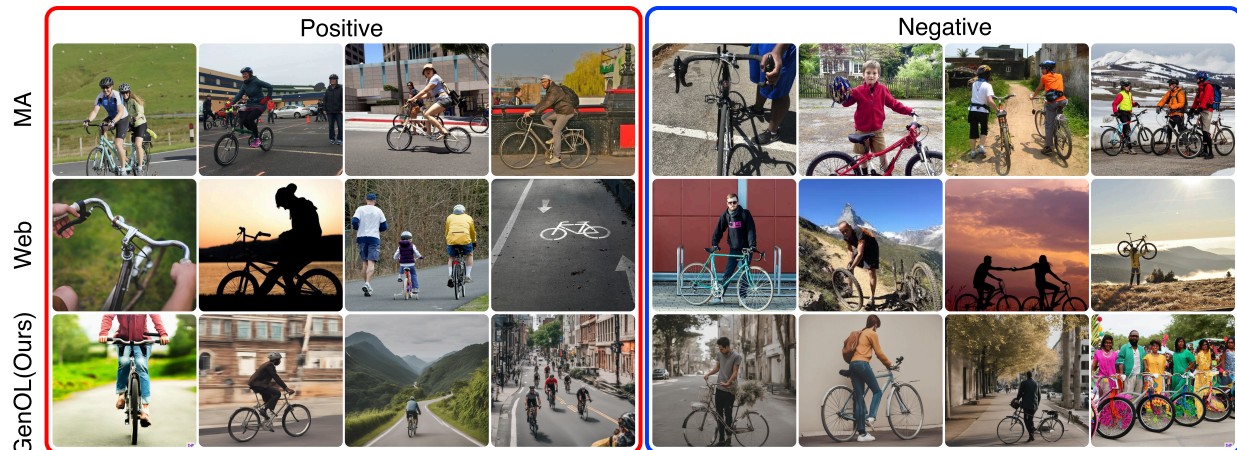

Figure 6: **Samples using different data acquisition methods for the same concept in the MVCIL setup.** The given concept is *'ride a bike'* from the Bongard-HOI benchmark. The left four images represent positive examples that depict the given concept, while the right four images represent negative examples that illustrate different concepts. Here, 'MA' refers to manually annotated data.

We compare computation and memory cost, and extra resources in Table 5. MA data requires 3,000,000 minutes of human working hours to filter out outliers. Web scraping baselines (*i.e.*, C2C and IE) do not require any GPU resources; instead, they rely on web browsers for crawling. Generative baselines, such as Glide-Syn, LE, CHB, SC, CCG, and our proposed GenOL, utilize 32 RTX 4090 GPUs for image generation. Additionally, these methods require 12GB of memory to store the weights of generative models.

| Method | Wall time (min) | Storage (GB) | Extra resources |
|---|---|---|---|
| MA | 3,000,000 | 0 | Human annotation |
| IE | 330 | 0 | Web browsers |
| C2C | 300 | 0 | |
| Glide-Syn | 254 | 4.5 | 32 × RTX 4090 GPUs |
| Real-Fake | 480 | 32 | |
| LE | 240 | | |
| CHB | 222 | | |
| SC | 952 | 12 | 32 × RTX 4090 GPUs |
| CCG | 952 | | |
| GenOL | 182 | | |

Table 5: **Comparison of computational and memory budget, and extra resources for acquiring DomainNet.**

The major reason for the lower wall time of GenOL compared to other generative baselines is the use of class-agnostic prompts, which make its prompt generation time independent of the total number of concepts, as mentioned in Section A.22.

## 5.6 Ablation Study

We conduct an ablation study on two components of GenOL, *i.e.*, HIRPG and CONAN using the ImageNet-1k pretrained ViT-Base models, and summarize the results in Table 6. Our findings show that both components significantly enhance ID and OOD domain performance. Additional ablation studies using ResNet-18 and HIRPG (comprising HIG and RPG) are provided in Sections A.7 and A.8, respectively.

We provide detailed studies in the appendix for the space's sake. Specifically, we explore fine-grained benchmarks (Section A.21), hyperparameter analysis (Section A.23), qualitative analysis of both prompts generated by HIRPG (Section A.34) and images generated by GenOL (Section A.29), and details of RMD calculation (Section A.32).

## 6 Conclusion

Online CL represents a practical learning paradigm, but the assumption of well-curated and annotated data limits its real-world applicability. To address this, we introduce a unified name-only CL framework that integrates generators with the continual learner, termed 'Generative name only Online Learning' (GenOL). Within the GenOL framework, we propose a diverse prompt generation method (*i.e.*, HIRPG) and complexity-guided ensembling (*i.e.*, CONAN). Extensive experiments validate the performance im-

| Method | PACS | | | | DomainNet | | | |
| --- | --- | --- | --- | --- | --- | --- | --- | --- |
| | ID | | OOD | | ID | | OOD | |
| | $A_{\text{AUC}}$ ↑ | $A_{last}$ ↑ | $A_{\text{AUC}}$ ↑ | $A_{last}$ ↑ | $A_{\text{AUC}}$ ↑ | $A_{last}$ ↑ | $A_{\text{AUC}}$ ↑ | $A_{last}$ ↑ |
| Vanilla GenOL | 72.91±1.40 | 56.85±2.68 | 40.39±1.67 | 27.11±2.90 | 30.96±0.34 | 22.52±0.46 | 11.17±0.25 | 7.78±0.21 |
| (+) HIRPG | 78.52±1.90 | 72.40±2.40 | 45.46±1.59 | 36.76±2.35 | 37.90±0.31 | 30.37±0.64 | 15.30±0.19 | 11.31±0.29 |
| (+) CONAN | 77.31±1.58 | 64.39±2.40 | 48.01±2.05 | 35.22±2.64 | 37.81±0.47 | 30.15±0.25 | 14.61±0.29 | 10.83±0.20 |
| (+) HIRPG & CONAN (**Ours**) | **79.32**±1.97 | **72.46**±0.42 | **53.88**±1.57 | **41.31**±2.42 | **42.73**±0.25 | **36.09**±0.50 | **18.64**±0.28 | **14.68**±0.16 |

Table 6: **Ablations for proposed components of GenOL.** Vanilla GenOL denotes generating 50 different prompts using an LLM without employing RPG or HIG, and using a single T2I generator, *i.e.*, SDXL.

provements achieved by both components of the GenOL framework, demonstrating its effectiveness in both ID and OOD domains compared to webly-supervised and human supervision.

# 7 Limitations and Future Work

While GenOL replaces manual annotation by generating diverse images that only require *concepts* using HIRPG and CONAN, we acknowledge the limitations arising from the use of generative models. Here, we outline these limitations and propose future directions to address them.

**Scalability.** Recent studies (Yuan et al., 2024; Hammoud et al., 2024) demonstrate that training with text-to-image generated samples (10-100× more than real data) achieves comparable performance to real data training. However, such scaling is impractical for name-only CL setup, as they require substantial time and computational cost, while CL requires fast adaptation (Koh et al., 2021; Ghunaim et al., 2023). In practice, there are various applications where a few manually annotated samples are provided along with concept names. Prior work Yuan et al. (2024) demonstrates that training with mixed real and synthetic data enhances performance while reducing computational costs. Therefore, future work could investigate combining real data with GenOL-generated data to address scalability challenges more effectively.

**Privacy and Copyright Concerns.** Generative models prevent potential data leakage in replay-based CL by eliminating the need to store real data in episodic memory (Shin et al., 2017; Liu et al., 2024). However, they introduce privacy concerns through potential memorization of training data (Wang et al., 2023; Carlini et al., 2023). To address immediate risks, we exclude person-related generated data from our released dataset. We believe that advancing privacy-preserving (Xu et al., 2023; Chen & Yan, 2024) and differential privacy (Chen et al., 2023a) generative models would effectively mitigate these concerns.

**Capability of Generation.** As mentioned in Section 1, GenOL and other generative baselines focus on name-only CL scenarios where new concepts are new to specific systems or domains, rather than entirely novel concepts globally. Therefore, we leverage generative models' large coverage of everyday concepts, but this reliance limits their performance on completely new concepts absent from their training data

To handle such entirely unseen concepts, generative models require continual adaptation with a few real examples of the new concept. As shown in Section A.26, even for previously seen concepts, continual adaptation of the generative model with a few real data significantly improved both ID and OOD performance. By extension, we argue that providing a few real samples for completely unseen concepts can enable continual training of generative models, thereby generating abundant synthetic data for these new concepts. We believe that advances in efficient continual training methods for generative models (Smith et al., 2024; Song et al., 2024a; Uehara et al., 2024) will further accelerate GenOL 's ability to accommodate unseen concepts.

Moreover, GenOL 's flexibility, allowing for seamless replacement of the underlying foundation model, ensures it evolves alongside advances in foundation models (*e.g.*, GPT-3.5 → GPT-4o → GPT-5). This expansion enables the system to acquire data for an increasingly broad range of concepts over time. As an example of the flexibility, Section A.23 shows that even when GPT-4o is replaced with LLaMA3-8B or Qwen3-8B, GenOL consistently outperforms the baselines, demonstrating its potential to sustain strong performance with future models. In other words, while GenOL does not directly implement continual adaptation, its flexible design ensures it advances alongside evolving foundation models, evolves alongside evolving foundation models.

## 8 Acknowledgement

This work was partly supported by the IITP grants (No.RS-2022-II220077, No.RS-2022-II220113, No.RS-2022-II220959, No.RS-2022-II220871, No.RS-2021-II211343 (SNU AI), No.RS-2021-II212068 (AI Innov. Hub), No. RS-2025-25442338 (AI Star Fellowship-SNU), RS-2022-II220951) funded by the Korea government (MSIT) and a grant of Korean ARPA-H Project through the Korea Health Industry Development Institute (KHIDI), funded by the Ministry of Health & Welfare, Republic of Korea (RS-2025-25424639).

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

# A  Appendix

## A.1  Details about Multi-modal Visual-Concept Incremental Learning Setup

Beyond CIL setups, we also assess GenOL in multimodal tasks, such as context-dependent visual reasoning tasks, focusing on Bongard-HOI (Jiang et al., 2022) and Bongard-OpenWorld (Wu et al., 2024a). These benchmarks are based on two desirable characteristics of classical Bongard problems: (1) few-shot concept learning and (2) context-dependent reasoning. The former refers to the ability to induce visual concepts from a small number of examples, while the latter indicates that the label of a query image may vary depending on the given context (*i.e.*, positive and negative support set). Specifically, as shown in Figures 7 and 8, given a positive support set and a negative support set for a particular concept (*e.g.*, "ride a bike"), we consider two types of tasks that address the following queries: (1) *"What is the concept exclusively depicted by the positive support set?"* and (2) *"Given a query image, does the query image belong to the positive or negative support set?"* We refer to these tasks as CA (Concept Answering) and P/N, respectively. In addition, we provide a detailed description of each visual concept reasoning benchmark.

**Bongard-HOI (Jiang et al., 2022).**  Bongard-HOI denotes a concept $c = \langle a, o \rangle$ as a visual relationship tuple, where $a$, $o$ are the class labels of action and object, respectively. Following Bongard's characteristic, there are positive support set $\mathcal{I}_p$ and negative support set $\mathcal{I}_n$, where $\mathcal{I}_p$ and $\mathcal{I}_n$ have different concepts. Specifically, if the concept of $\mathcal{I}_p$ is $\langle a, c \rangle$, $\mathcal{I}_n$ is composed of data with concept $c' = \langle \bar{a}, o \rangle$, where $\bar{a} \neq a$. As a result, images from both $\mathcal{I}_n$ and $\mathcal{I}_p$ contain the same categories of objects, with the only difference being the action labels, making it impossible to trivially distinguish positive images from negative ones through visual recognition of object categories alone (*i.e.*, hard negative examples). We provide examples of Bongard-HOI-CA & Bongard-HOI-P/N (Jiang et al., 2022) in Figure 7.

**Bongard-OpenWorld (Wu et al., 2024a).**  In contrast to Bongard-HOI, which has a structured concept $c$ represented as (action, object), Bongard-OpenWorld utilizes a free-form sentence as $c$ to describe the content depicted by all images in the positive set $\mathcal{I}_p$ exclusively. Specifically, concepts are obtained by the annotators, who are instructed to write visual concepts by following a predefined set of categories. We provide examples of Bongard-OpenWorld-CA & Bongard-OpenWorld-P/N (Wu et al., 2024a) in Figure 8.

Note that since the input consists of both text queries and images (*i.e.*, support sets and a query image) and outputs are sentences, we use multimodal large language models (MLLMs), such as LLaVA (Liu et al., 2023b), which connects a vision encoder with an LLM for general-purpose visual and language understanding. For further implementation details, such as the prompts we use, see Section A.4.

## A.2  Details about Experiment Setup

To set a domain generalization benchmarks (*i.e.*, PACS (Zhou et al., 2020), DomainNet (Neyshabur et al., 2020), ImageNet-R (Hendrycks et al., 2021) and CIFAR-10-W (Sun et al., 2024)) for a class incremental learning (CIL) setup, we divide it into multiple disjoint tasks. We assume a disjoint setup (Parisi et al., 2019), where tasks do not share classes. We summarize the in-distribution (ID) domain, the out-of-distribution (OOD) domains, the total number of classes per dataset, the number of classes per task, and the number of tasks for each dataset in Table 7. Within each dataset, all tasks have the same size, except PACS, which has a total of 7 classes. For PACS, the first task includes 3 classes, while the subsequent tasks include data for 2 classes each. For CIFAR-10-W, even though CIFAR-10 Krizhevsky et al. (2009) can use MA data, the image resolution of CIFAR-10 is 32×32, while CIFAR-10-W has a resolution of 224×224, leading to performance degradation. Therefore, we exclude comparison with MA in the CIFAR-10-W experiments. For multi-modal visual-concept incremental learning (MVCIL) setup, we summarize the total number of concepts, number of tasks, and number of concepts per task in Table 8.

Note that we run five different task splits using five different random seeds and report the average and standard error of the mean (SEM) for all experiments, except for ImageNet-R due to computational resource constraints, following (Koh et al., 2023; Seo et al., 2024).

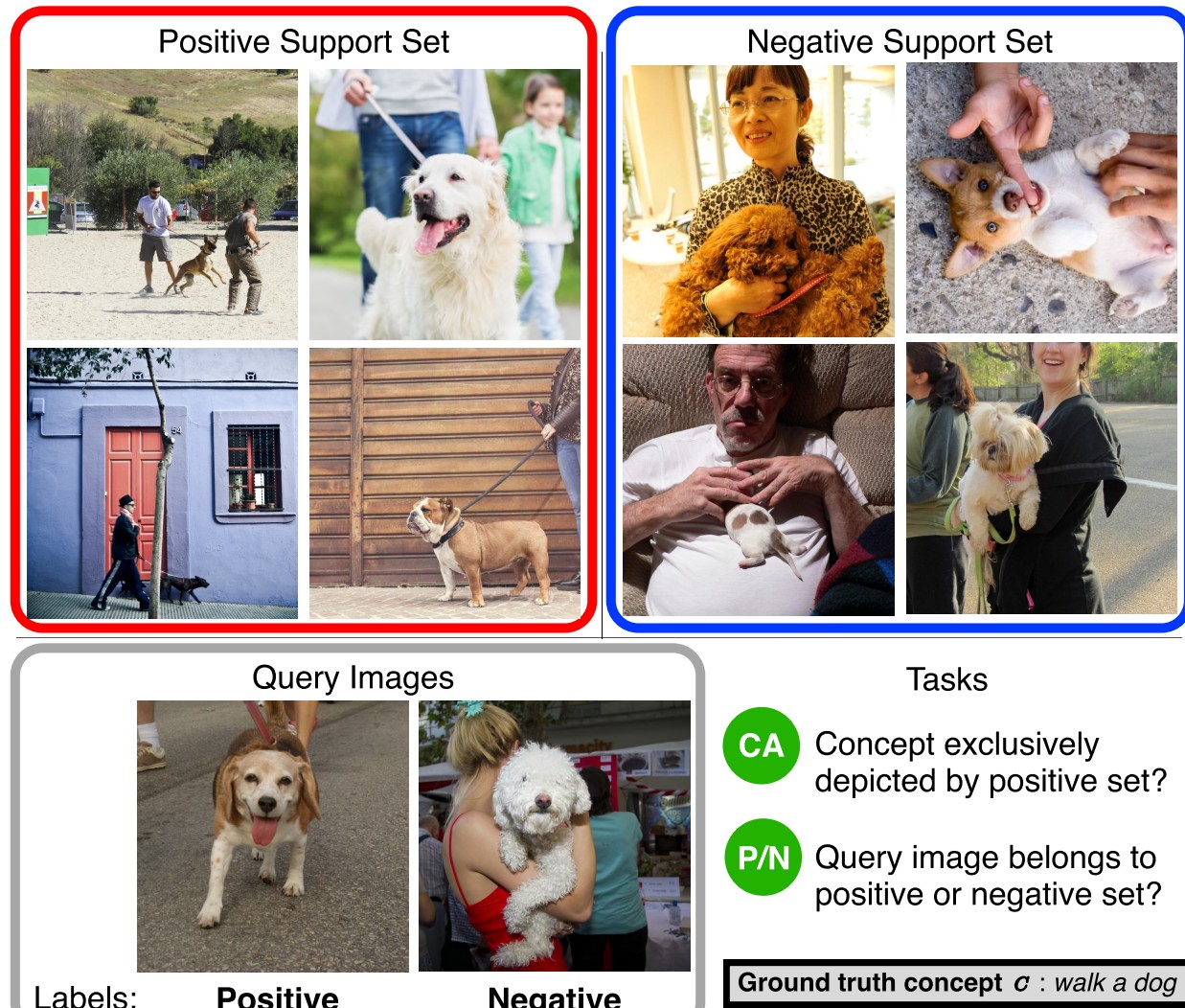

Figure 7: **An example of the Bongard-HOI task.** CA refers to the concept answering task, while P/N refers to the classifying whether a query image belongs to the positive or negative set.

| Dataset | ID domain | OOD domains | total # of classes | # of tasks | # of classes / task |
|---------|-----------|-------------|--------------------|------------|---------------------|
| PACS | Photo | Art, Cartoon, Sketch | 7 | 3 | One task: 3, Others: 2 |
| CIFAR-10-W | - | CIFAR-10-W | 10 | 5 | 2 |
| DomainNet | Real | Clipart, Painting, Sketch | 345 | 5 | 69 |
| ImageNet-R | ImageNet | ImageNet-R | 1000 | 5 | 20 |

Table 7: **Task configurations for the CIL setup on each domain generalization dataset.** We split

| Dataset | Form of Concepts | total # of concepts | # of tasks | # of concepts / task |
|---------|------------------|---------------------|------------|----------------------|
| Bongard-OpenWorld | Free-form | 10 | 5 | 2 |
| Bongard-HOI | (action, object) | 50 | 5 | 10 |

Table 8: **Task configurations for the MVCI setup on each Bongard benchmark.**

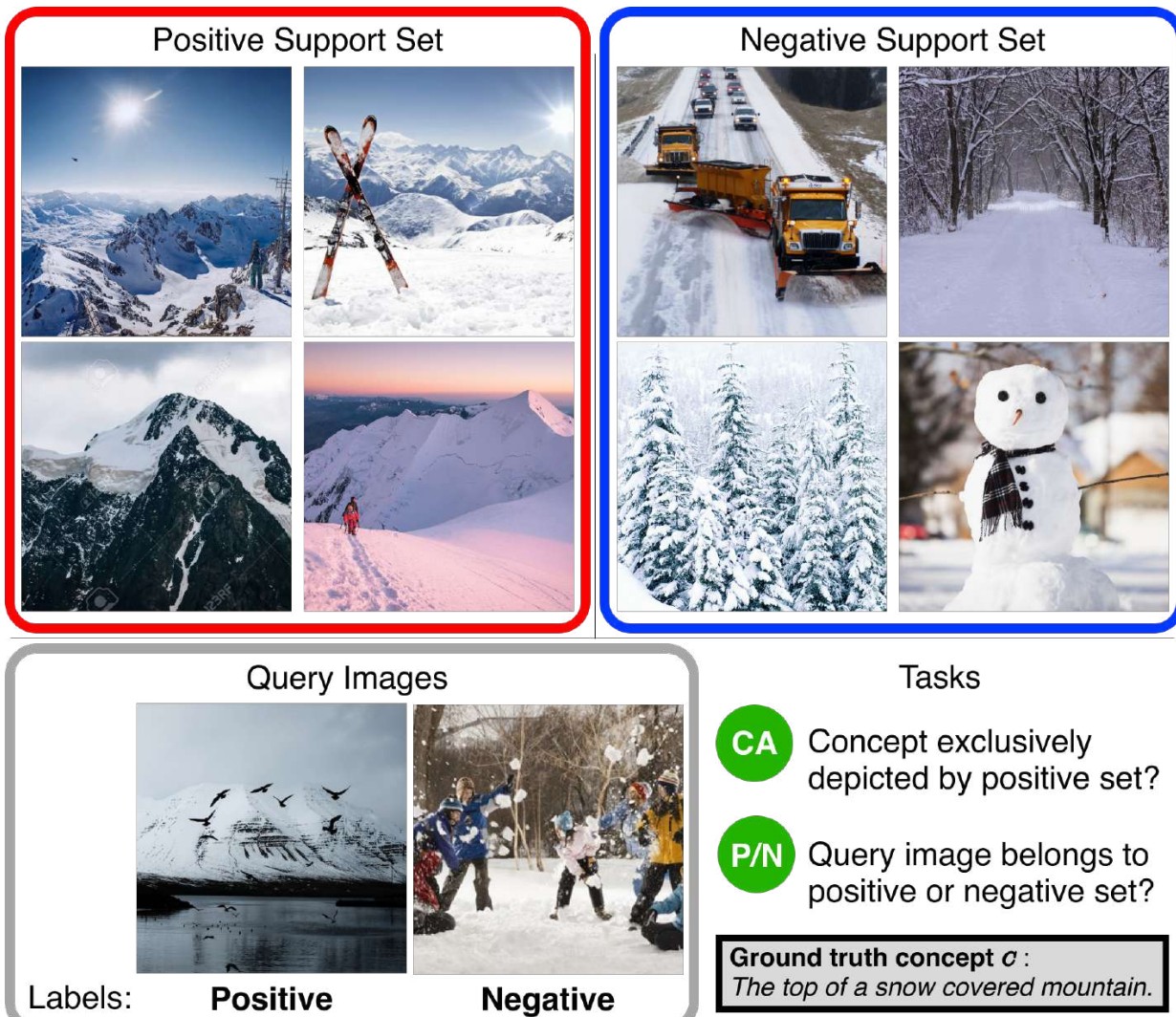

Figure 8: **An example of the Bongard-OpenWorld task.** CA refers to the concept answering task, while P/N refers to the classifying whether a query image belongs to the positive or negative set. The concept $c$ is free-form, such as sentences.

### A.3 Prompt Diversification and Concept-specific Prompt Generation Baselines

For a fair comparison, we used the same number of diversified prompts (*i.e.*, 50) for all prompt diversification methods, including our proposed HIRPG. Moreover, for all LLM-based prompt generators, we consistently used GPT-4o (Wu et al., 2024b) as LLM.

**LE (He et al., 2023).** LE leverages an off-the-shelf word-to-sentence T5 model, pre-trained on the "Colossal Clean Crawled Corpus" dataset (Raffel et al., 2020) and fine-tuned on the CommonGen dataset (Lin et al., 2020), to increase the diversity of language prompts and generated images, with the aim of better harnessing the potential of synthesized data. Specifically, the concepts are entered into the word-to-sentence model, which generates diversified sentences containing the concept. These diversified sentences are then used as prompts for the text-to-image generation process.

**CCG (Hammoud et al., 2024).** For a given concept set $C$, CCG (Concept-based Captions Generation) uses an LLM generator $G_{LLM}$ to produce concept-specific prompts for T2I models. The designed prompt $p$ for each concept $c \in C$ is structured as follows:

> Your task is to write me an image caption that includes and visually describes a scene around a concept. Your concept is $c$. Output one single grammatically correct caption that is no longer than 15 words. Do not output any notes, word counts, facts, etc. Output one single sentence only.

Formally, the set of generated captions for concept c can be defined as $T = \{t_{c,n} \sim G_{\text{LLM}}(p, c)\}, \forall c \in C, \forall n \in \{1, 2, ..., N\}$, where $N$ is the number of desired prompts for each concept.

**CHB (Sarıyıldız et al., 2023).** To increase the visual diversity of the output images, CHB (Concept Name + Hypernym + Background) combines background information along with hypernyms, which helps reduce semantic ambiguity. They assume that class $c$ can be seen 'inside' a scene or background. Therefore, to enhance visual diversity, CHB combines the concept name and its hypernym (as defined by the Word-Net (Miller, 1995) graph) with scene classes from the Places365 dataset (López-Cifuentes et al., 2020) as background for each concept. Formally, $p_c$ can be defined as "$p_c = c$, $h_c$ inside $b$", where $c$ refers to the concept name, $h_c$ refers to the hypernym of the concept $c$, and $b$ refers to the background.

**SC (Tian et al., 2024a).** SC (Synthesizing Captions) considers three types of templates for each concept $c$: $c \rightarrow caption$, $(c, bg) \rightarrow caption$, and $(c, rel) \rightarrow caption$.

- $c \rightarrow caption$. They generate a sentence directly from the concept name $c$ using LLM.

- $(c, bg) \rightarrow caption$. Similar to CHB (Sarıyıldız et al., 2023), they combine the visual concept $c$ with a background $bg$. However, while CHB randomly selects $bg$ from the Places365 dataset, they generate a list of suitable backgrounds for the chosen concepts using LLM, which helps avoid unlikely combinations, such as "a blue whale on a football field".

- $(c, rel) \rightarrow caption$. They consider pairing a given visual concept $c$ with a positional relationship word $rel$, such as "in front of". To add variety, $rel$ is randomly selected from a predefined set of 10 relationship words. Using an LLM, they then generate captions that reflect the selected relationship word in relation to the concept.

**Real-Fake (Yuan et al., 2024).** Real-Fake aligns both data and concept-conditional distributions through Maximum Mean Discrepancy (MMD)-based loss (Gretton et al., 2006) to minimize the discrepancy between real and synthetic data distributions. For prompt generation, it leverages BLIP2 (Li et al.) to generate captions that incorporate concept-relevant information. These captions are combined with concept names and intra-concept visual features to guide the generation process.

**IE (Li et al., 2023b).** IE (Internet Explorer) dynamically queries search engines by combining concepts from the WordNet hierarchy (Miller, 1995) with descriptors generated by GPT-J (Wang & Komatsuzaki, 2021). For instance, the concept 'duck' can be combined with descriptors like 'baby' or 'red', resulting in queries such as 'baby duck' or 'red duck'. The descriptors are generated using a prompt template like "The {concept} is [descriptor]" and sampled with temperatures to ensure diversity. The retrieved images are filtered based on their similarity to the target dataset using a relevance reward metric.

### A.4 Implementation Details

We use ResNet18 (He et al., 2016) and Vision Transformer (ViT) (Dosovitskiy & Brox, 2016) as network architectures for the class-incremental learning (CIL) setup. Due to the large number of parameters in ViT, training it from scratch in an online setup resulted in lower performance. Therefore, we used the weights of a model pre-trained on ImageNet-1K (Russakovsky et al., 2015) as initial weights for ViT. For data augmentation, we consistently applied RandAugment (Cubuk et al., 2020) in all experiments. For task split, we adopt a disjoint setup, where tasks do not share classes (Parisi et al., 2019). We used the GPT-4 model (Achiam et al., 2023) for all LLM-based prompt generation baselines, including HIRPG, for a fair comparison. Furthermore, to ensure a fair comparison among manually annotated data, generated data, and

web-scraped data, we used an equal number of data in all experiments. Regarding the web-scraped data, we first acquire 20% more samples than necessary to filter out noisy data, following (Prabhu et al., 2024). To achieve this, we utilized pre-trained CLIP (Radford et al., 2021) for filtering, which excludes the most noisy bottom samples, resulting in a cleaned subset used for training, following (Schuhmann et al., 2022). We used 8×RTX 4090 GPUs to generate images using text-to-image generative models.

**Prompts.** For the prompt diversification module $\psi$, we use the following system prompt to sequentially generate the prompts:

```
To generate images using a text-to-image generation model, I need to create a prompt.  Keep
the domain photorealistic and use different visual scenes and visual styles or different
color profiles/ palettes.  Here is a list of prompts that I have previously generated
<previously generated prompts>.  Please create a new prompt that does not overlap with these.
```

In Bongard-OpenWorld-P/N, we use the following prompt:

```
'positive' images:<image>...<image>
'negative' images:<image>...<image>
'query' image:<image>

Given 4 'positive' images and 4 'negative' images, where 'positive' images share 'common'
visual concepts and 'negative' images cannot, the 'common' visual concepts exclusively
depicted by the 'positive' images.  Here, 'common' sentence from 'positive' images is common
concept.  And then given 1 'query' image, please determine whether it belongs to 'positive'
or 'negative'.

Your answer is:
```

In Bongard-OpenWorld-CA, we use the following prompt:

```
'positive' images:<image>...<image>
'negative' images:<image>...<image>

Given 4 'positive' images and 4 'negative' images, where 'positive' images can be summarized
as 1 'common' sentence and 'negative' images cannot, the 'common' sentence describes a set
of concepts that are common to 'positive' images.  Please give the 'common' sentence from
'positive' images.

Your answer is:
```

The system prompt $P_S$ for HIRPG we use is as follows:

```
To generate images using a text-to-image generative model, I need to create a prompt ~~~
Here is a list of prompts that I have previously generated.  Please create a new prompt that does not overlap
with {base prompt & previously generated prompts} ~~~.
```

In Bongard-HOI-P/N, we use the following prompt:

```
'positive' images:<image>...<image>
'negative' images:<image>...<image>
'query' image:<image>

Given 2 'positive' images and 2 'negative' images, where both 'positive' and 'negative'
images share a 'common' object, and only 'positive' images share a 'common' action whereas
'negative' images have different actions compared to the 'positive' images, the 'common'
action is exclusively depicted by the 'positive' images.  And then given 1 'query' image,
please determine whether it belongs to 'positive' or 'negative' You must choose your answer
from the Choice List.  Choice list:[Positive, Negative].

Your answer is:
```

In Bongard-HOI-CA, we use the following prompt:

```
'positive' images:<image>...<image>
'negative' images:<image>...<image>

Given 2 'positive' images and 2 'negative' images, where both 'positive' and 'negative'
images share a 'common' object, and only 'positive' images share a 'common' action whereas
'negative' images have different actions compared to the 'positive' images, the 'common'
action is exclusively depicted by the 'positive' images.  Your job is to find the 'common'
action within the 'positive' images.  You must choose your answer from the Choice List.
Choice List:  [choice lists].

Your answer is:
```

**Hyperparameters.** For $\tau$, which refers to the temperature of the softmax function in CONAN, is uniformly set to 0.5 across all datasets. For $L$, the truncation ratio used in RMD score normalization, we set it to 5% for all experiments. For diverse prompt generation baselines, as well as GenOL, we generate 50 different prompts for all baselines across all benchmarks, including HIRPG, to ensure a fair comparison. Specifically, in GenOL, we set depth $D = 2$, and $K = 7$ for all setups to generate 50 prompts using HIRPG. For the optimizer and the learning rate (LR) scheduler in the CIL setup, we employed the Adam optimizer with an initial LR of 0.0003 and the Constant LR scheduler, respectively, following prior works (Koh et al., 2023; Seo et al., 2024). In the MVCIL setup, we use Adam optimizer with LR $5 \times 10^{-5}$ and Constant LR scheduler.

Following (Koh et al., 2021; 2023; Kim et al., 2024a), we conduct batch training for each incoming sample. Specifically, for PACS, CIFAR-10, DomainNet, and ImageNet, the number of batch iterations per incoming sample is set to 2, 2, 3, and 0.5, respectively, with batch sizes of 16, 16, 128, and 256. Episodic memory sizes are configured as 200, 2000, 10000, and 80000 for PACS, CIFAR-10-W, DomainNet, and ImageNet, respectively.

For MVCIL setups, the number of batch iterations per incoming sample is set to 0.5, with a batch size of 2 and a memory size of 500 in both Bongard-HOI and Bongard-OpenWorld. Unlike the CIL setup, where data is composed solely of image and label pairs, in the MVCIL setup, each set contains both negative and positive examples corresponding to a given concept. Therefore, we store 500 sets in episodic memory. In MVCIL benchmarks, *i.e.*, Bongard-HOI and Bongard-OpenWorld, we used 2 positive images and 2 negative images for a support set and 4 positive images and 4 negative images for a support set, respectively. For the MVCIL setup, we use the LLaVA-1.5-7B (Liu et al., 2023b).

We provide hyperparameter analysis of temperature $\tau$, depth $D$ and width $K$ of hierarchical tree structure, and truncate ration $L$ in Section A.23.

## A.5 Comparison of HIRPG with Diverse Prompt Generation Methods.

To evaluate the effectiveness of HIRPG in diverse prompt generation, we further qualitatively analyze the generated images based on two metrics: Recognizability (Fan et al., 2024), which evaluates whether the images accurately represent the intended concepts, and Diversity (Naeem et al., 2020), which assesses the variation between images. While our goal is to generate diverse images using varied prompts, it is important that the generated images accurately represent the desired concepts. Thus, the prompts must achieve both diversity and recognizability. For a fair comparison, we generate 50 text prompts using each prompt diversification baseline and use SDXL to generate the same number of images for all baselines, including HIRPG, as mentioned in Section A.4. We summarize the results in Table 9.

| Method | PACS | | DomainNet | |
|---|---|---|---|---|
| | Recognizability ↑ | Diversity ↑ | Recognizability ↑ | Diversity ↑ |
| LE (He et al., 2023) | 65.39 | 0.27 | 38.49 | 0.31 |
| CHB (Sarıyıldız et al., 2023) | 62.96 | 0.16 | 41.57 | 0.24 |
| SC (Tian et al., 2024a) | 71.50 | 0.19 | 33.19 | 0.20 |
| CCG (Hammoud et al., 2024) | 68.78 | 0.18 | 32.71 | 0.19 |
| HIRPG (Ours) | **90.77** | **0.31** | **52.83** | **0.35** |

Table 9: **Comparison of prompt diversification methods.** We compare the Recognizability and Diversity of images generated using text prompts derived from prompt generation methods in conjunction with a text-to-image generative model.

As shown in the table, HIRPG not only generates more diverse data, but also produces more recognizable data compared to baselines. Overall, DomainNet exhibits higher Diversity. This is because it has fewer images per class, resulting in a smaller number of generated images per prompt. For detailed descriptions of the baselines and metrics (*i.e.*, Rec and Div), see Sections A.3 and A.13, respectively.

## A.6 Selecting Hard Negative Concepts in GenOL on MVCIL Setups

In the MVCIL setup, which requires negative concepts to learn a concept, MA data and web-scraping baselines (i.e., C2C and IE) use manually selected hard negative concepts. In contrast, we leverage LLMs' hard negative sample selection capability (Moreira et al., 2024) to eliminate the need for human annotation. We provide prompts that we used for selecting hard negative concepts below.

For the Bongard-HOI benchmark, Given a (object, concept) pair, *e.g.*, (ride, a bike), GenOL retrieves hard negative concept using an LLM and the following prompt:

> To train a model that distinguishes between positive and negative images, you need to choose $N$ negative actions from the following negative action list. When choosing negative actions, you should consider the available actions from the object. For example, if the object is 'bird', possible actions are 'chase', 'feed', 'no interaction', 'watch', etc. If the object is 'orange', possible actions are 'cut', 'hold', 'no interaction', 'peel', etc. You should choose hard negative actions that are clearly distinguishable from positive actions among the possible actions.
> object: <object class>
> positive action: <positive action>
> negative action list: <action set>
> Please select a total of $N$ negative actions. The response format must be strictly result: ['negative action1', 'negative action2', ... ], and all negative actions must be included in the negative action list.

For the Bongard-OpenWorld benchmark, GenOL retrieves hard negative concept using an LLM and the following prompt:

> To create an image using a text-to-image generation model, I want to create a prompt. Below, a prompt for a positive image will be provided, and the goal is to generate a prompt for a negative image. It is important that the negative prompt partially overlaps with the positive prompt and has slight differences. For example, if the positive prompt is 'Dogs are running', then 'Dogs are drinking water' would be the negative prompt. Please create N 'negative' prompt sentences (under 5 words) that fits this description. Please ensure the response format is strictly 'prompt: answer'.
> Positive prompt: <positive prompt>.

## A.7 Additional Ablation Study of GenOL

In addition to Section 5.6, we conduct an ablation study on two components of GenOL, namely HIRPG and CONAN, using the ResNet-18 model. We use the same number of images for each baseline to ensure a fair comparison, and summarize the results in Table 10.

Similar to the ablation study with ViT-base, both components significantly enhance performance in both ID and OOD domains.

| | PACS | | | | DomainNet | | | |
| | ID | | OOD | | ID | | OOD | |
| Method | $A_{\mathrm{AUC}}$ ↑ | $A_{last}$ ↑ | $A_{\mathrm{AUC}}$ ↑ | $A_{last}$ ↑ | $A_{\mathrm{AUC}}$ ↑ | $A_{last}$ ↑ | $A_{\mathrm{AUC}}$ ↑ | $A_{last}$ ↑ |
|---|---|---|---|---|---|---|---|---|
| Vanilla GenOL | 47.74±1.52 | 47.30±2.38 | 31.66±1.45 | 25.41±0.66 | 20.82±0.39 | 17.19±0.34 | 7.09±0.21 | 5.55±0.11 |
| (+) HIRPG | 51.36±2.59 | 51.63±2.49 | 34.12±1.27 | 28.18±1.32 | 27.72±0.30 | 23.71±0.39 | 10.70±0.19 | 8.75±0.13 |
| (+) CONAN | 50.02±2.52 | 45.34±4.25 | 33.94±1.37 | 27.30±1.16 | 28.17±0.35 | 24.12±0.11 | 9.76±0.17 | 8.18±0.15 |
| (+) HIRPG & CONAN (**Ours**) | **55.89±3.06** | **55.43±2.49** | **38.53±1.15** | **33.73±1.82** | **34.60±0.31** | **29.99±0.11** | **14.53±0.22** | **12.65±0.09** |

Table 10: **Ablations for proposed components of GenOL.** Vanilla GenOL refers to generating 50 different prompts using an LLM without employing RPG or HIG, and using a single T2I generator, *i.e.*, SDXL. We use ResNet-18 as a backbone architecture.

## A.8 Ablation Study of HIRPG

We conduct an ablation study on HIRPG to investigate the benefits of each proposed component, hierarchical generation (HIG) and recurrent prompt generation (RPG), in PACS and DomainNet. For a fair comparison, we generate 50 different prompts and use SDXL to generate images for all baselines. For HIG, we use a 7-ary tree with a depth of 2, the same configuration as in HIRPG.

The results are summarized in Table 11. In the table, vanilla prompt generation refers to generating $N$ different prompts using an LLM without applying RPG or HIG. Specifically, we use the following prompts for vanilla prompt generation:

> To generate images using a text-to-image generation model, I need to create 50 prompts. Keep the domain photorealistic and use different visual scenes and visual styles or different color profiles/ palettes. Please create 50 prompts that do not overlap with each other. Please ensure that each response includes the word '[concept]'. For example, 'A photo of a [concept].', 'A detailed sketch of [concept].', 'A hyper-realistic portrait of [concept].', etc.

As shown in the table, applying RPG alone even degrades the performance compared to vanilla prompt generation. This degradation occurs because, as iterative steps progress, the length of the LLM input increases, making it challenging to utilize the information within the extended context effectively (*i.e.*, *lost-in-the-middle challenge* (Liu et al., 2023c; An et al., 2024)), as discussed in Section 4.1. In contrast, combining RPG with HIG addresses the lengthy input problem, leading to significantly improved performance in both in-distribution (ID) and out-of-distribution on PACS and DomainNet.

| Method | PACS | | | | DomainNet | | | |
|---|---|---|---|---|---|---|---|---|
| | ID | | OOD | | ID | | OOD | |
| | $A_{\text{AUC}}$ ↑ | $A_{last}$ ↑ | $A_{\text{AUC}}$ ↑ | $A_{last}$ ↑ | $A_{\text{AUC}}$ ↑ | $A_{last}$ ↑ | $A_{\text{AUC}}$ ↑ | $A_{last}$ ↑ |
| Vanila Prompt Generation | 47.74±1.52 | 47.30±2.38 | 31.66±1.45 | 25.41±0.66 | 20.82±0.39 | 17.19±0.34 | 7.09±0.21 | 5.55±0.11 |
| (+) RPG | 45.55±1.55 | 47.60±1.90 | 32.07±1.70 | 25.53±1.74 | 17.62±0.35 | 13.96±0.25 | 6.88±0.15 | 5.30±0.11 |
| (+) HIG + RPG (Ours) | **51.36±2.59** | **51.63±2.49** | **34.12±1.27** | **28.18±1.32** | **27.72±0.30** | **23.71±0.39** | **10.70±0.19** | **8.75±0.13** |

Table 11: **Benefits of components of the proposed prompt generation method**. RPG refers to the recurrent prompt generation, and HIG refers to the hierarchical generation. Vanilla prompt generation refers to generating 50 different prompts using an LLM without incorporating RPG or HIG.

## A.9 Data Ensemble Baselines

Gradient-based methods (Killamsetty et al., 2021b;a; Shin et al., 2023) minimize the distance between the gradients from the entire dataset $T$ and from the selected coreset $S$ ($S \subset T$) as follows:

$$\min_{\mathbf{w},S} \left\| \sum_{(x,y) \in T} \frac{\nabla_\theta l(x,y;\theta)}{|T|} - \sum_{(x,y) \in S} \frac{w_x \nabla_\theta l(x,y;\theta)}{\|\mathbf{w}\|_1} \right\|_2, \tag{5}$$

where $\mathbf{w}$ is the vector of learnable weights for the data in selected coreset $S$, $l$ refers to the loss function, $\theta$ denotes the model parameters, and $\| \ \|_1$, $\| \ \|_2$ represent the L1 norm and L2 norm, respectively. To solve Equation 5, **GradMatch** (Killamsetty et al., 2021a) uses orthogonal matching pursuit algorithm (Elenberg et al., 2016), while **CRAIG** (Mirzasoleiman et al., 2020) uses submodular maximization.

**LCMat (Shin et al., 2023).** While Craig and GradMatch minimize the gradient difference between $T$ and the $S$, LCMat matches the loss curvatures of the $T$ and $S$ over the model parameter space, inspired by the observation that a loss function $L$ quantifies the fitness of the model parameters $\theta$ under a specific dataset. Specifically, they claim that even though optimizing $S$ toward $T$ with respect to $\theta$ would decrease the loss difference between $T$ and $S$ in $\theta$ (*i.e.*, $|L(T;\theta) - L(S;\theta)|$), if the loss difference increases with a small perturbation $\epsilon$ in $\theta$ (*i.e.*, $|L(T;\theta+\epsilon) - L(S;\theta+\epsilon)|$), it indicates a lack of generalization on $\theta + \epsilon$, or an over-fitted reduction of $S$ by $\theta$. Since this generalization failure on the locality of $\theta$ subsequently results in the large difference of loss surfaces between $T$ and $S$, they propose an objective that maximize the robustness of $\theta$ under perturbation $\epsilon$ as follows:

$$\min_{S}(L_{abs}(T,S;\theta+\epsilon) - L_{abs}(T,S;\theta)), \tag{6}$$

where $L_{abs}$ refers to the loss difference between T and S on $\theta$ (*i.e.*, $L(T;\theta) - L(S;\theta)$).

**Moderate (Xia et al., 2023).** Moderate selects data points with scores close to the score median, using the median as a proxy for the score distribution in statistics.

Specifically, given a well-trained feature extractor $f(\cdot)$ and the full training data $T = \{t_1, t_2, \ldots, t_n\}$, the process begins by computing the hidden representations (or embeddings) of all data points in $T$, *i.e.*, $\{z_1 = f(t_1), z_2 = f(t_2), \ldots, z_n = f(t_n)\}$. Next, the $\ell_2$ distance from the hidden representation of each data point to the class prototype of its corresponding class is calculated. Formally, for a sample $t$ belonging to class $c$, its distance $d(t)$ is given by $d(t) = \|z - z^c\|_2$, where $z = f(t)$ and $z^c$ is the prototype of class $c$. Subsequently, all data points are sorted in ascending order according to their distance, which are denoted by $\{d(\tilde{t}_1), d(\tilde{t}_2), \ldots, d(\tilde{t}_n)\}$. Finally, data points near the distance median are selected as the coreset $S$.

**Uncertainty (Coleman et al., 2020).** Uncertainty suggests that data samples with a lower level of confidence in model predictions will have a greater influence on the formation of the decision boundary. For uncertainty scores, we utilize Entropy, following the approach of Shin et al. (2023), among the methods LeastConfidence, Entropy, and Margin.

**Glister (Killamsetty et al., 2021b).** Glister is a generalization-based data selection method that optimizes generalization error via a bi-level optimization problem to select the coreset $S$, aiming to maximize the log-likelihood on a held-out validation set.

## A.10 Description of Name-Only Classification Baselines.

**Glide-Syn (He et al., 2023).** This approach takes *category name* as input and employs the word-to-sentence T5 model (pre-trained on 'Colossal Clean Crawled Corpus' dataset (Raffel et al., 2020) and finetuned on 'CommonGen' dataset (Lin et al., 2019)), to generate diverse concept-specific sentences. After generating diverse sentences using the word-to-sentence T5 model, they generate corresponding images using prompts and the Glide (Nichol et al., 2021) text-to-image generative model. Finally, they introduce a clip filter to reduce noise and enhance robustness.

**CLIP-ZS (Radford et al., 2021).** CLIP-ZS refers to zero-shot classification using a pre-trained CLIP model, where the model classifies images without any additional training, leveraging its knowledge from large-scale pre-training. Since CLIP is pre-trained on large-scale web dataset, it demonstrates impressive zero-shot performance (Qian et al., 2024).

**SuS-X-SD (Udandarao et al., 2023).** This approach uses generated SuS (Support Sets) to ensure accurate predictions for target categories by taking only categories as input. Specifically, SuS-X-SD generates support sets using Stable Diffusion (Podell et al., 2023) and uses them as a combination with the pre-trained vision language model and an adapter module named TiP-X for inference.

**CuPL (Pratt et al., 2023).** While the standard zero-shot open vocabulary image classification model, *e.g.*, CLIP (Radford et al., 2021), uses only the set of base prompts, *i.e.*, 'A photo of {*category name*}', and target images for classification, CuPL proposes to use customized prompts using LLM. Specifically, they propose using GPT-3 (Brown et al., 2020), but we replace it with GPT-4o (Wu et al., 2024b), which is a stronger LLM and the one used in our proposed GenOL, for a fair comparison.

**CALIP (Guo et al., 2023).** CALIP enhances the zero-shot performance of CLIP (Radford et al., 2021) by introducing a parameter-free attention module. This module enables visual and textual representations to interact and explore cross-modal informative features via attention. As a result, image representations are enriched with textual-aware signals, and text representations are guided by visual features, leading to better adaptive zero-shot alignment.

**SD-Clf (Li et al., 2023a).** SD-Clf leverages large-scale text-to-image diffusion models, such as Stable Diffusion (Podell et al., 2023), for classification tasks. Given an input $x$ and a finite set of classes $C$, the model computes the class-conditional likelihoods $p_\theta(x|c)$. By selecting an appropriate prior distribution $p(c)$ and applying Bayes' theorem, SD-Clf predicts class probabilities $p(c|x)$, effectively classifying the input based on the computed likelihoods.

## A.11 Justification for the Use of RMD Score

Many recent works endeavor to assess the diversity (Naeem et al., 2020; Han et al., 2022; Kim et al., 2024b), complexity (Hwang et al., 2023), aesthetics (Somepalli et al., 2024; Khajehabdollahi et al., 2019), and realism (Chen et al., 2023b; 2024; 2023c) of the generated images. In our work, we use the relative Mahalanobis distance (RMD) score (Cui et al., 2023), to evaluate the complexity of the generated samples. The reason for selecting RMD is its independence from the need for real samples in its calculation, while other diversity metrics, such as the Rarity Score (Han et al., 2022) and the TopP&R (Kim et al., 2024b) require *real* samples, *i.e.*, data that have not been generated. Note that our proposed framework, GenOL, operates exclusively with *concept* inputs rather than *real* data, thus, we cannot access *real* data.

As we can see in Figure 9, the Rarity score and the RMD score exhibit similar trends, showing the ability of the RMD score to effectively measure complexity even in the absence of real samples.

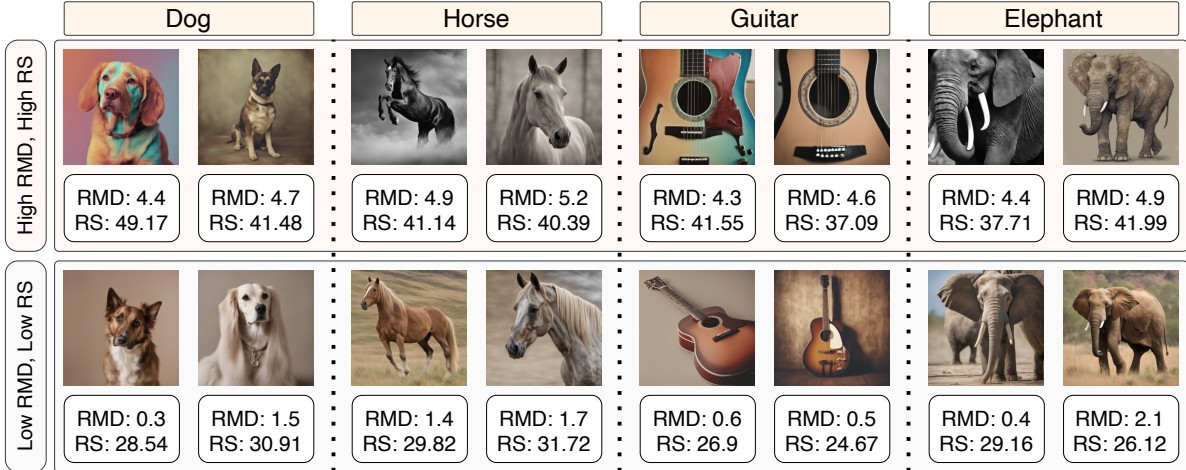

Figure 9: **Examples of samples with high RMD & high Rarity scores, as well as samples with low RMD & low Rarity scores.** The average RMD scores for Dog, Horse, Guitar, and Elephant are 2.91, 3.03, 2.43, and 3.25, respectively, while the corresponding average Rarity scores are 33.59, 33.58, 33.57, and 33.18.

### A.12   Comparison between CONAN and Various RMD-based Ensemble

We compare CONAN with various ensemble approaches that use the RMD score in different strategies on PACS and summarize the results in Table 12. The table reveals that CONAN significantly outperforms others in both ID and Out-OOD domains. Furthermore, with the exception of **CONAN**, all ensemble methods even exhibit a decrease in performance compared to the scenario where no ensembling[1] is applied. The $k$-highest RMD ensemble, which excludes easy samples, leads to insufficient learning in the class-representative region, while the $k$-lowest RMD concentrates solely on easy samples, resulting in limited diversity. Inverse CONAN employs the inverse of the probabilities utilized in CONAN. Similar to the $k$-lowest RMD ensemble, it tends to prioritize easy samples, resulting in limited diversity and accuracy loss. Equal weight ensemble refers to naively selecting outputs generated by each generator at the same ratio, without considering diversity.

| Ensemble Method Δ | ID | | OOD | |
|---|---|---|---|---|
| | $A_{\text{AUC}}$ ↑ | $A_{last}$ ↑ | $A_{\text{AUC}}$ ↑ | $A_{last}$ ↑ |
| No ensembling | 51.36±2.59 | 51.63±2.49 | 34.12±1.27 | 27.18±1.32 |
| Equal weight ensemble | 50.56±2.32 | 50.03±2.13 | 35.49±1.41 | 27.13±3.44 |
| $k$-highest RMD ensemble | 52.80±2.82 | 50.09±3.06 | 36.24±1.52 | 30.09±1.35 |
| $k$-lowest RMD ensemble | 41.72±2.88 | 37.98±2.19 | 31.17±2.34 | 24.58±2.49 |
| Inverse CONAN | 45.01±3.03 | 38.70±4.12 | 32.94±1.62 | 27.61±1.97 |
| CONAN (Ours) | **55.89±3.06** | **55.43±2.49** | **38.53±1.15** | **33.73±1.82** |

Table 12: Comparison of ensemble methods in PACS (Zhou et al., 2020), using ER (Rolnick et al., 2019) for all ensemble methods. The proposed ensemble method outperforms other ensemble methods.

### A.13   Details about Metrics

**Area Under the Curve of Accuracy ($A_{\text{AUC}}$).**   In an online CL setup, the model is trained using the stream data in real time, thus, the model needs to be used for inference at every moment rather than the predefined time point (*e.g.*, end of the task) (Koh et al., 2021; Caccia et al., 2022; Banerjee et al., 2023; Pellegrini et al., 2020). Therefore, to measure inference performance at any time, we evaluated the model at

---

[1]*No ensembling* denotes the usage of images generated exclusively through SDXL (Podell et al., 2023).

regular intervals during a specified evaluation period and then calculated the area under the accuracy curve, denoted $A_{\text{AUC}}$ (Koh et al., 2021; Caccia et al., 2022; Koh et al., 2023), which is defined as follows:

$$A_{\text{AUC}} = \sum_{i=1}^{k} f(i\Delta n)\Delta n, \tag{7}$$

where the step size $\Delta n$ is the number of samples encountered between inference queries and $f(\cdot)$ is the accuracy in the curve of the # of samples-to-accuracy plot. High $A_{\text{AUC}}$ indicates that the model maintains good inference performance throughout the entire training process.

**Recognizability.** Following Boutin et al. (2022) and Fan et al. (2024), we evaluate whether the images accurately represent the intended concepts by computing the F1 score for each class. We pre-train ViT-Base from scratch with MA data and use it to classify the images generated from each prompt diversification baseline. Recognizability is then calculated by averaging the F1 scores across all classes.

**Diversity.** To assess the diversity of generated images, Naeem et al. (2020) measures coverage, defined as the ratio of real samples encompassed by the generated samples. Specifically, they calculate the fraction of real samples whose $k$-nearest neighborhoods contain at least one generated sample. Formally, given the embedded real and generated data, represented by $\{X_i\}$ and $\{Y_j\}$ from an ImageNet pre-trained ViT-Base feature extractor, coverage is defined as:

$$\text{coverage} := \frac{1}{N} \sum_{i=1}^{N} \mathbb{1}_{\exists j \text{ s.t. } Y_j \in B(X_i, \text{NND}_k(X_i))}, \tag{8}$$

where $\text{NND}_k(X_i)$ denotes the distance from $X_i$ to the $k^{th}$ nearest neighboring among $\{X_i\}$ excluding itself and $B(x, r)$ denotes the sphere in $\mathbb{R}^D$ around $x$ with radius $r$.

**Consensus-based Image Description Evaluation (CIDEr).** In the MVCIL-CA setups, to compare model-predicted sentences with ground-truth sentences, we use CiDER (Vedantam et al., 2015), which measures the similarity between generated and ground-truth sentences while also capturing aspects such as grammaticality, saliency, importance, and both precision and recall. CIDEr (Vedantam et al., 2015) aims to automatically evaluate how well a predicted sentence, $s_p$, matches the consensus of a set of ground-truth sentences, $S = \{s_{gt,1}, \ldots, s_{gt,N}\}$. The intuition is that the measure of consensus should encode how often n-grams from the candidate sentence appear in the reference sentences. In contrast, $n$-grams that are absent from the reference sentences should not appear in the candidate sentence. To encode this intuition, they calculate the TF-IDF (Robertson, 2004) vectors for the $n$-gram elements within the candidate and reference sentences by computing the cosine similarity between the two vectors. Formally, CIDEr for $n$-grams is calculated as follows:

$$\text{CIDEr}_n(s_p, s_{gt}) = \frac{g^n(s_p) \cdot g^n(s_{gt})}{\|g^n(s_p)\|\|g^n(s_{gt})\|}, \tag{9}$$

where $g(s)$ represents the vectorized form of a sentence $s$, obtained by calculating the TF-IDF values for its $n$-gram elements. Finally, they combine the scores from $n$-grams of varying lengths as follows:

$$\text{CIDEr} = \frac{1}{N} \sum_{i=1}^{N} \text{CIDEr}_n. \tag{10}$$

Following Vedantam et al. (2015), we use $N = 4$ and define the set of ground truth sentences in the positive set as $S$.

## A.14 Details about Generators

For the set of generators $\mathcal{G}$, we use five text-to-image generative models: SDXL (Podell et al., 2023), Deep-Floyd IF[2], SD3 (Esser et al., 2024), CogView2 (Ding et al., 2022), and Auraflow[3]. As illustrated in Figure

---

[2]https://github.com/deep-floyd/IF
[3]https://huggingface.co/fal/AuraFlow

10, different generators produce varied samples when prompted with identical prompts conditioned on the same concept. Details of each generator are as follows:

**SDXL.** SDXL is an enhanced latent diffusion model for text-to-image synthesis, building upon the previous versions of Stable Diffusion. Specifically, SDXL introduces three key improvements: (1) a UNet (Ronneberger et al., 2015) backbone that is $3\times$ larger than in previous Stable Diffusion models, (2) an additional conditioning technique, and (3) a diffusion-based refinement model to further improve the visual quality of generated images.

**DeepFloyd IF.** DeepFloyd IF utilizes a frozen text encoder alongside three cascaded pixel diffusion stages. Initially, the base model produces a 64x64 image from a text prompt, which is then progressively enhanced by two super-resolution models to reach 256x256 and ultimately 1024x1024 pixels. At every stage, the model uses a frozen T5 transformer-based text encoder to derive text embeddings, which are then passed into a UNet.

**CogView2.** CogView2 pretrain a 6B-parameter transformer using a straightforward and adaptable self-supervised task, resulting in a cross-modal general language model (CogLM). This model is then fine-tuned for efficient super-resolution tasks. The hierarchical generation process is composed of three steps: (1) A batch of low-resolution images ($20 \times 20$ tokens) is first generated using the pretrained CogLM. Optionally, poor-quality samples can be filtered out based on the perplexity of CogLM image captioning, following the post-selection method introduced in CogView (Ding et al., 2021). (2) These generated images are then upscaled to $60 \times 60$-token images via a direct super-resolution module fine-tuned from the pretrained CogLM. (3) Finally, these high-resolution images are refined through another iterative super-resolution module fine-tuned from CogLM.

**SD3.** SD3 enhances current noise sampling methods for training rectified flow models (Liu et al., 2023d) by steering them toward perceptually significant scales. In addition, SD3 introduces a new transformer-based architecture for text-to-image generation, employing distinct weights for the two modalities. This design facilitates a bidirectional flow of information between image and text tokens, leading to improved typography, text comprehension, and higher human preference ratings.

**AuraFlow.** AuraFlow, inspired by SD3, is currently the largest text-to-image generation model. It introduces several modifications to SD3, including the removal of most MMDiT blocks (Esser et al., 2024) and their replacement with larger DiT encoder blocks (Peebles & Xie, 2023).

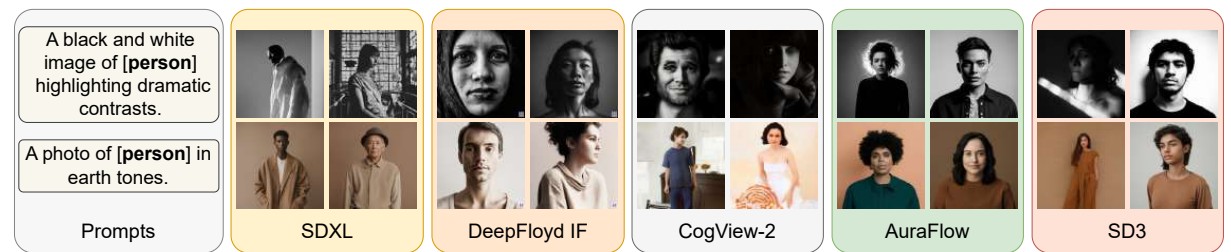

Figure 10: Examples of PACS (Zhou et al., 2020) generated samples from various generators using two of the prompt rewrites. Illustrations from the concept 'Person' are showcased.

## A.15 Extended Related Work

**Continual learning setups.** Many recent works propose realistic CL scenarios, such as blurry (Prabhu et al., 2020; Bang et al., 2021), i-blurry (Koh et al., 2021), continuous (Shanahan et al., 2021; Koh et al., 2023), and noisy (Bang et al., 2022) setups. However, they only focus on the data distribution of the stream data, rather than the acquisition of data for new concepts, for which the model needs to be learned. Recently, C2C (Prabhu et al., 2024) used web-scraped data for continual learning, to address the high cost of manual

data annotation and the difficulty in acquiring real-time data for the target concepts the model needs to learn. However, web-scraped data suffer from noise, low controllability, and usage restrictions, as highlighted in Figure 1. Please refer to Section A.30 for a more detailed explanation of the web-scraped data.

**Methods for Continual Learning.** Replay-based method, which stores data from previous tasks in episodic memory for replay, is one of the most widely used approaches, due to its effectiveness in preventing catastrophic forgetting (Zhang et al., 2023b; Yoo et al., 2024; Kozal et al., 2024). However, despite their effectiveness in preventing forgetting, they raise data privacy concerns due to the storage of real data from previous tasks in episodic memory. To address these privacy concerns, pseudo-replay approaches have been proposed (Graffieti et al., 2023; Thandiackal et al., 2021; Van de Ven et al., 2020; Shin et al., 2017; Van de Ven & Tolias, 2018), which leverage generative models to generate images of previous tasks instead of storing actual data in episodic memory. While these approaches utilize generative models similar to our GenOL framework, they still require manually annotated data to train the generative model (Shin et al., 2017; Van de Ven et al., 2020; Van de Ven & Tolias, 2018). In contrast, our GenOL framework eliminates the need for any manually annotated data, relying solely on the category names the model aims to learn.

### A.16  Quantitative Analysis on CIFAR-10-W

We compared GenOL and generative baselines combined with CONAN, for a fair comparison on CIFAR-10-W (Sun et al., 2024). As mentioned in Section A.2, since CIFAR-10-W only contains data on the OOD domains of CIFAR-10, we evaluated only the performance of the out-of-distribution (OOD) domain.

In addition, since CIFAR-10-W is a web-scraped dataset, the domain of CIFAR-10-W and the web-scraped data are the same. Therefore, we excluded web-scrapping baselines in experiments on CIFAR-10-W. We summarize the results in Table 13.

| Method | CIFAR-10-W | |
|---|---|---|
| | $A_{\mathrm{AUC}} \uparrow$ | $A_{last} \uparrow$ |
| Glide-Syn (ICLR 2023) | 47.14±0.80 | 34.13±0.54 |
| LE (ICLR 2023) | 47.20±0.67 | 34.03±0.60 |
| (+) CONAN | 51.69±0.70 | 41.32±1.38 |
| CHB (CVPR 2023) | 45.30±0.62 | 31.20±0.54 |
| (+) CONAN | 51.29±0.72 | 37.06±1.77 |
| SC (CVPR 2024) | 44.75±0.62 | 30.41±0.84 |
| (+) CONAN | 48.60±0.63 | 36.39±0.88 |
| CCG (CVPRW 2024) | 38.96±0.94 | 24.71±0.70 |
| (+) CONAN | 41.32±0.99 | 28.94±0.77 |
| HIRPG | 52.52±0.37 | 41.04±1.26 |
| (+) CONAN (**Ours**) | **55.53±0.41** | **43.51±1.13** |

Table 13: **Quantitative comparison with diverse prompt generation baselines on CIFAR-10-W.**

### A.17  Pseudocode for the GenOL

Algorithm 1 provides a detailed pseudocode for GenOL. When a new concept is encountered, the prompt generation module $\psi$ generates concept-specific prompts. These prompts are then used by a set of T2I generators $G$ to create concept-specific images. Subsequently, the ensembler $\Delta$ selects a coreset from these generated images for efficiency, instead of training on the entire dataset. The continual learner is then trained using this selected ensemble set. During training, GenOL also stores a small portion of previously generated

---

**Algorithm 1** GenOL

---

1: **Input** Model $f_\theta$, Prompt generation module $\psi$, Set of Generators $\mathcal{G}$, Ensembler $\Delta$, Concept stream $\mathcal{C}$, Learning rate $\mu$, Episodic memory $\mathcal{M}$
2: **for** $y \in \mathcal{Y}$ **do**            ▷ New concept arrives from concept stream $\mathcal{Y}$
3:     **Generate** $\mathcal{P}_c \leftarrow \psi(c)$       ▷ Generate prompt set $\mathcal{P}_c$ for a given concept $c$ using $\psi$
4:     **Generate** $\{\mathcal{X}_c^{(i)}\}_{i=1}^{|G|} \leftarrow \mathcal{G}(\mathcal{P}_c)$       ▷ Generate image set $\mathcal{X}_c$ using $\mathcal{G}$ and $\mathcal{P}_c$
5:     $(\mathcal{X}_c, c) \leftarrow \Delta(\{\mathcal{X}_c^{(i)}\}_{i=1}^{|G|})$       ▷ Ensemble generated image set using ensembler $\Delta$
6:     $\mathcal{L} = \mathcal{L}_{\text{CE}}(f_\theta(\mathcal{X}_c), c)$       ▷ Calculate cross entropy loss
7:     **Update** $\theta \leftarrow \theta - \mu \cdot \nabla_\theta \mathcal{L}$       ▷ Update model
8:     **Update** $\mathcal{M} \leftarrow \text{ReservoirSampler}(\mathcal{M}, (X_c, c))$       ▷ Update episodic memory
9: **end for**
10: **Output** $f_\theta$

---

**Algorithm 2** RPG

---

1: **Input** Maximum number of leaf nodes of $K$-ary Tree $K$, System prompt $P_s$, Large language model LLM, Prompt of parent node $P_{d,k}$
2: $\mathcal{P} \leftarrow \emptyset$       ▷ Initialize the generated prompt set $\mathcal{P}$
3: $k' \leftarrow 1$       ▷ Initialize the number of child node of $P_{d,k}$
4: **while** $k' \leq K$ **do**       ▷ Generate $k'_{th}$ child node of $P_{d,k}$
5:     **if** $k' = 1$ **then**
6:         $P_{d+1,k'} \leftarrow \text{LLM}(P_s, P_{d,k})$
7:     **else**
8:         $P_{d+1,k'} \leftarrow \text{LLM}(P_s, P_{d,k} \cup \mathcal{P})$       ▷ Recurrently forward the previously generated prompts $\mathcal{P}$
9:     **end if**
10:     $\mathcal{P} \leftarrow \mathcal{P} \cup \{P_{d+1,k'}\}$       ▷ Add the currently generated prompts to $\mathcal{P}$
11:     $k' \leftarrow k' + 1$
12: **end while**
13: **Output** $\mathcal{P}$

---

**Algorithm 3** HIRPG

---

1: **Input** Newly encountered concept $y$, Maximum number of leaf nodes of $K$-ary Tree $K$, Prompt of parent node $P_{d,k}$
2: $\mathcal{P} \leftarrow \text{RPG}(P_{d,k})$       ▷ Generate $K$ number of prompts using RPG
3: $\mathcal{P}_{ch} \leftarrow \emptyset$       ▷ Initialize the prompt set generated from the child nodes
4: **if** $d < D$ **then**
5:     **for** $P_{k'} \in \mathcal{P}$ **do**
6:         $\mathcal{P}_{ch} \leftarrow \mathcal{P}_{ch} \cup \text{HIRPG}(P_{k'}, d+1)$       ▷ Merge prompt generated in child noes
7:     **end for**
8: **end if**
9: **Output** $\mathcal{P} \cup \mathcal{P}_{ch}$       ▷ Merge the prompts generated in the child nodes and the current node, then return

---

samples in episodic memory $M$. Although GenOL can generate images related to previously encountered concepts, retaining these samples helps to reduce computational overhead.

Additionally, Algorithm 3, Algorithm 4, and Algorithm 5 provide a detailed pseudo code for prompt generation module $\psi$, a set of generators $G$, ensembler $\Delta$, respectively, which are components of GenOL.

### A.18   Details about Web-Scrapping

For web-scrapping, we follow C2C (Prabhu et al., 2024), which proposes scraping data from the web using category names. C2C (Prabhu et al., 2024) uses four search engines, including Flickr, Google, Bing, and

---

**Algorithm 4** Set of Generators $\mathcal{G}$

---

1: **Input** Rewritten prompt set $\mathbf{P}$, Generative models $\mathcal{G} = \{g_1, g_2, ..., g_{|\mathcal{G}|}\}$
2: $U_1, ..., U_{|\mathcal{G}|} \leftarrow \emptyset, ..., \emptyset$       ▷ Initialize the sets of generated images for each model
3: **for** $p \in \mathbf{P}$ **do**
4:    **for** $g_i \in \mathcal{G}$ **do**
5:      **Generate** $x_y^{(i)} \leftarrow g_i(p)$     ▷ Generate image $x_y^{(i)}$ using prompt $p$ and generative model $g_i$
6:      $U_i \leftarrow U_i \cup \{x_y^{(i)}\}$          ▷ Append $x_y^{(i)}$ to the set $U_i$
7:    **end for**
8: **end for**
9: **Output** $\{U_1, U_2, ..., U_{|\mathcal{G}|}\}$        ▷ Return the generated image sets for each model

---

**Algorithm 5** Ensembler $\Delta$

---

1: **Input** Generated image sets $\{U_1, U_2, ..., U_{|\mathcal{G}|}\}$, Coreset size $|V|$, Temperature parameter $\tau$
2: $U \leftarrow \bigcup_{i=1}^{|\mathcal{G}|} U_i$        ▷ Combine all generated image sets into a single set $U$
3: **for** each sample $(x_i, y_i) \in U$ **do**
4:    **Compute** $\mathcal{RMD}(x_i, y_i) \leftarrow \mathcal{M}(x_i, y_i) - \mathcal{M}_{\text{agn}}(x_i)$    ▷ Compute RMD scores for each sample
5: **end for**
6: **Truncate** $\mathcal{RMD}(x_i, y_i)$          ▷ Remove outliers from RMD scores
7: **Normalize** $\overline{\mathcal{RMD}}(x_i, y_i) \leftarrow \mathcal{RMD}(x_i, y_i)$      ▷ Apply Z-score normalization
8: **Compute** $p_{x|y} = \dfrac{e^{\overline{\mathcal{RMD}}_{x|y}/\tau}}{\sum_{x' \in \mathcal{U}} e^{\overline{\mathcal{RMD}}_{x'|y}/\tau}}$ for each $x \in U$    ▷ Compute the selection probability using softmax function
9: **Select** $\mathbf{V} \leftarrow$ Sample $|V|$ images from $U$ based on probabilities $p_{x|y}$
10: **Output** Coreset $\mathbf{V}$

---

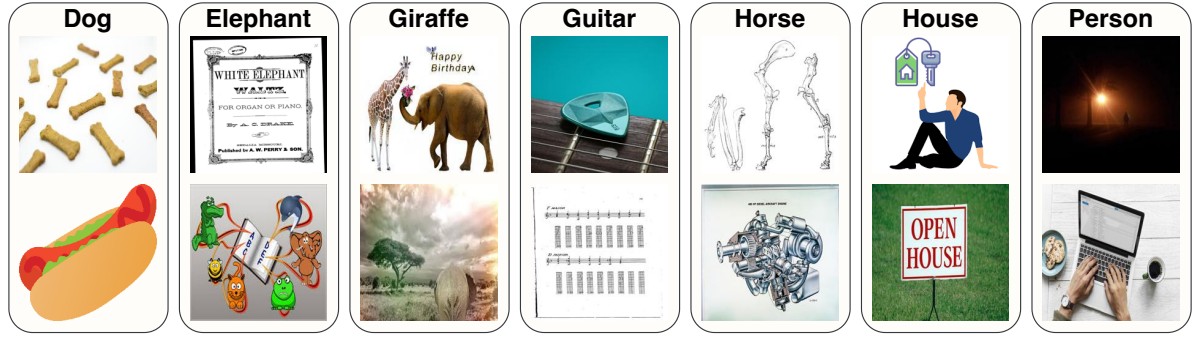

Figure 11: Examples of noisy raw data obtained via web-scraping for the classes in the PACS dataset.

DuckDuckGo, using the publicly available querying tool[4] to collect URLs. While C2C uses four search engines for scraping, we only use three search engines, *i.e.*, Flickr, Google, and Bing, since ICrawler did not support web data scraping from DuckDuckGo at the time of our attempt on February 20, 2024. After collecting the URLs from each search engine, we use a multi-threaded downloader[5] to quickly download the images, following (Prabhu et al., 2024). For Flickr, we are able to download approximately 500 images per minute due to the rapid URL collection facilitated by the official API. Meanwhile, for Google and Bing, the download rate is slower, at approximately 100 images per minute on a single CPU machine. However, the download rate depends on the network conditions and the status of the API and the search engine. In C2C, the ensemble of web-scrapped data from search engines is weighted differently for each benchmark. For example, in the Stanford Cars benchmark, the weights are Google: Bing: Flickr = 5:4:2, while in the Flowers benchmark, they are 1:1:2, respectively. Since we use different benchmarks compared to C2C, we

---

[4]https://github.com/hellock/icrawler
[5]https://github.com/rom1504/img2dataset

select equal weight selection to ensemble web-scrapped data, *i.e.*, Google: Bing: Flickr = 1:1:1, which is one of the most straightforward and widely used ensemble techniques (Shahhosseini et al., 2022; Ju et al., 2018).

Datasets scraped from search engines such as Flickr, Google, and Bing contain uncurated (*i.e.*, noisy) samples. To clean these datasets, following (Schuhmann et al., 2022), we use a pre-trained CLIP (Radford et al., 2021) model to measure the similarity between the images and corresponding promts. Specifically, we scraped 10% more data than the required dataset size (*i.e.*, the number of manually annotated data) and removed samples with a low CLIP similarity score for each experiment. Although prior work (Prabhu et al., 2024) addressed the ambiguity of queries through manual query design, such as adding an auxiliary suffix to refine queries, in an online CL scenario, where new concepts stream in real-time, such hand-crafted query designs for each concept are limited.

In summary, data noise, network dependency, and the need for manual query design specific to each concept restrict the use of web-scrapped data in real-world scenarios where new concepts are encountered in real-time.

### A.19 Comparison of CONAN with Hard Negative Sampling Methods

In addition to comparing CONAN with coreset selection methods, we also evaluate it against hard negative sampling methods, which prioritize hard samples based on their own selection criteria, including HCL (Robinson et al., 2021) and H-SCL (Jiang et al., 2024). We summarize the results in Table 14.

As shown in Table 14, CONAN outperforms hard negative sampling baselines in both PACS and DomainNet. We believe that the lower performance of hard negative sampling methods stems from the lack of class-representative samples, which are crucial in coreset, similar to the lower performance with $k$-highest RMD selection strategy in Section A.12.

| Method | PACS | | | | DomainNet | | | |
| | ID | | OOD | | ID | | OOD | |
| | $A_{\text{AUC}} \uparrow$ | $A_{last} \uparrow$ | $A_{\text{AUC}} \uparrow$ | $A_{last} \uparrow$ | $A_{\text{AUC}} \uparrow$ | $A_{last} \uparrow$ | $A_{\text{AUC}} \uparrow$ | $A_{last} \uparrow$ |
|---|---|---|---|---|---|---|---|---|
| HCL | 51.18±2.76 | 46.10±3.76 | 36.52±1.66 | 29.25±2.27 | 30.34±0.78 | 24.76±0.82 | 13.26±0.56 | 11.56±0.43 |
| H-SCL | 51.55±1.69 | 47.41±2.91 | 35.99±1.20 | 27.57±1.70 | 32.22±0.34 | 26.47±0.25 | 13.88±0.13 | 11.36±0.20 |
| CONAN (**Ours**) | **55.89±3.06** | **55.43±2.49** | **38.53±1.15** | **33.73±1.82** | **34.60±0.31** | **30.09±0.11** | **14.53±0.22** | **12.65±0.09** |

Table 14: **Quantitative comparison between hard negative sampling baselines on CIL setup.** CONAN outperforms hard negative sampling baselines in both ID and OOD domains.

### A.20 Additional Analysis of Bias

We summarize the attribution keywords used for analysis of bias in Tab. 15.

**Effect of Bias**  To analyze the impact of biased data in continual learning, we continually fine-tune a CLIP model on the PACS dataset acquired by baselines and GenOL, which includes the *person* class. We then evaluate whether the predictions for the *person* class exhibit biases toward specific genders. Specifically, we fine-tune a pre-trained CLIP model on the person class using the prompt '*A photo of a person*' and person images generated by each method. If the training data acquired by a method is biased toward a specific gender, the model will likely make biased predictions for that gender in the test set. During evaluation, we measure accuracy, which evaluates whether the model correctly predicts the ground truth gender of the test data, and we summarize the results in Table 16. As shown in the table, CLIP models trained on data generated by CHB and

| Method | Accuracy |
|---|---|
| C2C | 71.26 |
| CHB | 64.37 |
| CCG | 64.72 |
| SC | 72.16 |
| LE | 74.71 |
| GenOL (Ours) | **77.01** |

Table 16: **Accuracy of the fine-tuned CLIP model on the 'person' class in the PACS dataset.**

CCG, which exhibit significant biases toward specific genders in Figure 5, tend to reflect these biases in their predictions. In contrast, CLIP models trained using data generated by GenOL correctly predict gender with higher accuracy (*i.e.*, less bias), highlighting the unbiased and diverse data generated by GenOL.

| Bias Type | Category | Attribution Keywords |
|---|---|---|
| Gender | Female | ['she', 'her', 'hers', 'woman', 'female', 'girl'] |
| | Male | ['he', 'him', 'his', 'man', 'male', 'boy'] |
| Race | Black | ['Black person', 'Black man', 'Black woman', 'Black boy', 'Black girl'] |
| | White | ['White person', 'White man', 'White woman', 'White boy', 'White girl'] |
| | Asian | ['Asian person', 'Asian man', 'Asian woman', 'Asian boy', 'Asian girl'] |

Table 15: **Attention keywords for Gender/Race bias.** We categorize gender bias into female and male and race bias into Black, White, and Asian, respectively.

**Geographical Bias** In addition to gender/race bias, we measure the geographical bias of objects, evaluating whether the generated content reflects artifacts and surroundings from across the globe, rather than disproportionately representing certain regions, as proposed by Basu et al. (2023). Specifically, similar to the method for measuring gender and race bias, we calculate the similarity between the text embedding of *'a high-definition image of a typical {concept} in {nation}'* and the image embeddings for all concepts in the dataset. We summarize the results in Figure 12. As we can see in the figure, data generated by GenOL does not produce data biased toward specific regions but instead generates data from various regions for the same concept, even when compared to MA.

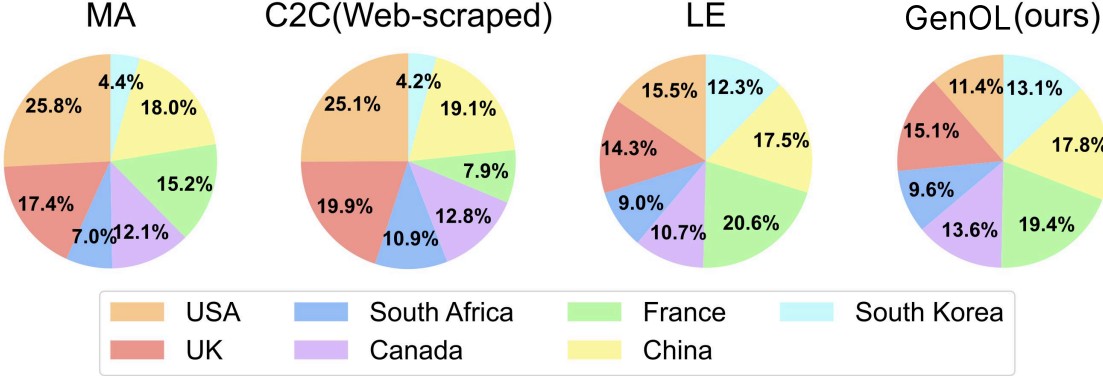

Figure 12: **Comparison of geographical bias between GenOL, web-scraped data (*i.e.*, C2C), and manually annotated (MA) data.** Compared to C2C and MA, GenOL and LE generate data that is more evenly distributed across various geographical regions.

### A.21 Quantitative Analysis on Fine-grained Benchmarks

We evaluate GenOL on fine-grained benchmarks. Specifically, as our focus extends beyond the accuracy of the ID domain to include that of the OOD domain, we conduct experiments on Birds-31 (Yu et al., 2024), a fine-grained domain generalization benchmark comprising CUB-200-2011 (He & Peng, 2019), NABirds (Horn et al., 2015), and iNaturalist2017 (Van Horn et al., 2018). We summarize the results in Table 17.

As shown in the table, GenOL outperforms the baselines and achieves performance comparable to MA. We attribute this success to GenOL's ability to effectively generate diverse images of fine-grained species through our proposed prompt diversification method (*i.e.*, HIRPG) and data ensembling method (*i.e.*, CONAN), as shown in Table 9 and Table 4.

| Method | ID | | OOD | |
|---|---|---|---|---|
| | $A_{\mathrm{AUC}}$ ↑ | $A_{last}$ ↑ | $A_{\mathrm{AUC}}$ ↑ | $A_{last}$ ↑ |
| MA | **26.19±1.12** | 16.09±1.97 | **21.29±0.76** | 12.98±0.74 |
| GenOL (Ours) | 24.25±0.41 | **20.06±0.93** | 18.62±0.33 | **13.49±1.01** |

Table 17: **Qualitative comparison of baselines on Birds-31.** GenOL shows performance comparable to manually annotated (MA) data.

### A.22 Comparison of concept-specific prompts and concept-agnostic prompts

We can categorize prompt diversification strategies into two categories: concept-agnostic diversification and concept-aware diversification. Concept-agnostic diversification strategy first generates a set of diverse prompts, *e.g.*, {A vibrant photo of {concept} during sunrise with a warm color palette, A cinematic wide-angle shot of {concept} at dusk, ...}, and then inserts a given concept into the concept placeholder. As a result, prompts for all concepts follow the same sample templates. In contrast, concept-aware prompts generate unique prompts for each concept, which can differ in aspects like background and color schemes.

LE, SC, and CCG generate concept-aware prompts, which cause prompt generation time to scale with the total number of classes, as revising prompts using LLMs, such as GPT-3.5, can enhance their naturalness by incorporating suitable backgrounds and styles for each class. In contrast, GenOL employs the same template across all classes and benchmarks, inspired by the renowned psychologist Dr. K. Anders Ericsson, who argued that *high-end performance of human results from extensive practice beyond one's comfort zone* Ericsson et al. (1993); Huang et al. (2022). In other words, we believe that samples generated from such unnatural prompts can facilitate the learning of concepts from more diverse viewpoints, thereby enhancing generalization ability. To this end, by applying fixed templates across all concepts rather than generating concept-specific prompts, we can significantly reduce computational costs—*i.e.*, prompt generation time becomes independent of the total number of concepts—while also providing opportunities to train with more challenging samples.

### A.23 Effect of Hyperparameters

Hyperparameters, including temperature $\tau$, truncate ratio $L$, depth $D$ and width $K$ of $K$-ary tree, are selected through a hyperparameter search on DomainNet and are consistently applied to the other benchmarks.

**Effect of Depth $D$ and Width $K$** Our proposed prompt diversification method (*i.e.*, HIRPG) utilizes a hierarchical tree structure. Specifically, we construct a complete $K$-ary tree with a depth of $D$, allowing us to generate $\frac{K^{d+1}-1}{K-1}$ diverse prompts. To generate a desired number of diverse prompts, we can adjust the parameters by either increasing $K$ and decreasing $D$, or decreasing $K$ and increasing $D$. Below, we demonstrate the effects of the tree's depth ($D$) and width ($K$), and explain how we selected the hyperparameters used in HIRPG.

In Figure 13 and Figure 14, the case of $D = 1$ demonstrates low diversity and recognizability. This occurs because significantly increasing $K$ and

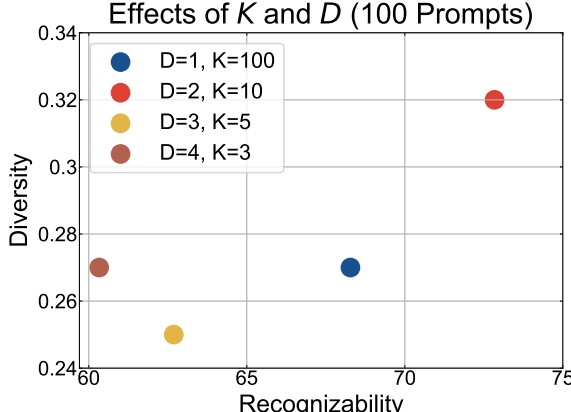

Figure 13: **Effect of $K$ and $D$ of K-ary tree in HIRPG.** We generate 100 different prompts using various combinations of $K$ and $D$ of the tree. We then use these prompts to generate images and measure the Recognizability and Diversity of the generated images.

decreasing $D$ can lead to difficulties in fully utilizing information within the long context, a challenge referred to as the 'lost-in-the-middle' problem (Liu et al., 2023c; An et al., 2024). Specifically, to generate $N$ distinct prompts, we iteratively generate prompts at each RPG step. In the final step, $N − 1$ previously generated prompts are used as negative examples. Providing such a long context to the LLM can hinder its ability to fully comprehend and effectively reference the negative examples (*i.e.*, previously generated prompts). As a result, the LLM may produce unrecognizable prompts or generate prompts that duplicate previously created ones.

The cases of $D = 3$ and $D = 4$, which correspond to low $K$ values, exhibit low recognizability. This is because providing only a few negative examples offers insufficient context. Specifically, recent studies (Agarwal et al., 2024) have observed significant performance improvements across various tasks when using many-shot examples, compared to few-shot examples, in in-context learning.

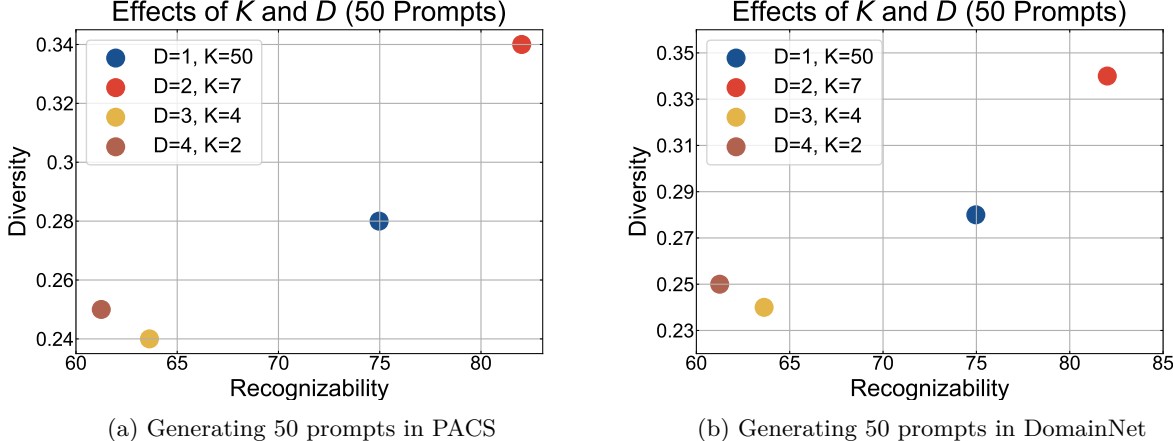

(a) Generating 50 prompts in PACS      (b) Generating 50 prompts in DomainNet

Figure 14: **Effect of $K$ and $D$ of K-ary tree in HIRPG.** To generate 50 different prompts on PACS and DomainNet, we generate prompts using various combinations of $K$ and $D$ of the tree. We then use these prompts to generate images and measure the Recognizability and Diversity of the generated images.

By balancing the trade-offs of both scenarios, we recommend selecting appropriate values for $K$ that avoid being excessively high or low, along with the corresponding $D$. In our experiments, we use $D = 4$ and $K = 7$ to generate 50 distinct prompts.

**Effect of Temperature $\tau$.** A lower temperature causes the ensemble method to focus more on samples with high RMD scores (*i.e.*, difficult samples). However, setting the temperature too low can hinder performance by excluding low RMD samples (*i.e.*, easy samples). Conversely, a higher temperature increases the likelihood of including samples with low RMD scores (*i.e.*, easy samples), but setting it too high results in an ensemble containing too many easy samples. Therefore, we select an appropriate temperature via a hyperparameter search. The results of this search are presented in Figure 15. While both excessively high and low temperatures lead to diminished performance, there is a wide range of temperatures that maintain stable performance.

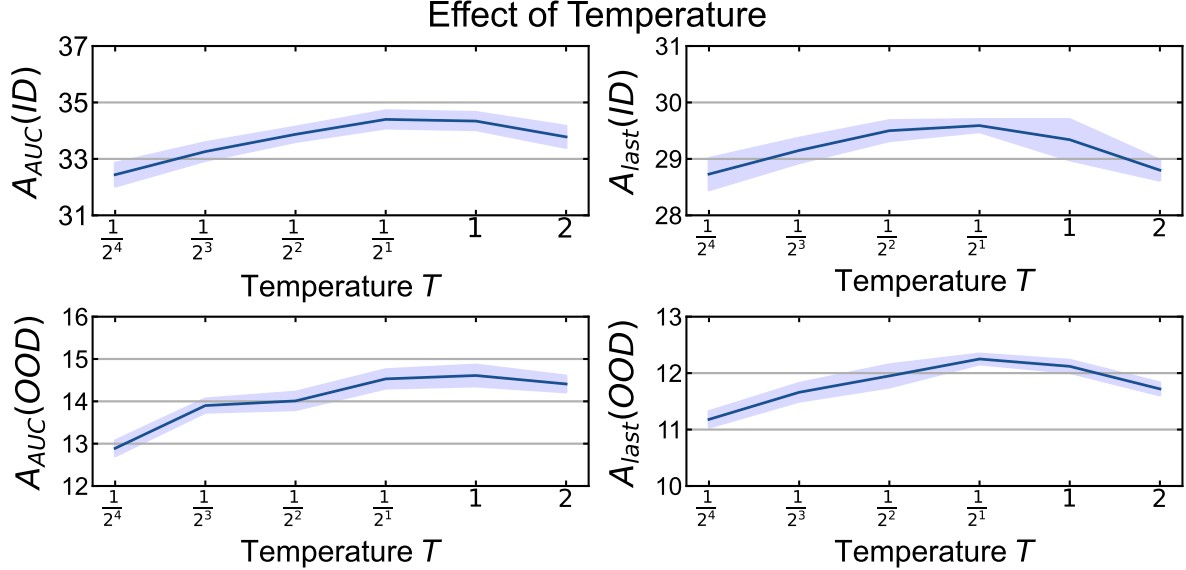

Figure 15: **Effect of temperature $T$.** We measure $A_{\mathrm{AUC}}$ and $A_{last}$ on both ID and OOD domains in DomainNet.

**Effect of Truncate Ratio** $L$. In CONAN, we truncate the samples with RMD scores in the upper and lower $L\%$ to minimize the impact of outliers on the probability distribution. Truncating a very small portion of the candidate set may cause the ensemble set to include outliers and generated images with artifacts. In contrast, truncating a large portion of the candidate set can discard difficult samples that are not outliers, as well as easy samples (*i.e.*, concept-representative samples) that are crucial for coreset construction (Bang et al., 2021). Therefore, as shown in Figure 16, both very high and low truncation ratios cause performance degradation. To this end, we select an appropriate truncation ratio by balancing the advantages and drawbacks of high and low truncate ratios.

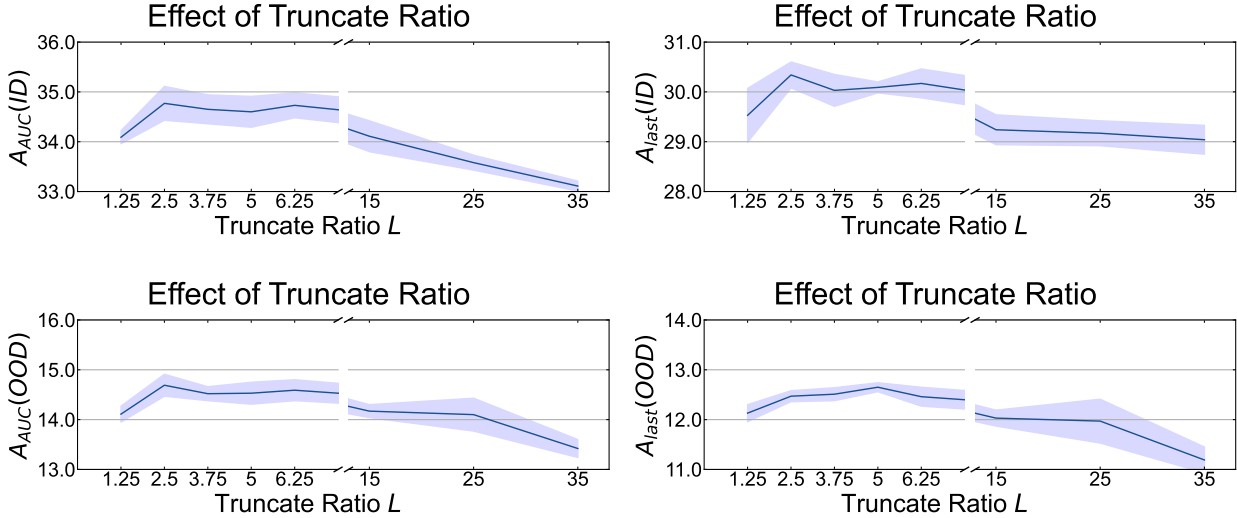

Figure 16: **Effect of Truncate Ratio** $L$. We adjust the truncation ratio over a wide range in DomainNet and measure $A_{AUC}$ and $A_{last}$ on both in-distribution and out-of-distribution domains.

### A.24 Quantitative comparison of name-only baselines on recent prompt-based CL methods.

In all data acquisition baselines, including our proposed GenOL, we consistently employ ER as the CL method, a widely adopted and strong CL strategy (Seo et al., 2025), to ensure a fair and consistent comparison. Beyond this, we also compare GenOL with other data acquisition baselines on recent prompt-based and parameter-efficient CL methods (i.e., MISA (Kang et al., 2025) and C-Prompt (Gao et al., 2024)), validating its applicability regardless of the underlying CL algorithm. We summarize the results for MISA and C-Prompt in Tab. 18 and Tab. 19, respectively. As shown, continual learning on data acquired by GenOL consistently outperforms baselines, highlighting that GenOL can be effectively combined with future state-of-the-art continual learning algorithms.

### A.25 Quantitative analysis using Qwen and LLaMA as LLMs

The LLMs used in both the generative baselines and our proposed GenOL can be readily replaced with different LLM architectures. To verify whether GenOL consistently outperforms the baselines across various LLMs, we compared its performance when using Qwen and LLaMA as the underlying LLMs, in addition to the GPT-4o results reported in Tab. 2. Specifically, we employ LLaMA-3-8B and Qwen-3-8B and summarize the results in Tab. 20 and Tab. 21, respectively. As shown, GenOL consistently surpasses the baselines regardless of the underlying LLM, although the absolute performance varies across models. These findings

| Method | PACS | | | | DomainNet | | | |
|---|---|---|---|---|---|---|---|---|
| | ID | | OOD | | ID | | OOD | |
| | $A_{\mathrm{AUC}}$ ↑ | $A_{last}$ ↑ | $A_{\mathrm{AUC}}$ ↑ | $A_{last}$ ↑ | $A_{\mathrm{AUC}}$ ↑ | $A_{last}$ ↑ | $A_{\mathrm{AUC}}$ ↑ | $A_{last}$ ↑ |
| LE | 72.08±2.32 | 54.70±2.31 | 44.73±1.75 | 32.79±1.68 | 29.27±1.54 | 22.51±2.37 | 9.90±1.07 | 7.42±2.14 |
| (+) CONAN | 75.76±1.82 | 57.66±1.93 | 51.88±2.25 | 36.13±0.92 | 37.13±2.12 | 32.14±0.97 | 13.22±2.03 | 11.37±1.26 |
| CHB | 70.02±2.28 | 62.02±2.28 | 35.74±1.76 | 22.41±1.54 | 41.02±2.34 | 30.19±2.97 | 4.60±1.21 | 11.74±2.28 |
| (+) CONAN | 67.48±1.10 | 48.86±2.02 | 43.53±0.52 | 27.07±1.83 | 52.42±2.12 | 46.62±1.44 | 19.39±1.08 | 16.29±1.68 |
| SC | 64.43±1.96 | 53.51±1.38 | 34.01±1.64 | 21.28±2.09 | 46.49±1.02 | 38.42±1.28 | 16.92±2.16 | 12.90±1.41 |
| (+) CONAN | 66.68±1.65 | 50.35±2.56 | 38.50±2.04 | 27.04±1.90 | 52.23±2.65 | 44.28±1.91 | 17.80±2.06 | 17.66±1.83 |
| SC | | | | | | | | |
| CCG | 59.85±2.39 | 52.72±1.73 | 33.97±1.68 | 23.00±2.33 | 30.17±2.74 | 23.86±1.52 | 10.52±1.67 | 7.55±1.65 |
| (+) CONAN | 63.27±2.16 | 53.02±1.26 | 40.81±1.98 | 32.42±1.88 | 45.17±1.21 | 38.72±2.13 | 16.28±1.37 | 13.33±2.15 |
| HIRPG | 76.75±2.62 | 71.41±2.33 | 44.71±2.17 | 36.40±2.36 | 51.70±2.32 | 44.60±2.29 | 21.85±2.05 | 18.12±0.95 |
| (+) CONAN (**Ours**) | **79.89±2.62** | **69.14±1.85** | **54.99±0.67** | **39.82±1.50** | **53.48±2.05** | **46.17±2.33** | **24.56±0.97** | **20.05±0.98** |
| MA | 87.92±1.85 | 81.80±1.74 | 40.30±1.69 | 28.96±2.02 | 73.78±2.03 | 68.74±1.05 | 25.55±0.95 | 22.85±1.88 |

Table 18: **Quantitative comparison of name-only baselines using MISA for continual learning.** MA refers to training a model with manually annotated data. The combination of HIRPG and CONAN refers to our proposed GenOL. We bolded the highest accuracy, excluding the MA data (*i.e.*, ideal scenario).

| Method | PACS | | | | DomainNet | | | |
|---|---|---|---|---|---|---|---|---|
| | ID | | OOD | | ID | | OOD | |
| | $A_{\mathrm{AUC}}$ ↑ | $A_{last}$ ↑ | $A_{\mathrm{AUC}}$ ↑ | $A_{last}$ ↑ | $A_{\mathrm{AUC}}$ ↑ | $A_{last}$ ↑ | $A_{\mathrm{AUC}}$ ↑ | $A_{last}$ ↑ |
| LE | 73.80±1.98 | 69.49±1.04 | 43.75±1.40 | 36.08±1.94 | 33.74±2.41 | 23.59±1.72 | 10.35±1.92 | 8.30±2.12 |
| (+) CONAN | 76.03±1.41 | 70.87±2.10 | 53.20±2.12 | 44.13±1.81 | 41.44±1.93 | 30.88±1.72 | 13.60±1.52 | 10.11±2.09 |
| CHB | 71.63±1.74 | 65.92±1.25 | 34.33±1.89 | 28.62±1.86 | 28.84±2.12 | 23.08±1.97 | 9.36±2.28 | 5.55±1.84 |
| (+) CONAN | 68.60±2.17 | 53.40±1.79 | 43.60±2.02 | 37.81±1.88 | **42.35±2.25** | 28.02±1.72 | 12.45±2.58 | 9.22±2.10 |
| SC | 59.02±1.95 | 51.47±1.19 | 28.19±1.15 | 20.14±1.20 | 21.43±2.23 | 10.11±1.27 | 6.89±1.95 | 3.45±1.99 |
| (+) CONAN | 67.81±2.46 | 57.10±1.24 | 38.26±1.76 | 25.62±2.44 | 29.48±1.47 | 27.46±2.67 | 9.84±1.35 | 6.87±2.03 |
| CCG | 61.73±2.18 | 46.22±2.57 | 29.09±1.76 | 27.54±1.98 | 19.27±1.10 | 13.29±1.44 | 5.77/±2.04 | 4.43±2.19 |
| (+) CONAN | 61.98±2.22 | 50.09±1.09 | 43.29±2.19 | 34.67±1.53 | 22.80±1.24 | 12.75±1.55 | 7.96±2.31 | 6.66±2.12 |
| HIRPG | **78.88±1.75** | 64.73±1.68 | 44.25±2.01 | 34.91±1.86 | 37.80±2.24 | 25.10±1.84 | 16.12±1.63 | 11.31±1.98 |
| (+) CONAN (**Ours**) | 77.62±1.53 | **72.96±1.11** | **54.24±1.59** | **44.38±1.03** | 41.17±2.42 | **31.41±2.59** | **18.19±1.65** | **15.11±1.84** |
| MA | 88.40±1.85 | 85.99±2.55 | 39.86±2.76 | 30.46±2.55 | 61.59±1.89 | 50.83±2.20 | 20.47±1.30 | 15.02±1.37 |

Table 19: **Quantitative comparison of name-only baselines using C-Prompt for continual learning.** MA refers to training a model with manually annotated data. The combination of HIRPG and CONAN refers to our proposed GenOL. We bolded the highest accuracy, excluding the MA data (*i.e.*, ideal scenario).

| Method | PACS | | | | DomainNet | | | |
|---|---|---|---|---|---|---|---|---|
| | ID | | OOD | | ID | | OOD | |
| | $A_{\mathrm{AUC}}$ ↑ | $A_{last}$ ↑ | $A_{\mathrm{AUC}}$ ↑ | $A_{last}$ ↑ | $A_{\mathrm{AUC}}$ ↑ | $A_{last}$ ↑ | $A_{\mathrm{AUC}}$ ↑ | $A_{last}$ ↑ |
| CHB | 48.02±1.80 | 45.64±3.19 | 29.39±1.40 | 20.65±0.55 | 14.71±1.82 | 12.13±1.08 | 2.59±1.13 | 2.35±1.76 |
| (+) CONAN | 49.35±1.95 | 47.12±2.22 | 30.67±1.16 | 23.06±1.25 | 28.41±0.69 | 23.81±0.12 | 8.96±0.19 | 7.01±0.18 |
| SC | 48.44±2.50 | 45.54±2.37 | 30.55±1.35 | 23.37±0.44 | 16.88±0.28 | 12.76±0.26 | 5.05±0.18 | 3.70±0.10 |
| (+) CONAN | 47.49±1.74 | 46.25±2.94 | 31.30±1.53 | 26.85±1.38 | 22.43±2.08 | 22.21±1.59 | 9.17±2.23 | 8.61±1.72 |
| CCG | 45.55±2.76 | 43.46±1.76 | 31.06±1.60 | 24.08±0.72 | 16.40±0.20 | 13.03±0.29 | 4.89±0.12 | 3.59±0.03 |
| (+) CONAN | 48.71±2.27 | 47.20±2.13 | 32.41±1.46 | 25.08±0.60 | 17.98±1.99 | 16.69±2.10 | 8.05±1.14 | 5.96±2.90 |
| HIRPG | 47.36±2.62 | 44.31±2.90 | 33.62±1.52 | 26.23±3.15 | 18.86±2.14 | 13.90±2.08 | 6.81±0.94 | 5.54±2.00 |
| (+) CONAN (**Ours**) | **49.71±2.40** | **49.14±1.78** | **35.56±1.50** | **31.68±1.91** | **29.53±0.23** | **24.58±0.26** | **15.79±0.27** | **13.22±0.03** |
| MA | 67.10±4.07 | 61.95±0.92 | 27.75±1.44 | 20.90±0.95 | 51.13±0.28 | 42.95±0.15 | 13.48±0.09 | 10.69±0.07 |

Table 20: **Quantitative comparison of name-only baselines using LLaMA3-8B as the LLM.** MA refers to training a model with manually annotated data. The combination of HIRPG and CONAN refers to our proposed GenOL. We bolded the highest accuracy, excluding the MA data (*i.e.*, ideal scenario).

| Method | PACS | | | | DomainNet | | | |
|---|---|---|---|---|---|---|---|---|
| | ID | | OOD | | ID | | OOD | |
| | $A_{\text{AUC}}\uparrow$ | $A_{last}\uparrow$ | $A_{\text{AUC}}\uparrow$ | $A_{last}\uparrow$ | $A_{\text{AUC}}\uparrow$ | $A_{last}\uparrow$ | $A_{\text{AUC}}\uparrow$ | $A_{last}\uparrow$ |
| CHB | 45.29±1.66 | 37.27±3.23 | 29.89±0.95 | 20.41±1.31 | 17.33±0.15 | 13.55±0.59 | 5.83±0.06 | 4.06±0.20 |
| (+) CONAN | 48.29±1.61 | 45.64±2.35 | 31.32±0.58 | 25.14±0.51 | 30.46±1.11 | 24.65±1.83 | 12.42±0.66 | 6.31±1.76 |
| SC | 47.26±2.46 | 45.70±1.14 | 30.62±1.67 | 22.72±1.11 | 12.47±0.27 | 10.15±0.26 | 4.11±0.24 | 3.13±0.13 |
| (+) CONAN | 50.02±2.66 | 49.97±2.09 | 31.02±2.01 | 23.29±1.93 | 21.11±0.29 | 18.88±0.29 | 6.14±0.12 | 5.25±0.03 |
| CCG | 47.55±2.76 | 45.46±1.76 | 31.06±1.60 | 24.08±0.72 | 16.98±0.27 | 15.15±0.26 | 5.21±0.09 | 4.30±0.07 |
| (+) CONAN | 48.18±2.59 | 49.45±1.17 | 31.72±1.76 | 25.03±0.90 | 16.98±0.27 | 15.15±0.26 | 5.21±0.09 | 4.30±0.07 |
| HIRPG | 48.57±2.40 | 48.73±1.67 | 33.61±1.78 | 27.34±1.10 | 20.64±0.61 | 16.68±0.38 | 8.71±0.26 | 6.83±0.10 |
| (+) CONAN (**Ours**) | **50.89±1.94** | **49.54±2.13** | **34.47±1.07** | **28.89±0.78** | **30.56±1.37** | **25.78±0.56** | **14.95±1.80** | **10.68±0.87** |
| MA | 67.10±4.07 | 61.95±0.92 | 27.75±1.44 | 20.90±0.95 | 51.13±0.28 | 42.95±0.15 | 13.48±0.09 | 10.69±0.07 |

Table 21: **Quantitative comparison of name-only baselines using Qwen3-8B as the LLM.** MA refers to training a model with manually annotated data. The combination of HIRPG and CONAN refers to our proposed GenOL. We bolded the highest accuracy, excluding the MA data (*i.e.*, ideal scenario).

suggest that as stronger LLMs become available, GenOL will be able to take advantage of them to achieve even greater performance.

### A.26 Comparison of CONAN with baselines using few-shot real samples

In real-world applications, beyond the name-only setup, several practical examples can be integrated with the concept. Accordingly, we extend our evaluation by comparing the proposed GenOL with few-shot baselines (*e.g.*, DreamCache (Aiello et al., 2025), DreamBooth (Ruiz et al., 2023), Bootpig (Purushwalkam et al., 2024)) that employ few-shot real samples for generation and summarize the results in Fig. 17. Since few-shot baselines fine-tune generative models using real samples, we also fine-tune the generative models in GenOL using LoRA, following Ruiz et al. (2023), for a fair comparison. Specifically, since GenOL employs multiple generators while few-shot generative baselines use a single generator, we compare the baselines with HIRPG, which excludes the ensemble method (*i.e.*, CONAN) in GenOL and only uses SDXL (Podell et al., 2023). Here, SDXL is fine-tuned with real data. We refer to HIRPG with fine-tuned SDXL as HIRPG-FT. Additionally, for reference, we present the results for the scenario where no real samples are used (*i.e.*, assuming zero-shot real samples). In this case, we refer to the baseline performance of HIRPG and GenOL as HIRPG-ZS and GenOL-ZS, respectively. As shown in the figure, HIRPG-FT outperforms the baselines in both ID and OOD domains on both PACS and DomainNet. This demonstrates that when HIRPG is combined with real sample fine-tuned SDXL, it generates data that is both diverse and highly recognizable, drawing from real data as a reference. Notably, increasing the number of real samples significantly improves ID accuracy; however, for OOD performance, accuracy converges after approximately three examples, with some baselines even showing a slight degradation as more real samples are added. This occurs because fine-tuning generative models leads to image generation that closely matches the style and background of the provided real examples, thereby enhancing ID accuracy. However, this simultaneously reduces dataset diversity and limits OOD generalization.

### A.27 Qualitative Results for Prompt Generation Methods

We also qualitatively evaluate the performance of our proposed prompt generation method, HIRPG, against existing prompt diversification baselines, including LE, CHB, SC, and CCG. The comparison is illustrated across multiple concepts from the PACS and DomainNet datasets, as shown in Table 22, Table 23, and Table 24. We observe that most methods are not able to generate diverse prompts as well as maintain coherence and logic across generated instances. Common issues across baseline methods include irrelevant content, repetitions, and overused phrases.

LE generates repetitive phrases across difference concepts that, while slightly different in wording, essentially convey the same meaning. In Table 22, despite that phrases differ in their choice of words, they describe the same visual concept: a subject illuminated by soft, warm light, typically seen at sunrise or sunset. CHB, on the other hand, generate prompts with nonsensical combinations of objects and environments, such as "house

inside an aquarium". While diverse, the prompts are not grounded in reality, which limits their practical use in downstream tasks.

SC and CCG methods produce more coherent and consistent prompts. However, they show a tendency toward redundancy, particularly in descriptors like "majestic" and "gallops" for the *horse* concept, or "charming" and "rustic" for the *house* concept, reducing the overall uniqueness of the generated prompts. Compared to these existing prompt diversification baselines, our proposed prompt generation method, HIRPG, successfully captures not only the diversity but also the originality and coherence within its generated prompts.

## Effect of number of real samples

(a) PACS

(b) DomainNet

Figure 17: **Effect of number of real samples on (a) PACS and (b) DomainNet in both ID and OOD accuracy.** HIRPG-ZS and GenOL-ZS refer to the zero-shot performance of HIRPG and GenOL, meaning that no real samples are used. HIRPG-FT refers to the performance of HIRPG after fine-tuning the SDXL with real examples.

| Concept | Method | Examples |
|---------|--------|----------|
| House | LE | • A calm house illuminated by the first light of morning.
• A sunrise house bathed in warm orange hues.
• A dew-drenched house in vibrant sunset colors.
• A serene house under a pastel sunrise.
• A peaceful house bathed in the soft glow of dawn. |
| | CHB | • house, building inside aquarium
• house, building inside bakery
• house, building inside music studio
• house, building inside wave
• house, building inside pizzeria |
| | SC | • house => A charming, rustic house with ivy-covered walls and a thatched roof, nestled amidst a flourishing garden of vibrant, blooming flowers, creating a serene and picturesque scene.
• A charming countryside house with a thatched roof and ivy-covered walls, surrounded by a lush, colorful garden blooming with flowers on a sunny day
• A charming rustic house stands besides a tranquil pond, surrounded by lush greenery and vibrant blooming plants, reflecting its serene image in the still water.
• A charming rustic house, nestled in a vibrant, colorful garden filled with blooming flowers and tall trees, creating a picturesque and cozy scene.
• A charming rustic house stands besides a tranquil pond, surrounded by lush greenery and vibrant blooming plants, reflecting its serene image in the still water. |
| | CCG | • A cozy, red-brick house with a white picket fence stands amidst colorful autumn trees.
• A cozy, rustic house stands peacefully surrounded by vibrant autumn trees and a cobblestone path..
• A cozy, rustic house nestled in a forest, surrounded by vibrant autumn leaves and tall trees.
• A cozy countryside house stands surrounded by vibrant autumn foliage under a clear, blue sky.
• A cozy wooden house surrounded by colorful autumn trees under a clear blue sky. |
| | HIRPG | • A high-contrast black and white photograph of house.
• A serene watercolor painting of house.
• A vibrant photo of house during sunrise with a warm color palette.
• A vivid painting of house using vibrant colors.
• A cinematic wide-angle shot of house at dusk. |

Table 22: **Prompt samples using different prompt generation methods for the concept *House* from PACS dataset.** Irrelevant content, repetitions, and overused phrases are marked in red, brown, and turquoise respectively. HIRPG demonstrates more relevant, diverse, and concise descriptions compared to other prompt generation baselines.

| Concept | Method | Examples |
|---|---|---|
| Diving Board | LE | • Scuba diver lands on the dry seas with his bare feet on a
• The man with the green board doing darts is really nice.
• Man in the snow at the beach... the guy on the scuba boat...
• Man on the seaboard while diving
• A man is caught in a sea of dolphins scuba diving on a |
| | CHB | • diving board, springboard inside home office
• diving board, springboard inside bakery
• diving board, springboard inside music studio
• diving board, springboard inside subway station
• diving board, springboard inside pizzeria |
| | SC | • A sleek, modern diving board extends over a pristine pool at an indoor diving facility, brightly lit with natural light streaming through large windows, casting reflections on the water's surface below.
• A sleek diving board extends over a shimmering blue swimming pool, surrounded by lush greenery and bustling with activity under the bright summer sun.
• A sleek diving board extends over an oceanview pool, with the shimmering blue waters of the ocean and a clear sky stretching out in the background.
• On a serene summer day, a sleek diving board extends over a sparkling zero-entry pool, inviting swimmers to take the plunge into its crystal-clear, gradually deepening waters.
• A sleek diving board extends over a crystal-clear synchronized swimming pool, surrounded by vibrant, choreographed swimmers creating mesmerizing patterns in the water. |
| | CCG | • A young girl prepares to leap off a wooden diving board into a sparkling pool below.
• A young girl jumps joyously off a colorful diving board into the sparkling blue pool below.
• A young girl poised on the diving board, ready to leap into the sparkling pool below.
• A young girl leaps joyfully off a high diving board into a sparkling blue pool.
• A young girl stands poised on a diving board, ready to leap into the sparkling pool. |
| | HIRPG | • A high-contrast black and white photograph of diving board.
• A serene watercolor painting of diving board.
• A vibrant photo of diving board during sunrise with a warm color palette.
• A vivid painting of diving board using vibrant colors.
• A cinematic wide-angle shot of diving board at dusk. |

Table 23: **Prompt samples using different prompt generation methods for the concept *Diving Board* from DomainNet dataset.** Irrelevant content, repetitions, and overused phrases are marked in red, brown, and turquoise respectively. HIRPG demonstrates more relevant, diverse, and concise descriptions compared to other prompt generation baselines.

| Concept | Method | Examples |
|---|---|---|
| The Great Wall of China | LE | <ul><li>The great wall of china was completed</li><li>The great wall of china</li><li>The Great Wall of the China</li><li>The great wall of the province of the city of Beijing</li><li>The Wall of China... a building in the countryside</li></ul> |
| | CHB | <ul><li>The Great Wall of China, wall inside home office</li><li>The Great Wall of China, wall inside bakery</li><li>The Great Wall of China, wall inside music studio</li><li>The Great Wall of China, wall inside subway station</li><li>The Great Wall of China wall inside pizzeria</li></ul> |
| | SC | <ul><li>The Great Wall of China winds majestically through the dense, vibrant greenery of a bamboo forest, creating a striking contrast between ancient architecture and natural beauty.</li><li>The Great Wall of China winds majestically through the landscape, surrounded by ancient historical courtyards that echo with the rich history of past dynasties.</li><li>The Great Wall of China winds majestically through the dense, vibrant greenery of a bamboo forest, creating a striking contrast between ancient architecture and natural beauty.</li><li>Majestic Great Wall of China winding through lush green hills, while sailboats gently glide across a serene lake in the foreground.</li><li>The Great Wall of China majestically winds its way through a vibrant, sunny meadow, with lush green grass and colorful wildflowers stretching out in the foreground.</li></ul> |
| | CCG | <ul><li>The ancient Great Wall of China winds through lush green hills under a bright blue sky.</li><li>Tourists wander along the ancient, winding Great Wall of China amidst lush mountains.</li><li>Tourists hike along the ancient stone path of the Great Wall winding through green hills.</li><li>Visitors hike along the winding, ancient Great Wall of China amidst lush, rolling green hills.</li><li>A majestic stretch of ancient stone wall winds over lush, rolling hills under a bright sky.</li></ul> |
| | HIRPG | <ul><li>A high-contrast black and white photograph of The Great Wall of China.</li><li>A serene watercolor painting of The Great Wall of China.</li><li>A vibrant photo of The Great Wall of China during sunrise with a warm color palette.</li><li>A vivid painting of The Great Wall of China using vibrant colors.</li><li>A cinematic wide-angle shot of The Great Wall of China at dusk.</li></ul> |

Table 24: **Prompt samples using different prompt generation methods for the concept *The Great Wall of China* from DomainNet dataset.** Irrelevant content, repetitions, and overused phrases are marked in red, brown, and turquoise respectively. HIRPG demonstrates more relevant, diverse, and concise descriptions compared to other prompt generation baselines.

### A.28 Expanding GenOL to the Joint Training Setup

We extend our proposed GenOL to the standard learning setup (*i.e.*, joint training setup), where all concepts to be learned are provided at once. In this setting, we compare GenOL not only with training-based methods, such as GLIDE, but also with training-free methods (*i.e.*, CLIP-ZS (Radford et al., 2021), SuS-X-SD (Udandarao et al., 2023), VisDesc (Menon & Vondrick, 2023), SD-Clf (Li et al., 2023a), and CUPL (Pratt et al., 2023)) that leverage pre-trained Vision-Language Models (VLMs), such as CLIP (Radford et al., 2021) or generative models, such as SDXL (Podell et al., 2023). Note that although these methods do not update model weights, they generate images for support sets or create customized prompts using LLMs to classify the target concept. We provide a detailed explanation of training-free baselines in Section A.10.

We first compare these methods using the same model, *i.e.*, ResNet-50-CLIP, a CLIP model with ResNet-50 as the vision encoder. Specifically, we utilize the YFCC100M (Thomee et al., 2016) pre-trained CLIP model. We summarize the results in Table 25. For training-dependent methods, we train the model for 10 epochs with the same amount of data across all baselines for a fair comparison. As shown in the table, GenOL significantly outperforms existing name-only classification baselines, as well as combinations of baselines with our proposed data ensemble method, *i.e.*, CONAN. Furthermore, compared to diverse prompt generation baselines (LE, CHB, SC, and CCG), our proposed HIRPG outperforms in both setups—with and without CONAN —demonstrating the effectiveness of our proposed components in a joint training setup.

Next, we compare the results with those obtained using only the CLIP-pretrained ResNet-50 for training-dependent methods. While the CLIP model can be employed for training-dependent methods, training VLMs demands substantial computational resources, compared to training-free baselines, hindering real-time adaptation and limiting their deployment in real-world applications(Koh et al., 2021; Caccia et al., 2022). Therefore, to improve training efficiency and enable faster adaptation to newly encountered concepts, we also compare the results of training solely on the vision encoder of the CLIP model for training-dependent methods. To assess training efficiency, we train them for 10 epochs, consistent with Table 25, and summarize the results in Table 26.

As shown in the table, several training-free methods outperform GenOL in the in-domain (ID) scenario. This advantage arises because they utilize off-the-shelf CLIP models, which are pre-trained on large-scale datasets, particularly in the photo domain, which we consider as ID in our experiments. However, despite the benefits of large-scale pre-training, these methods struggle to generalize in OOD scenarios.

In contrast, GenOL not only outperforms all baselines but also surpasses a model trained with manually annotated data in the OOD domains of PACS, CIFAR-10-W, and DomainNet. This demonstrates that large-scale pre-training alone does not guarantee good generalization across all downstream tasks in various domains, highlighting the necessity of few-epoch training for personalization and real-time adaptation in the name-only setup.

### A.29 Qualitative Analysis

We qualitatively compare web-scraped images, manually annotated images, and GenOL-generated images, highlighting diversity and recognizability of GenOL-generated images.

**Multi-modal Visual-concept Incremental Learning Setup.** In addition to Section 5.2, we qualitatively compare samples acquired through different data acquisition methods for the given concept in the Bongard-OpenWorld datasets, as shown in Figure 18. In Figure 18, GenOL effectively generates both positive and negative support sets. In contrast, web-scraped data include images that do not match the given concept '*A leopard relaxing on a tree branch.*' This discrepancy arises from the lengthy and free-form concepts used in Bongard OpenWorld, such as descriptive sentences, compared to the simpler object-action combinations in Bongard-HOI. In web scraping, those detailed and lengthy search queries may yield unrelated results.

Note that GenOL relies solely on positive concepts, as mentioned in Section 5.2. Specifically, in manually annotated (MA) data, high-quality annotators not only select positive support sets but also curate hard negative examples for the negative sets. In contrast, GenOL utilizes only positive concepts (*i.e.*, concepts that the model needs to learn) and automatically generates hard negative examples using text prompts created

| Type | Training Data | CIFAR-10-W | DomainNet | |
|---|---|---|---|---|
| | | OOD | ID | OOD |
| Training-free | CLIP-ZS (Radford et al., 2021) | 14.69 | 5.17 | 4.9 |
| | SuS-X-SD (Udandarao et al., 2023) | 53.08 | 20.06 | 7.5 |
| | VisDesc (Menon & Vondrick, 2023) | 51.83 | 16.87 | 6.52 |
| | CuPL (Pratt et al., 2023) | 50.5 | 18.25 | 6.36 |
| | CALIP (Guo et al., 2023) | 51.62 | 16.43 | 6.39 |
| | SD-Clf (Li et al., 2023a) | 52.48 | 12.27 | 11.85 |
| Training-dependent | Glide-syn (He et al., 2023) | 55.93 | 38.26 | 9.31 |
| | LE (He et al., 2023) | 73.51 | 47.43 | 14.7 |
| | (+) CONAN | 75.13 | 52.87 | 17.26 |
| | CHB (Sarıyıldız et al., 2023) | 70.61 | 45.28 | 14.62 |
| | (+) CONAN | 75.96 | 52.31 | 17.49 |
| | SC (Tian et al., 2024a) | 71.3 | 40.42 | 12.36 |
| | (+) CONAN | 75.04 | 49.64 | 15.19 |
| | CCG (Hammoud et al., 2024) | 58.25 | 39.32 | 11.57 |
| | (+) CONAN | 63.14 | 42.94 | 14.37 |
| | HIRPG | 74.47 | 52.30 | 20.18 |
| | (+) CONAN (**Ours**) | **77.64** | 54.85 | **22.66** |
| | Manually Annotated | 59.12 | **71.13** | 20.29 |

Table 25: **Quantitative comparison between different name-only baselines on joint training setup.** We employ the YFCC100M pre-trained ResNet50-CLIP, which uses ResNet50 as the vision encoder for the CLIP model, for all methods.

by large language models (LLMs), as demonstrated in Section A.6. Nonetheless, as shown in Figure 18, the negative samples generated by GenOL are not clearly distinct from the positive examples, which enhances the model's ability to differentiate between the concepts.

**Class Incremental Learning Setup.** We compare samples acquired through different generative name-only baselines in the CIL setup, as illustrated in Figure 19 and Figure 20.

## A.30 Manual Annotation vs. Web-Scraping vs. Generative data

In modern deep learning, the trajectory of advancement is heavily influenced by the exponential growth of training data and the corresponding models trained on these vast datasets. Foundation models are typically exposed to datasets in the order of billions during training, obtained predominantly through web scraping (Schuhmann et al., 2022; Xue et al., 2020; Zhu et al., 2024; Gao et al., 2020; Kocetkov et al., 2022; Bain et al., 2021). Although web scraping is a cost-effective method to produce high-quality datasets, studies underscore issues such as potential biases (Foerderer, 2023; Packer et al., 2018; Caliskan et al., 2017), copyright, privacy, and license concerns (Quang, 2021; Solon, 2019), and the risks of data contamination (Dekoninck et al., 2024; Li, 2023) or data leakage from evaluation (Balloccu et al., 2024).

As demonstrated in Figure 1, we highlight the key differences between Manually Annotated (MA), Web scraped, and Generated data on six different axes: (a) Controllability, (b) Storage issues, (c) Usage restrictions, (d) Privacy issues, (e) Acquisition cost, and (f) Noise. In this section, we aim to provide the definition of each of these axes and their corresponding implications on each type of data source.

| Type | Training Data | PACS | | DomainNet | | CIFAR-10-W |
|---|---|---|---|---|---|---|
| | | ID | OOD | ID | OOD | OOD |
| Training-free | CLIP-ZS (Radford et al., 2021) | **99.11** | 49.12 | 14.69 | 5.17 | 57.14 |
| | SuS-X-SD (Udandarao et al., 2023) | 95.55 | 47.81 | 20.06 | 7.5 | 53.08 |
| | VisDesc (Menon & Vondrick, 2023) | 93.77 | 46.09 | 16.87 | 6.52 | 51.83 |
| | CuPL (Pratt et al., 2023) | 89.32 | 46.51 | 18.25 | 6.36 | 50.50 |
| | CALIP (Guo et al., 2023) | 92.58 | 48.43 | 16.43 | 6.39 | 51.62 |
| | SD-Clf (Li et al., 2023a) | 92.58 | 48.43 | 12.27 | 11.85 | 52.48 |
| Training-dependent | Glide-syn (He et al., 2023) | 85.16 | 33.2 | 29.02 | 6.73 | 55.93 |
| | LE (He et al., 2023) | 88.43 | 38.03 | 40.74 | 10.47 | 73.51 |
| | (+) CONAN | 93.47 | 44.54 | 54.60 | 15.62 | 75.13 |
| | CHB (Sarıyıldız et al., 2023) | 83.38 | 30.98 | 35.97 | 9.60 | 70.61 |
| | (+) CONAN | 92.88 | 41.42 | 49.17 | 15.51 | 75.96 |
| | SC (Tian et al., 2024a) | 76.26 | 28.19 | 30.42 | 8.23 | 71.30 |
| | (+) CONAN | 85.46 | 42.05 | 44.66 | 12.01 | 75.04 |
| | CCG (Hammoud et al., 2024) | 81.01 | 31.71 | 26.59 | 6.89 | 58.25 |
| | (+) CONAN | 85.76 | 41.55 | 32.31 | 8.72 | 63.14 |
| | HIRPG | 89.91 | 43.98 | 46.19 | 17.80 | 74.47 |
| | (+) CONAN (**Ours**) | 94.36 | **60.75** | 51.85 | **21.01** | **77.64** |
| | Manually Annotated | 97.03 | 33.80 | **72.54** | 19.09 | 59.12 |

Table 26: **Quantitative comparison between different name-only baselines on joint training setup.** We employ the YFCC100M pre-trained ResNet50-CLIP for training-free methods, while for training-dependent methods, we utilize only the vision encoder of the CLIP model.

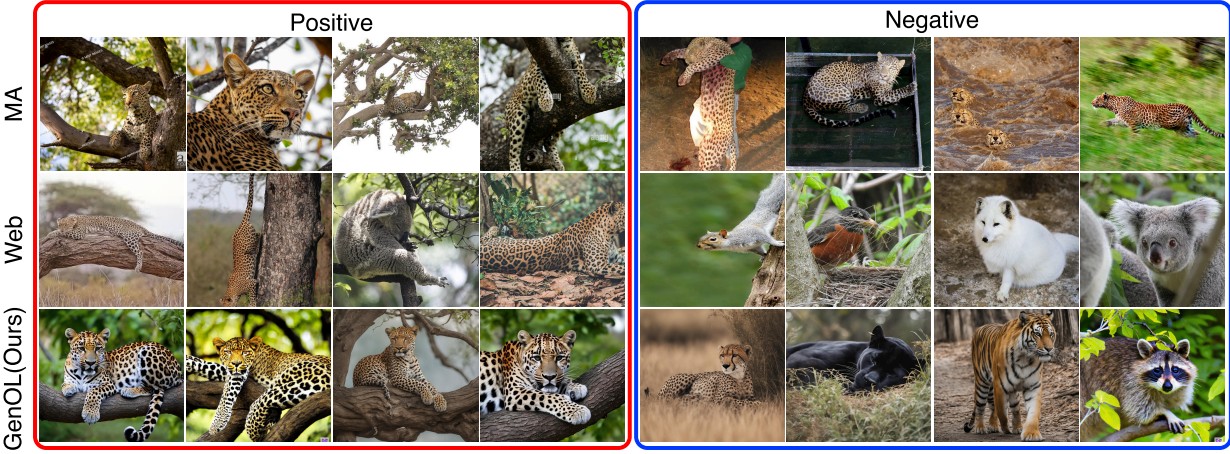

Figure 18: **Samples using different data acquisition methods for the same concept in the MVCIL setup.** The given concept is *'A leopard relaxing on a tree branch'* from the Bongard-OpenWorld benchmark. The left four images represent positive examples that depict the given concept, while the right four images represent negative examples that illustrate different concepts. Here, 'MA' refers to manually annotated data.

**Controllability.** encompasses the ability to generate or acquire images with various contexts, backgrounds, settings, and themes as desired. It pertains to the ability to obtain images depicting different concepts in compositions not commonly found in natural environments, as well as in domains relevant to the task at hand. Under this definition, we assert that the MA data exhibit low controllability. This limitation arises from its reliance on data captured from a finite set of scenarios or sensors, which inherently restricts the breadth of diverse settings where the concept can be observed. Web-scrapped data also suffer from low controllability for the same reasons. In contrast, the generated data have high controllability due to the ability of foundation text-to-image (T2I) generators to produce diverse images for each concept through varied prompting.

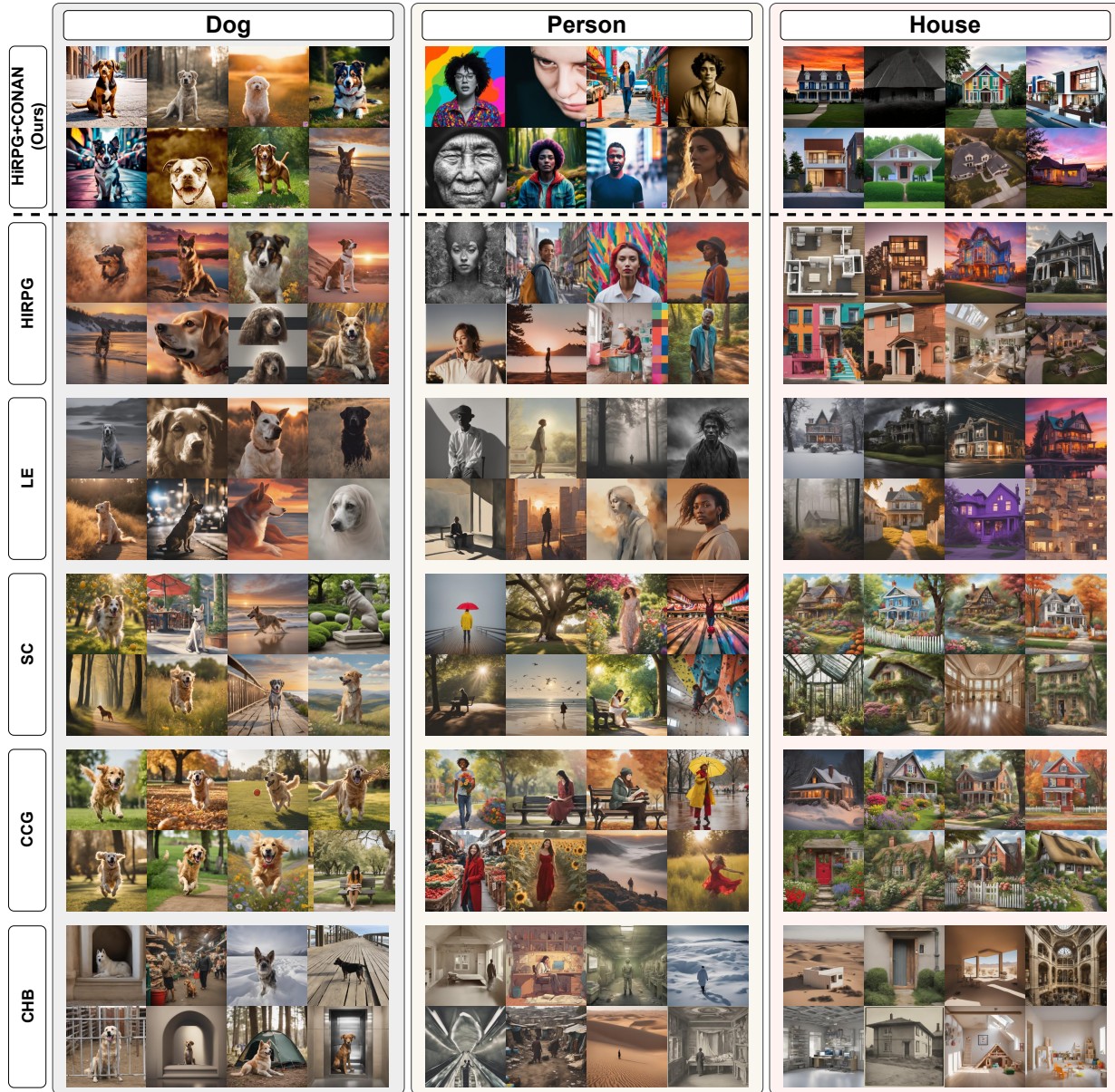

Figure 19: **Samples using different data acquisition methods for the same concept in the CIL setup.** The given concepts are *Dog, Person, House*, which are from PACS. Our proposed GenOL, *i.e.*, HIRPG + CONAN, produces a more diverse set of images that comprehensively cover various domains, ensuring broader representation and adaptability. In contrast, other methods tend to generate similar (e.g, *Dog* in CCG) or domain-limited (e.g, *House* in SC) images.

**Storage Issues.** Storing extensive data, locally or in the cloud, imposes additional costs, which can become impractical in environments constrained by limited total storage capacity. In addition, transmitting large, substantial data samples in a federated setup can face challenges arising from bandwidth and latency bottlenecks. In such contexts, depending on a large corpus of MA data becomes counterintuitive. On the other hand, both web-scraped and generated data present themselves as cost-effective alternatives for accessing substantial data volumes without necessitating explicit storage expenditures.

**Usage Restrictions.** encompass limitations imposed on the use of images for training machine/deep learning models, typically due to copyright or licensing protections. These restrictions arise from various

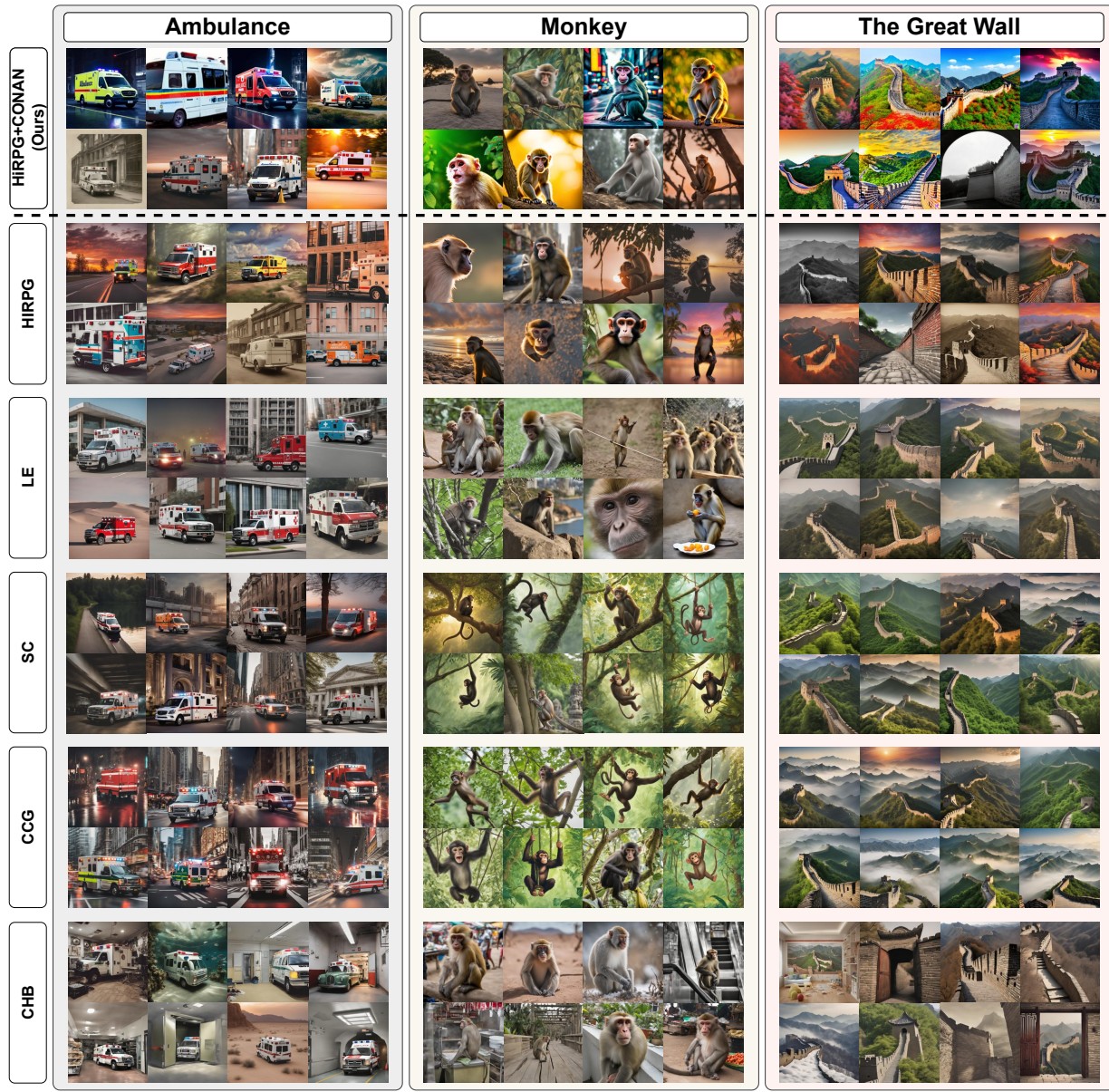

Figure 20: **Samples using different data acquisition methods for the same concept in the CIL setup.** The given concepts are *Ambulance, Monkey, The Great Wall*, which are from DomainNet. Our proposed GenOL, *i.e.*, HIRPG + CONAN, produces a more diverse set of images that comprehensively cover various domains, ensuring broader representation and adaptability. In contrast, other methods tend to generate similar (e.g., *The Great Wall* in LE) or domain-limited (e.g., *Ambulance* in CCG) images.

legal frameworks across different demographics, regulating, and sometimes prohibiting, the training of models on protected data for commercial deployment. This challenge is particularly prevalent in web-scraped data, where the abundance of protected data may not be adequately filtered (Khan & Hanna, 2020). In contrast, MA data bypass this issue, as it is presumed that the data are filtered or obtained from a proprietary source with appropriate permissions during annotation. Notably, generated data offer a more advantageous position, as they do not necessitate such filtering and encounter limited or no usage restrictions, thereby providing a readily available solution to issues arising from data protection concerns.

**Privacy Issues.** may arise when data samples inadvertently leak or explicitly contain sensitive, confidential, or private user information. Examples of such images could include those featuring people's faces (O'Sullivan, 2020; Murgia & Harlow, 2019) or personal objects that disclose identity-related details, such as addresses or financial assets. Once again, web data emerge as the primary source vulnerable to issues stemming from the use of private data, an issue extensively present in web-scraped data (Subramani et al., 2023; Wenger et al., 2022; Solon, 2019; Lukas et al., 2023). MA data are generally protected from privacy concerns due to prior filtering or explicit agreements on data usage before annotation. While generative models may also pose privacy risks, recent studies (Chen et al., 2018; 2019; Carvalho et al., 2022; Xu et al., 2023) have explored methods to enforce fairness and differential privacy in data synthesis, providing solutions for identity protection.

**Acquisition Cost.** refers to the total expenses incurred in obtaining a specific number of data samples necessary to train or evaluate the learner $f_\theta$ for a particular task. As emphasized in Section 1, MA data entail a substantial acquisition cost, primarily due to the expenses associated with densely annotating the data through human workers. This, coupled with the rigorous filtering process, makes MA data prohibitively expensive to acquire at scale. Although web data do not require such significant financial outlay for annotation, they do require intensive filtering, which contributes to an elevated cost and poses a barrier to constructing large datasets solely from web sources. In contrast, due to the advantages in controllability, generated data boast a notably low acquisition cost for generating large and diverse datasets.

**Noise.** pertains to instances where data that are not related to a concept are erroneously labeled as belonging to that concept. It may also mean discrepancies between the context of the data and the associated concept. As highlighted in Section A.18, web data often exhibit a high degree of noise, necessitating extensive filtering or label correction processes. In contrast, both MA data and generated data are less susceptible to such noise. In the case of MA data, the presumption of prior filtering serves as a primary solution to mitigate noisy data. Meanwhile, for generated data, the advantages of controllability enable the mitigation of noise resulting from inconsistencies in concept-image alignment. Despite the drawback of requiring GPU usage, T2I model inference incurs lower costs compared to MA due to its ability to generate pure images, making it a more cost-effective option.

### A.31 Extended Quantitative Analysis

We perform additional comparisons with various combinations of diverse prompt generation baselines and data ensemble methods on DomainNet, and summarize the results in Table 27. For all data ensemble methods, including our proposed CONAN, we select an equal number of samples for the coreset to ensure a fair comparison. As shown in the table, CONAN is not only effective when combined with HIRPG, but also shows strong performance when paired with other prompt generation baselines, demonstrating its plug-and-play applicability across various methods.

### A.32 Details of RMD Score Calculation

The RMD (Cui et al., 2023) score of a sample $(x_i, y_i)$ is defined as follows:

$$\mathcal{RMD}(x_i, y_i) = \mathcal{M}_{cls}(x_i, y_i) - \mathcal{M}_{\text{agn}}(x_i),$$
$$\mathcal{M}_{cls}(x_i, y_i) = -\left(G(x_i) - \mu_{y_i}\right)^T \Sigma^{-1} \left(G(x_i) - \mu_{y_i}\right),$$
$$\mathcal{M}_{\text{agn}}(x_i) = -\left(G(x_i) - \mu_{agn}\right)^T \Sigma_{agn}^{-1} \left(G(x_i) - \mu_{agn}\right), \tag{11}$$

where $G$ represents the feature extractor, $\mathcal{M}_{cls}(x_i, y_i)$ denotes the Mahalanobis distance from $G(x_i)$ to the corresponding class mean vector $\mu_{y_i} = \frac{1}{N_i} \sum_{y_j = y_i} G(x_j)$, with $N_i$ being the count of samples labeled as $y_i$, $\Sigma^{-1}$ denotes the inverse of the averaged covariance matrix across classes. Furthermore, $\mathcal{M}_{\text{agn}}(x_i)$ represents the class-agnostic Mahalanobis distance, where $\mu_{\text{agn}}$ denotes the overall sample mean, and $\Sigma_{\text{agn}}^{-1}$ denotes the inverse covariance for the class-agnostic case.

In the online CL setup, where data arrive in a continuous stream, it is not feasible to calculate $\mu$ and $\Sigma$ of the entire dataset. Instead, a necessity arises to continuously update these statistical parameters to accommodate the dynamic nature of the incoming data stream.

| Method | ID | | OOD | |
|---|---|---|---|---|
| | $A_{\mathrm{AUC}}\uparrow$ | $A_{last}\uparrow$ | $A_{\mathrm{AUC}}\uparrow$ | $A_{last}\uparrow$ |
| LE | 20.01±0.27 | 15.38±0.31 | 6.40±0.13 | 4.59±0.09 |
| (+) Uncertainty | 14.66±0.30 | 9.40±0.14 | 4.85±0.08 | 3.08±0.06 |
| (+) CRAIG | 28.64±0.55 | 23.91±0.27 | 8.53±0.27 | 6.83±0.07 |
| (+) Glister | 17.53±0.44 | 11.57±0.25 | 5.67±0.15 | 3.61±0.06 |
| (+) GradMatch | 27.68±0.68 | 22.89±0.14 | 8.57±0.31 | 6.86±0.07 |
| (+) AdaCore | 24.73±0.56 | 18.95±0.25 | 7.62±0.17 | 5.53±0.12 |
| (+) LCMat | 27.72±0.49 | 23.10±0.23 | 8.50±0.22 | 6.84±0.02 |
| (+) Moderate | 21.33±0.54 | 15.91±0.27 | 6.47±0.20 | 4.56±0.11 |
| (+) CONAN | **30.80±0.63** | **25.33±0.20** | **9.54±0.25** | **7.59±0.17** |
| CHB | 16.69±0.16 | 13.45±0.19 | 5.61±0.11 | 4.18±0.05 |
| (+) Uncertainty | 11.15±0.35 | 7.06±0.15 | 3.97±0.09 | 2.41±0.05 |
| (+) CRAIG | 26.42±0.35 | 22.49±0.33 | 8.11±0.09 | 6.61±0.20 |
| (+) Glister | 14.07±0.13 | 9.38±0.06 | 4.68±0.05 | 2.98±0.04 |
| (+) GradMatch | 25.20±0.36 | 21.58±0.27 | 7.97±0.15 | 6.51±0.14 |
| (+) AdaCore | 22.29±0.31 | 17.27±0.16 | 7.23±0.09 | 5.27±0.08 |
| (+) LCMat | 24.99±0.37 | 21.46±0.24 | 7.99±0.12 | 6.68±0.10 |
| (+) Moderate | 18.64±0.24 | 13.96±0.10 | 5.92±0.07 | 4.08±0.06 |
| (+) CONAN | **29.06±0.37** | **24.52±0.17** | **9.28±0.14** | **7.56±0.14** |
| SC | 11.89±0.17 | 8.66±0.20 | 3.90±0.07 | 2.68±0.04 |
| (+) Uncertainty | 10.32±0.26 | 6.41±0.20 | 3.17±0.05 | 1.87±0.05 |
| (+) CRAIG | 20.05±0.25 | 17.13±0.16 | 6.02±0.12 | 4.83±0.08 |
| (+) Glister | 11.30±0.24 | 7.24±0.08 | 3.42±0.07 | 2.10±0.04 |
| (+) GradMatch | 19.83±0.38 | 16.82±0.19 | 5.94±0.10 | 4.78±0.08 |
| (+) AdaCore | 17.67±0.37 | 13.29±0.38 | 5.19±0.13 | 3.69±0.06 |
| (+) LCMat | 19.86±0.32 | 16.79±0.29 | 5.98±0.11 | 4.77±0.07 |
| (+) Moderate | 14.17±0.24 | 10.34±0.06 | 4.03±0.07 | 2.72±0.05 |
| (+) CONAN | **22.36±0.34** | **19.13±0.32** | **6.71±0.15** | **5.48±0.13** |
| CCG | 12.55±0.22 | 10.21±0.26 | 4.03±0.10 | 2.91±0.10 |
| (+) Uncertainty | 14.73±0.41 | 10.65±0.22 | 4.48±0.14 | 3.13±0.07 |
| (+) CRAIG | 16.72±0.27 | 14.23±0.30 | 5.16±0.08 | 4.11±0.05 |
| (+) Glister | 14.51±0.38 | 10.54±0.15 | 4.46±0.12 | 3.08±0.03 |
| (+) GradMatch | 16.75±0.26 | 14.31±0.21 | 5.15±0.11 | 4.10±0.07 |
| (+) AdaCore | 17.11±0.36 | 13.87±0.18 | 5.20±0.14 | 3.93±0.11 |
| (+) LCMat | 16.71±0.23 | 14.08±0.26 | 5.15±0.09 | 4.05±0.05 |
| (+) Moderate | 14.52±0.29 | 11.01±0.14 | 4.40±0.10 | 3.15±0.07 |
| (+) CONAN | **18.32±0.42** | **15.83±0.34** | **5.78±0.17** | **4.70±0.14** |
| HIRPG (**Ours**) | 27.72±0.30 | 23.71±0.39 | 10.70±0.19 | 8.75±0.13 |
| (+) Uncertainty | 21.90±0.37 | 15.70±0.08 | 10.01±0.23 | 7.19±0.11 |
| (+) CRAIG | 32.53±0.20 | 28.44±0.23 | 13.25±0.15 | 11.53±0.06 |
| (+) Glister | 23.16±0.37 | 16.98±0.35 | 10.56±0.26 | 7.60±0.18 |
| (+) GradMatch | 32.53±0.43 | 28.36±0.41 | 13.48±0.31 | 11.74±0.18 |
| (+) AdaCore | 32.15±0.55 | 26.83±0.18 | 13.62±0.27 | 11.37±0.04 |
| (+) LCMat | 32.38±0.44 | 28.36±0.32 | 13.42±0.26 | 11.76±0.17 |
| (+) Moderate | 25.57±0.42 | 20.38±0.16 | 10.53±0.29 | 8.17±0.13 |
| (+) CONAN (**Ours**) | **34.60±0.31** | **30.09±0.11** | **14.53±0.22** | **12.65±0.09** |

Table 27: **Quantitative comparison between data selection methods with different diverse prompt generation baselines on DomainNet.** Uncertainty, CRAIG, Glister, GradMatch, Adacore, and LCMat require fine-tuning on the full dataset to compute gradients of the fine-tuned model, despite using a pre-trained model for initialization. In contrast, Moderate and CONAN do not require any fine-tuning.

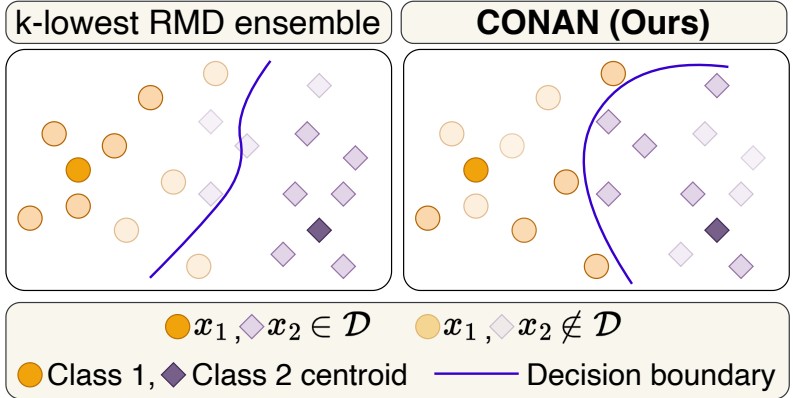

Figure 21: **CONAN** helps in finding a tighter decision boundary due to having a higher probability of including high RMD scored samples in the ensemble of generated data $\mathcal{D}$. Intuitively, high RMD scored samples are farther away from their class prototype but closer to other class samples in comparison to low RMD scored samples which are concentrated closer around the class prototype. Note that CONAN includes not only high RMD samples but also some low RMD (*i.e.*, class-representative) samples.

Starting with the initially computed mean vector $\mu_{y_i}$ and the covariance matrix $\Sigma$ from $N$ samples, the arrival of a new sample $x_{N+1}$ triggers an update. The updated mean vector $\mu_{\text{new}}$ is computed incrementally using a simple moving average (SMA), as follows:

$$\mu_{\text{new}} = \frac{N\mu_{\text{old}} + x_{N+1}}{N+1}. \tag{12}$$

Similarly, we calculate $\Sigma$ using a simple moving variance. Specifically, the update for the new covariance matrix $\Sigma_{\text{new}}$ is calculated using the deviation of the new sample from the old mean $\Delta = x_{N+1} - \mu_{\text{old}}$, and its deviation from the new mean $\Delta_{\text{new}} = x_{N+1} - \mu_{\text{new}}$.

Formally, we formulate the update process as follows:

$$\Sigma_{\text{new}} = \frac{1}{N+1} \left( N\Sigma_{\text{old}} + \Delta\Delta_{\text{new}}^T \right). \tag{13}$$

The update process for the class-agnostic mean vector $\mu_{\text{agn}}$ and covariance $\Sigma_{\text{agn}}$ follows the same incremental approach as described for the class-specific components.

CONAN includes many samples with high RMD scores in the ensemble dataset. Not only does it include samples with the highest RMD scores, but it also probabilistically incorporates samples with low RMD scores. This approach ensures a core-set ensemble, and we illustrate the effect of CONAN in Figure 21

### A.33 Scaling Behavior

Recent scaling law studies (Hernandez et al., 2021; Hoffmann et al., 2022) offer predictive insight into model performance by scaling computation, data, and model capacity. Despite the limited exploration of scaling in continual learning settings (Ramasesh et al., 2022), and particularly with synthetic data (Fan et al., 2024) being confined to static frameworks, our empirical analysis in Figure 22 delves into scaling dynamics with varying proportions of generated data for online continual learning setup.

For ResNet18 (He et al., 2016) and ViT (Dosovitskiy & Brox, 2016), we observe a consistent linear improvement trend in both ID and OOD $A_{\text{AUC}}$ as the volume of generated data increases, across the PACS (Zhou et al., 2020) dataset. This scaling behavior underscores the positive correlation between performance improvement and larger generated data ensembles in online continual learning, reinforcing the rationale for the use of generators in the absence of annotated data.

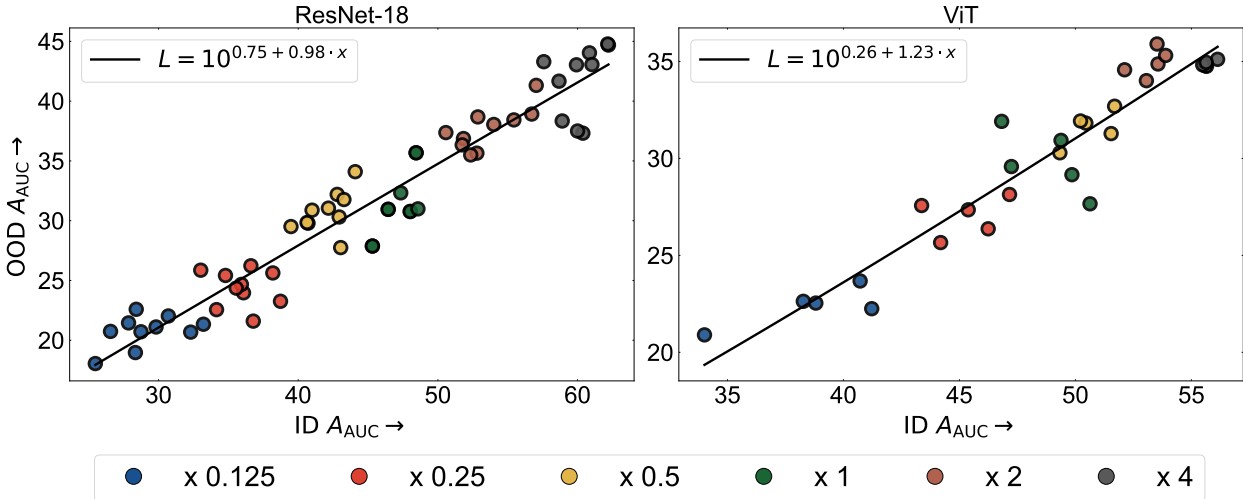

Figure 22: **Ensemble scaling behavior of (a) ResNet18 (He et al., 2016) and (b) ViT (Dosovitskiy & Brox, 2016) for ID $A_{\mathrm{AUC}}$ vs. OOD $A_{\mathrm{AUC}}$ on the PACS dataset (Zhou et al., 2020) using ER (Rolnick et al., 2019).** A consistent linear improvement trend exists in both ID and OOD $A_{\mathrm{AUC}}$ as the volume of generated data increases. (x 1) denotes the ensemble volume in primary experiments, the default data budget.

### A.34 Analysis of Similarity between Prompts Generated by HIRPG

We empirically demonstrate that the overlap between generated prompts from different nodes is rare by measuring the similarity of prompts generated at each node. To measure similarity, we first construct a K-ary Tree with $K = 7$ and $D = 2$. Next, we divide the generated prompts into 7 groups, where $i$-th group refers to the set of prompts generated by RPG using the prompt of $i$-th node at depth 1 is used as the initial negative prompt (*i.e.*, $P_B$). We then measure the group-wise similarity by calculating the average of all pairwise similarities between prompts generated from each node, where the similarity between prompts is measured by the cosine similarity of their text embeddings extracted by Sentence-BERT (Reimers, 2019). We summarize the group-wise similarity of generated prompts in a similarity matrix, as shown in Figure 23.

As shown in the similarity matrix, the similarities between the same nodes (*i.e.*, antidiagonal elements) are generally low. This is because prompts generated at the same node are iteratively generated while considering each other as negative examples. Moreover, similarities between different nodes (*i.e.*, non-antidiagonal elements) are also generally low, although slightly higher than the antidiagonal elements. We believe this is attributed to a characteristic of LLM, where different examples provided during in-context learning lead to varied outputs (Su et al., 2022; Agarwal et al., 2024). Specifically, since RPG at each node begins with distinct initial negative examples passed from the previous depth, it generates different prompts, even though the model cannot directly reference prompts generated by other nodes as negative examples.

### A.35 Broader Impact Statement

Our work involves generating training data using generative models, which may raise potential concerns regarding privacy and bias. To mitigate privacy concerns, we exclude any person-related generated data from the released dataset, as discussed in Section 7. We believe that advancing privacy-preserving (Xu et al., 2023; Chen & Yan, 2024) and differential privacy (Chen et al., 2023a) generative models would effectively mitigate these concerns.

Regarding bias, if the generated data contain bias and the model is continually trained on such data, the bias is likely to be amplified over time. Although we demonstrate in Fig.4 and Fig.5 of Section 5.3 that GenOL 's diverse image generation implicitly leads to producing less-biased data than MA and other generative baselines, explicit bias-prevention strategies are essential to mitigate the risk of generating harmful biases,

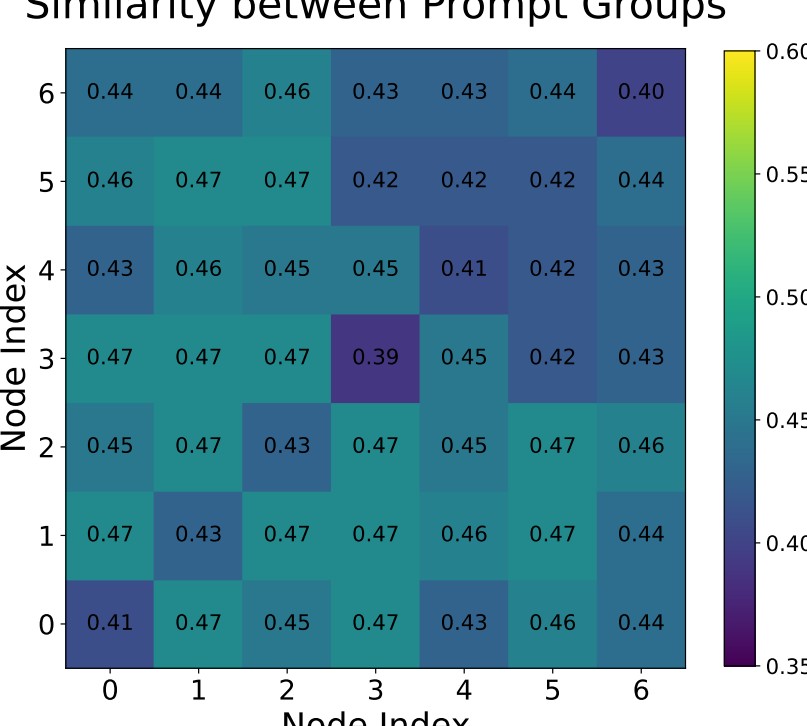

Figure 23: **Similarity matrix of prompts generated at each node in HIRPG in a $K$-ary structure with $D = 2$ and $K = 7$ on DomainNet.** Node index $i$ refers to the set of prompts generated by RPG, where the prompt of the $i$-th node at depth 1 is used as the initial negative example. We measure the similarity between two nodes by calculating the average of all pairwise similarities between prompts generated from each node. The similarity between prompts is measured by the cosine similarity of their respective text embeddings.

which can be critical for real-world deployment. We believe that editing bias-inducing layers or parameters in generative models (Wang et al., 2024; Xu et al., 2025), as well as fine-tuning generative models trained with small and unbiased datasets (Malakouti & Kovashka, 2025), can effectively prevent bias propagation in GenOL and reduce the risk of deploying biased models.

### A.36 Failure Case Analysis of GenOL

While GenOL produces higher-quality and less noisy data than other generative and web-scraping baselines (see Sec.A.29), it sometimes generates unnatural images. For example, in the 'People embracing each other' row of Bongard-OpenWorld (Fig. 24), GenOL produces implausible arm and leg arrangements. Similarly, in the 'Keyboard Piano' row, the generated keyboard contains distorted keys that deviate from a standard piano keyboard. In Bongard-HOI, similar issues appear: in 'Eat Banana' and 'Kick Sports Ball', human limb positions are unrealistic, and in the third column of 'Kick Sports Ball', a figure with three legs is generated. Likewise, in 'Hug Dog', instead of the intended concept of a person hugging a dog, GenOL sometimes generates a dog hugging another dog. These cases highlight that text-to-image generative models often require highly detailed textual descriptions to faithfully capture the intended concept.

Despite these imperfections, we argue that the inclusion of such unnatural images in the training set can be beneficial. Specifically, we believe that exposure to both natural and imperfect samples promotes robustness and improves generalizability, which contributes to the strong out-of-domain performance reported in Table 3,

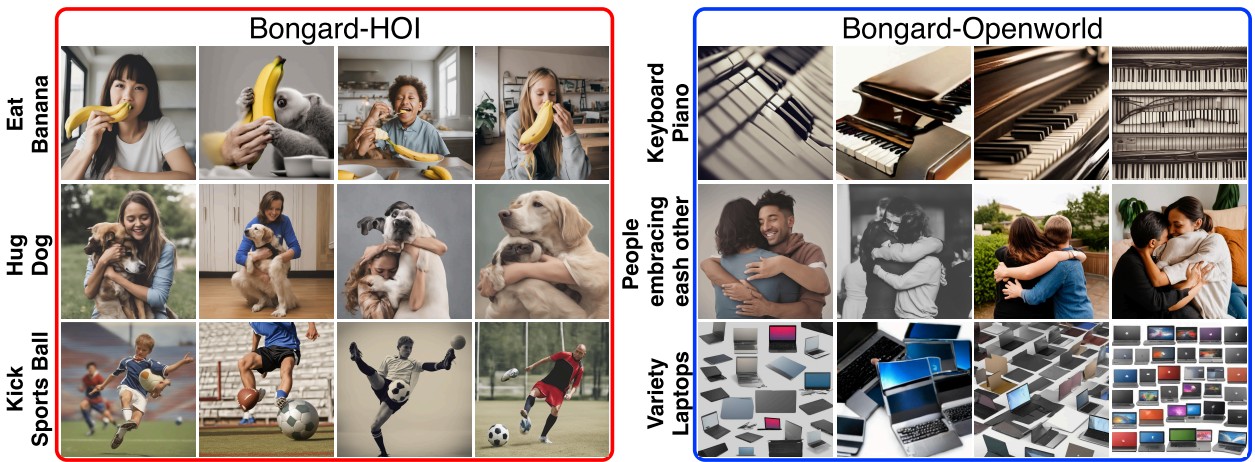

Figure 24: **Failure case of GenOL.** GenOL occasionally produces implausible arm–leg configurations and distorted visual outputs.

Table 2, and Table 1. Consequently, models trained with data generated by GenOL are better equipped to handle noisy or imperfect images, facilitating more reliable real-world deployment.

