# OpenReview forum: "GenOL: Generating Diverse Examples for Name-only Online Learning"
_TMLR — Accepted by TMLR_

### Review · Reviewer_XSeX · 2025-07-15

**Summary Of Contributions:**

This paper presents a new method for the problem of name-only continual learning. Instead of manually annotating samples for new concepts, or fetching samples from the web, new samples are generated using a complex prompt generative module coupled with a new ensembling technique, both ensuring diversity in the generations. The proposed method sets a new SOTA in name-only continual learning on various benchmarks such as PACS or DomainNet.

**Audience:**

Yes

**Broader Impact Concerns:**

No ethical concerns specific to this work, except the use of generative model its associated well-known issues.

**Claims And Evidence:**

Yes

**Requested Changes:**

Major points are:
- Clarifying how the results were run (taken from concurrent papers as opposed to rerun by authors), and explain why
- Study and discussion on the impact of different generative models

Other clarification and presentation points in weaknesses are minor.

**Strengths And Weaknesses:**

Strengths:

- The idea to bring generative models to the name-only continual learning setup is novel and has multiple advantages compared to web search. The mechanisms to ensure diversity: hierarchical prompt generator and coreset selection are also new, very effective in the proposed evaluation setup, and could give inspiration to other generative work.

- Many implementation details for full reproduction of the experiments are available in the appendix. The method is compared to many baselines on a wide set of benchmarks, ensuring the validity of the approach.


Weaknesses:

- Small presentation detail: in the table I would prefer that method names are followed by citation links rather than conference where the work was published.

- Why leaving Imanet-R results in Appendix ? This is a larger scale dataset compared to the others, and the results are good so it would make sense to put these results in the main text. Maybe because of the lack of baselines ? Why not run more baselines on Imagenet-R ?

- My understanding is that most of the results presented in the paper are run by the author themselves. Can you clarify in the text whether this is the case, and for which results ? Why not use numbers in the concurrent papers ?

- The name-only continual learning setup is very niche and feels very artificial, is it related to any real-world application ?

- The use of generative models is not necessarily better than scrapping the web in terms of diversity / privacy / licences. This highly depends on the choice of generative model, the data it was trained on ect. The paper mentions: “However web-scraped data suffer from noise, low controllability, and usage restrictions”, the exact same can apply here. The performance of the model is also highly dependent on the choice of generative model. For example, the paper uses GPT-4o, but how would the performance change if another model was used instead ? Finally, in this case GPT-4o is proprietary, that means less control over what data is used and generated and less understanding of how the model works. I would like to see a discussion or additional experiments with different generative models.

- The introduction should put a bit more emphasis more on the fact that this paper is targeting continual learning with new concepts, as opposed to continual learning under unknown distribution of known concepts.

---

> ### Author Response · Authors · 2025-08-18
> **Answers to the questions of Reviewer XSeX (1/3)**
>
> > **Q1.** Small presentation detail: in Table 2 would prefer that method names are followed by citation links rather than conference where the work was published.
>
> $\to$ Thank you for the suggestion! In the revision, we have updated Tables 2 and 3 to include citation links after the method names instead of the conference venues. For Table 4, we retained the conference venues, as replacing them with citation links would significantly reduce font size and impair readability. Nonetheless, all baselines in Table 4 are already accompanied by citation links in the baseline description paragraph of Sec. 5.1.
>
> > **Q2.** Why is leaving Imanet-R results in the Appendix? This is a larger-scale dataset compared to the others, and the results are good, so it would make sense to put these results in the main text. Maybe because of the lack of baselines? Why not run more baselines on Imagenet-R?
>
> $\to$ As you correctly pointed out, we placed the ImageNet-R results in the appendix due to the lack of baseline results. ImageNet-R includes 1,000 classes, making it significantly more computationally demanding than other benchmarks such as PACS or DomainNet. Given limited time and resources, we compared our method, GenCL, only with the strongest web-scraping-based baseline (i.e., IE) and the strongest generative baseline (i.e., SC).
>
> Recognizing the importance of ImageNet-R for evaluating the scalability of baselines in the name-only CL setup, we have run additional experiments on the remaining baselines after the initial submission. In the revision, we now include these expanded results and have moved the ImageNet-R table to the main paper (Table 1), as you suggested. As shown in the table, GenCL outperforms both generative and web-scraping baselines, as well as manually annotated data. Thank you for the valuable suggestion.
>
>
> > **Q3.** My understanding is that most of the results presented in the paper are run by the authors themselves. Can you clarify in the text whether this is the case, and for which results? Why not use numbers in the concurrent papers?
>
> $\to$  All results presented in the paper were reproduced by us, as our experimental setup and benchmarks differ from those used in prior works. For each baseline, we strictly followed the official code provided by the respective papers, making only the minimal modification of adding our benchmark concept names to obtain the required data. All baselines were then executed under the same experimental setup and unified configuration to ensure a fair comparison.
>
> Specifically, for the setup, our work tackles the name-only continual learning setup, where novel concepts are introduced incrementally during the training process. In contrast, most prior works report results under the assumption of a joint training setup, where all target concepts are given at once. Although one baseline (i.e., C2C) also explores a name-only CL setup, its reported results are limited to fine-grained classification benchmarks (e.g., Flowers102, CUB-200) and only evaluate on in-domain test data. In contrast, we focus on domain generalization benchmarks (e.g., PACS, DomainNet, and ImageNet-R) to evaluate both in-domain and out-of-domain generalization performance. We also include experiments on a fine-grained benchmark (i.e., Birds-31), but use a domain generalization version that spans multiple domains, unlike prior works that use in-domain-only fine-grained datasets.
> Given these differences in setup, benchmarks, and evaluation protocols, previously reported numbers are not directly comparable. Therefore, we reproduced all baseline results using each paper’s official code, modifying only the concept names to align with our benchmarks, and evaluated them under the name-only continual learning setup with unified configuration, for a fair comparison.

---

> ### Author Response · Authors · 2025-08-18
> **Answers to the questions of Reviewer XSeX (2/3)**
>
> > **Q4.** The name-only continual learning setup is very niche and feels very artificial. Is it related to any real-world application?
>
> $\to$ Thank you for the insightful question. We argue that name-only CL can be highly useful in many real-world applications, particularly when the novel concepts are complex, which makes manual annotation significantly more expensive than for common concepts
>
> For example, consider a traffic violation detection model. Suppose a recent regulation update changes the legal interpretation of motorcycles on sidewalks: previously, possessing a motorcycle on a sidewalk was considered a violation, but under the new rule, only riding a motorcycle on sidewalks is regarded as a violation, while holding or walking it along by hand is no longer penalized. Before this update, detecting the presence of a motorcycle was sufficient to detect traffic violations. However, after the update, the model must distinguish between ‘riding a motorcycle’ and ‘holding a motorcycle’—two semantically distinct actions involving the same object. While collecting and annotating data for simple objects like "motorcycle" is relatively straightforward, acquiring data for these more nuanced, compositional concepts (e.g., ‘riding a motorcycle’, ‘holding a motorcycle’, ‘washing a motorcycle’) requires substantial human labor and greatly prolongs the data acquisition process [1, 2]. Note that in continual learning, such annotation delays significantly hinder real-time adaptation to newly emerging concepts [3, 4].
>
> To assess whether name-only continual learning, especially with our proposed GenCL, can effectively replace manual annotation for such complex concepts, we evaluated on the Bongard-HOI benchmark, which contains 50 object–action composite concepts. As shown in Table 3, our proposed GenCL not only outperformed web-scraping and other generative baselines, but also surpassed models trained with manually annotated data, demonstrating that name-only CL with GenCL can effectively replace manual annotation in such complex concepts while can requiring significantly less human effort and shorter data acquisition time.
>
>
> > **Q5.** The use of generative models is not necessarily better than scrapping the web in terms of diversity / privacy / licences. This highly depends on the choice of generative model, the data it was trained on ect. The paper mentions: “However, web-scraped data suffer from noise, low controllability, and usage restrictions”, the exact same can apply here.
>
> $\to$ Great point! We respectively clarify that our paper does not claim that generative models produce more diverse data and pose less privacy concern than web-scraped sources. Specifically, regarding diversity, rather than claiming generative models produce more diverse data, we explicitly pointed out their limited diversity in Sec. 2.2, noting that existing generative baselines suffer from this limitation despite being capable of producing unlimited images. Similarly, regarding privacy, rather than claiming that generated data pose fewer privacy concerns, we explicitly acknowledged in Sec. 7 the privacy risks associated with generated data and suggested future research directions to address these concerns. We intentionally did not emphasize any inherent advantage in terms of privacy or diversity, since these factors indeed depend on the specific generative model used, as you noted. If there are any parts of our paper that may have contributed to this misunderstanding, we would greatly appreciate it if you could point them out so that we may revise them accordingly.
>
> For the usage licenses and noise in data, we respectfully argue that recent strong generative models, under current copyright laws, often generate data that are less noisy and easier to use compared to web-scraped data, while these advantages may depend on the choice of generative model. Regarding usage restrictions and copyright, many governments, including the US [5], EU [6], and South Korea [7], consider content generated entirely by AI to be non-copyrightable and therefore free to use, except in cases where a human is significantly involved in the generative process. Based on those regulations, as our proposed name-only generative framework (GenCL) generates data based solely on a given concept name and does not require any human labor, the generated data are copyright-free. On the other hand, web-scraping is often subject to usage restrictions due to factors like API monetization (e.g., X, formerly Twitter), trademarks, and other legal constraints [8, 9]. Regarding the data noise, web-scraped data often become noisy when search queries are detailed (e.g., longer queries produce less relevant data) [10], as also shown in the qualitative analysis (Fig. 6), while text-to-image (T2I) generative models tend to produce more relevant and less noisy data even with detailed queries, thanks to their text-based controllability [11].

---

> ### Author Response · Authors · 2025-08-18
> **Answers to the questions of Reviewer XSeX (3/3)**
>
> > **Q6.** The performance of the model is also highly dependent on the choice of generative model. I would like to see a discussion or additional experiments with different generative models.
>
> >> **Q6.1** For example, the paper uses GPT-4o, but how would the performance change if another model was used instead?
>
> $\to$ Thank you for the insightful question! To demonstrate that our method maintains strong performance even when the text prompt generative model is changed, we conducted additional experiments replacing GPT-4o with LLaMA3-8B and Qwen3-8B. As shown in Table 19 and Table 20 in Sec. 25, our proposed GenCL consistently outperforms baselines when either Qwen-3-8B or LLaMA-3-8B is employed. These results indicate that, while the absolute performance varies across different LLMs, GenCL does not rely on a specific LLM and generalizes well across a variety of models.
>
>
> >> **Q6.2** Finally, in this case GPT-4o is proprietary, that means less control over what data is used and generated and less understanding of how the model works.
>
> $\to$ Thank you again for the great comment! We would first like to clarify that while we used GPT-4o to generate text prompts following previous works [12, 13], we employed **only open-source text-to-image generative models**, i.e., SDXL, DeepFloyd-IF, Stable Diffusion 3 (SD3), CogView2, and Auraflow, to generate images, as detailed in Sec. A.14. Furthermore, as shown in Sec. 25, replacing GPT-4o with open-source LLMs such as LLaMA3-8B and Qwen3-8B still shows consistent performance gains, demonstrating that all components of GenCL can be implemented with open-source models while maintaining strong results. Using open-source models will allow for better tracking and understanding of their internal workings and provide greater controllability over the generated data.
>
>
> [1] Saito et al., Gu et al., Language-only Efficient Training of Zero-shot Composed Image Retrieval, CVPR 2023
>
> [2] Gu et al., Language-only Efficient Training of Zero-shot Composed Image Retrieval, CVPR 2024
>
> [3] Koh et al., Online continual learning on class incremental blurry task configuration with anytime inference, ICLR 2022
>
> [4] Caccia et al., On anytime learning at macroscale, CoLLAs 2022
>
> [5] US Copyright and Artificial Intelligence (Page 7 - Executive Summary), U.S. Copyright (https://www.copyright.gov/ai/)
>
> [6] Generative AI and Copyright (Sec. 1.3 - Copyright Law in the EU: key principles), EU Parliament
>
> [7] Korean Music Copyright Association (KOMCA) banned registration of songs created with AI, Music Business Worldwide, 2025 April (https://www.musicbusinessworldwide.com/south-koreas-largest-music-copyright-collective-just-banned-registration-of-songs-created-with-ai/)
>
> [8] Kazmali et al., Web Scraping: Legal and Ethical Considerations in General and Local Context - A Review, Procedia Computer Science 2025
>
> [9] Brown et al., Web Scraping for Research: Legal, Ethical, Institutional, and Scientific Considerations, arXiv 2024.10
>
> [10] Lotfi et al., Web Scraping Techniques and Applications: A Literature, SCRS 2021
>
> [11] Cao et al., Controllable Generation with Text-to-Image Diffusion Models: A Survey, arXiv 2403
>
> [12] Tian et al., Learning Vision from Models Rivals Learning Vision from Data, CVPR 2024
>
> [13] Hammoud et al., SynthCLIP: Are We Ready for a Fully Synthetic CLIP Training?, CVPRW 2024

---

> > ### Comment · Reviewer_XSeX · 2025-09-01
> > **Small mistake in Table 19**
> >
> > Thank you these answers, my concerns are fully addressed. I just want to point to a very small mistake in the newly added Table 19: first column, 77.62 from HIRPG (+) CONAN is bold while it's not the best result of the column, which is 78.88 from the HIRPG baseline.

---

> ### Author Response · Authors · 2025-09-02
> **Thank you for your response**
>
> Dear Reviewer XSeX,
>
> Thank you for your thoughtful comments and for taking the time to review our manuscript. We are glad that our additional clarifications and experiments have addressed your concerns. We also sincerely appreciate you pointing out our mistake in Table 19. We have corrected this formatting error and uploaded the revised version of the manuscript.
>
> If you have any further comments or suggestions, please do not hesitate to let us know. We are committed to improving the quality of our work and truly value your feedback.
>
> Thank you once again.
>
> Sincerely,
>
> Authors

---

### Review · Reviewer_4rEq · 2025-07-22

**Summary Of Contributions:**

This paper introduces GenCL, a framework for "name-only" continual learning (CL) that leverages generative models to synthesize training data given only the names of new concepts, removing the need for manual annotation or web-scraping. The central innovations are (1) HIerarchical Recurrent Prompt Generation (HIRPG), a prompt diversification technique that uses in-context LLM prompting to generate diverse text inputs for text-to-image models, and (2) COmplexity-NAvigating eNsembler (CONAN), an ensemble method that selects a diverse, minimally redundant coreset from multiple generative models using a relative Mahalanobis distance scheme. Empirical results show GenCL outperforms both web-scraped and even annotated data on various CL benchmarks, achieving higher out-of-distribution (OOD) performance and demonstrating lower demographic biases in generated data.

**Audience:**

Yes

**Broader Impact Concerns:**

This work involves generating training data using generative models, which may raise potential privacy concerns. To mitigate such risks, this work excludes any person-related generated data from the released dataset, as discussed in Section 7. I believe that advancing privacy-preserving and differential privacy generative models would effectively mitigate these concerns.

**Claims And Evidence:**

Yes

**Requested Changes:**

Please see strengths and weaknesses.

**Strengths And Weaknesses:**

**Strengths**

1. Addresses a Practical Bottleneck: Tackles the high annotation and data curation cost in online continual learning by operating in a "name only" regime, highly relevant for real-world deployment scenarios.

2. Solid Empirical Validation: The experiments demonstrate that GenCL not only outperforms web-scraped baselines but also surpasses fully supervised models, especially on OOD test splits, showing strong generalization.

3. Innovative Diversification Techniques: The proposed HIRPG and CONAN address key challenges in data diversity for synthetic training, as seen through both quantitative and qualitative analyses.

4. Bias and Fairness Analysis: The paper goes beyond accuracy focus, quantifying gender, race, and geographic bias and demonstrating GenCL generates more demographically balanced data than other automated or manual methods.

**Weaknesses**

1. Overreliance on Existing Popular Generative Models: The success of GenCL is fundamentally tied to the coverage and bias profile of the pre-trained text-to-image generators. While Section 7 briefly addresses this, the main results focus on concepts these generators already "know," not genuinely out-of-vocabulary or low-resource concepts. The “future work” section mentions this gap, but stronger discussion or a limited experiment on rare/unseen concepts would add value.

2. Limited Innovation Relative to Prior Generative Replay: The use of generative models for continual learning, as well as replay and coreset selection, is well-trodden territory. The main novelty lies in integrating recent T2I models with hierarchical LLM-driven prompt diversification and diversity-based ensembling, but the core conceptual structure is relatively incremental.

3. Prompt Engineering Complexity Not Fully Quantified: Although HIRPG is presented as efficient and diverse, the effect of prompt parameters, temperature, and tree depth on diversity and downstream accuracy is not thoroughly dissected in the main paper. Computation/memory costs are said to be reported in the appendix, but main-text quantitative results on cost vs. benefit are lacking.

---

> ### Author Response · Authors · 2025-08-18
> **Answers to the questions of Reviewer 4rEq (1/2)**
>
> > **Q1.** Overreliance on Existing Popular Generative Models: The success of GenCL is fundamentally tied to the coverage and bias profile of the pre-trained text-to-image generators. While Section 7 briefly addresses this, the main results focus on concepts these generators already "know," not genuinely out-of-vocabulary or low-resource concepts. The “future work” section mentions this gap, but a stronger discussion or a limited experiment on rare/unseen concepts would add value.
>
>
> $\to$ Thank you for your valuable feedback! We have added a more detailed discussion in Section 7 of the revision, along with additional experiments. Specifically, we argue that the use of a few-shot examples enables the continual adaptation of generative models to unseen concepts. This makes generating previously unknown concepts feasible. Moreover, the flexible design of GenCL allows it to evolve alongside advancements in foundation models (e.g., GPT-3.5 → GPT-4o → GPT-5), thereby expanding its capacity to acquire data for a wider range of concepts over time.
>
> To support this, we have conducted two additional experiments:
> - Continual Adaptation with Few-Shot Learning: In these experiments, we provide a few real samples to the generative model, which significantly improves both ID and OOD performance, even for previously seen concepts (Sec. A.26). By extension, we argue that providing a small number of real samples for completely unseen concepts can enable continual training of generative models, thereby generating abundant synthetic data for these unseen concepts. We believe that efficient continual training methods for generative models [1] will further accelerate GenCL’s ability to handle entirely unseen categories.
>
> - Replacement of GenCL’s Underlying LLMs: To demonstrate GenCL’s flexibility, we replace GPT-4o with alternative foundation models, such as LLaMA3-8B and Qwen3-8B. As shown in Sec. A.25, these experiments show that GenCL consistently outperforms baselines with these updated models, demonstrating its potential to achieve strong performance with future models. In other words, while GenCL does not directly implement continual adaptation, its flexible design ensures it advances alongside evolving foundation models, expanding its ability to acquire data for broader concepts over time.
>
> > **Q2.** Limited Innovation Relative to Prior Generative Replay: The use of generative models for continual learning, as well as replay and coreset selection, is well-trodden territory. The main novelty lies in integrating recent T2I models with hierarchical LLM-driven prompt diversification and diversity-based ensembling, but the core conceptual structure is relatively incremental.
>
> $\to$ We respectfully argue that our proposed name-only generative learning is fundamentally distinct from conventional generative replay in terms of its reliance on real data. Traditional generative replay methods normally reproduce the distribution of previously encountered data for replay, avoiding explicit storage of raw samples to mitigate privacy concerns. However, achieving this requires **fine-tuning generative models on real data** to align the replay distribution with the original task distribution [2, 3, 4, 5].
>
> In contrast, name-only generative learning eliminates real-data dependency. It relies solely on concept names to synthesize training data and thus does not involve training the generative model during continual learning. Although the generated data may not perfectly replicate the real-data distribution, it generates diverse samples by leveraging the broad knowledge embedded in large generative models, thus enabling robust performance across diverse domains (i.e., domain generalization).

---

> ### Author Response · Authors · 2025-08-18
> **Answers to the questions of Reviewer 4rEq (2/2)**
>
> > **Q3.** Prompt Engineering Complexity Not Fully Quantified
>
>
> >> **Q3.1** Although HIRPG is presented as efficient and diverse, the effect of prompt parameters, temperature, and tree depth on diversity and downstream accuracy is not thoroughly dissected in the main paper.
>
> $\to$  We respectfully clarify that we did analyze the effects of key prompt parameters, such as temperature on accuracy, and tree depth/width on both diversity and recognizability, in Sec. A.23. This analysis is briefly referenced in the main paper following Sec. 5.6. While focusing on presenting comprehensive quantitative and qualitative results in the main paper, we had to place the detailed hyperparameter analysis in the appendix due to space constraints. Please let us know if further elaboration is needed; we will be happy to strengthen this section.
>
>
> >> **Q3.2** Computation/memory costs are said to be reported in the appendix, but main-text quantitative results on cost vs. benefit are lacking.
>
>
> $\to$  Thank you for highlighting this. As per your suggestion, we have moved the computation and memory cost comparison from the appendix to the main text (Sec. 5.5). As shown in Table 6, GenCL requires less wall time than manual annotation and web-scraped data. While generative baselines, including GenCL, require memory to store the generative model weights (unlike web scraping baselines, which do not require memory), they do not necessitate connected networks or browsers for crawling.
>
> [1] Uehara et al., Feedback efficient online fine-tuning of diffusion models, ICML 2024
>
> [2] Liu et al., Generative Feature Replay For Class-Incremental Learning, CVPRW 2020
>
> [3] Gao et al., DDGR: Continual Learning with Deep Diffusion-based Generative Replay, ICML 2023
>
> [4] Kim et al., SDDGR: Stable Diffusion-based Deep Generative Replay for Class Incremental Object Detection, CVPR 2024
>
> [5] Meng et al., DiffClass: Diffusion-Based Class Incremental Learning, ECCV 2024

---

### Review · Reviewer_wpN7 · 2025-08-07

**Summary Of Contributions:**

This paper proposes GenCL for continual learning using only concept names, combining hierarchical prompt generation (HIRPG) with complexity-guided data selection (CONAN). The work addresses a relevant problem with solid experimental methodology but has notable limitations.

**Audience:**

Yes

**Broader Impact Concerns:**

**Inadequate Broader Impact Statement**

1. The authors have included a broader impact section, however, it does not address the concern of potentially present biases in generative and/or LLMs. Furthermore, generative models can potentially generate harmful and/or biased content, which requires careful removal from the dataset. The authors do not clarify how to mitigate such an issue.

2. In continual learning scenarios, biased synthetic data could lead to progressive bias accumulation as new concepts are learned, potentially amplifying discrimination over time. The automated nature of the generation process means biases could propagate unchecked across multiple learning episodes. Additionally, the approach could enable rapid deployment of biased models in real-world applications without adequate human oversight, particularly problematic in sensitive domains like hiring, lending, or law enforcement, where continual learning systems might be deployed.

**Claims And Evidence:**

Yes

**Requested Changes:**

**Add Critical Missing Baselines**

1. Few-shot learning: Compare with 1, 3, 5, 10 real examples per concept (most obvious practical alternative).

2. Modern continual learning methods: Include recent prompt-based and parameter-efficient approaches.

3. Stronger web-scraping: Use more sophisticated data collection beyond the basic 3-search-engine approach.

**Strengths And Weaknesses:**

**Strengths**

1. Practically important problem: Manual annotation costs are a genuine bottleneck in continual learning. This paper addresses this limitation with the help of pretrained generative models.

2. Comprehensive evaluation: Extensive experiments across multiple datasets with thorough ablation studies.

3. Strong empirical results: Consistent improvements over baselines, particularly on out-of-distribution domains.

**Weaknesses**

1. Missing critical baselines: No comparison with few-shot learning using, say, 1-5 real examples, which is the most obvious practical alternative, considering the fact that the proposed approach is heavily dependent on the pretrained models.

2. Limited technical novelty: Combines existing techniques (hierarchical generation, difficulty-based selection) without fundamental algorithmic advances.

3. Experimental design concerns: ID/OOD splits may favor synthetic data; evaluation focuses on standard visual concepts.

4. Foundation model dependency: Performance is bounded by pre-trained model capabilities with no learning mechanism and/or continual adaptation, which is a crucial aspect of any CL approach.

5. Scalability issues: Requires substantial computational resources; limited analysis of deployment costs.

**Additional Comments**

1. HIRPG is well-motivated but fundamentally limited by underlying LLM capabilities.

2. CONAN's RMD-based selection is reasonable but lacks comparison with sophisticated active learning approaches

3. Missing failure case analysis and human evaluation of generated image quality

---

> ### Author Response · Authors · 2025-08-18
> **Answers to the questions of Reviewer wpN7 (1/5)**
>
> > **Q1.** Foundation model dependency: Performance is bounded by pre-trained model capabilities with no learning mechanism and/or continual adaptation, which is a crucial aspect of any CL approach.
>
> $\to$ Thank you for the insightful comment. We argue that GenCL’s flexible design, which allows seamless replacement of the underlying foundation model, mitigates being bound by the capabilities of a fixed pre-trained model. This enables GenCL to grow stronger alongside the continually updated strong foundation models (e.g., GPT-3.5 → GPT-4o → GPT-5) that are updated with the latest knowledge, whereas several baselines are tightly coupled to a specific foundation model and thus cannot exploit evolving models. For example, LE [1] designs its data acquisition pipeline specifically for the T5-sentence model, making replacement with stronger LLMs infeasible. In contrast, GenCL is not tailored to any particular LLM or generative model, enabling not only replacement with increasingly capable foundation models but also the selective use of smaller models when used in resource-constrained environments.
>
> To demonstrate its flexibility and generalizability beyond any specific foundation model, we replaced the GPT-4o, an LLM that was originally employed in GenCL, with LLaMA3-8B and Qwen3-8B, and summarize the results in Sec. A.25 of the revision. As shown, GenCL consistently outperforms baselines, indicating strong potential to maintain or improve performance when deployed with more capable future foundation models. The same replacement strategy would be applied to the generative models.  In other words, while GenCL does not directly implement continual adaptation, its flexible design allows it to advance alongside continually improving foundation models, making it an increasingly powerful data acquisition framework over time.
>
>
> > **Q2.** Limited technical novelty: Combines existing techniques (hierarchical generation, difficulty-based selection) without fundamental algorithmic advances.
>
> $\to$ We respectfully clarify that our proposed components (i.e., CONAN and HIRPG) are distinct from existing techniques in both objective and methodology.
>
> For the proposed sample selection strategy (i.e., CONAN), while the high-level intuition of difficulty-based selection has been explored, our approach differs in both (1) how difficulty is defined and (2) how it is used for sample selection. Prior works have defined difficulty via entropy [2, 3], classification margin (difference between top-1 and top-2 predicted probabilities) [4, 5], or learnable parameters [6]. In contrast, we define difficulty using the relative Mahalanobis distance, which captures concept-conditional feature separability in a way not considered in prior work. In terms of how difficulty is used, prior deterministic approaches either pick the most difficult samples [7] or choose moderate-difficulty ones by excluding both extremes [8]. In contrast, our method employs a probabilistic selection where higher difficulty increases the likelihood of selection, while still allowing lower-difficulty samples to be included. We believe these differences directly contributed to the performance gains observed in Table 4 (comparison to coreset selection baselines) and Table 15 in Sec. A.19 (comparison to hard-negative selection baselines).
>
> For the proposed hierarchical generation strategy (i.e., HIRPG), our objective fundamentally differs from that of prior work using hierarchical structures. Previous studies have used hierarchy to first produce an outline or plan (sections, sub-tasks) and then generate details for each item separately [9, 10], or to split long inputs into chunks for partial summarization and final integration [11, 12]. In contrast, we use a hierarchical generation strategy specifically to produce diverse prompts, enabling richer variation in generated data, while sharing the general benefits of hierarchical structures.

---

> ### Author Response · Authors · 2025-08-18
> **Answers to the questions of Reviewer wpN7 (2/5)**
>
> > **Q3.** Experimental design concerns:
>
> >> **Q3.1.** ID/OOD splits may favor synthetic data
>
> $\to$ Thank you for your comment. We respectfully emphasize that our ID/OOD domain evaluation design is intended to assess domain generalizability in continual learning, an aspect consistently highlighted in prior continual learning research [13, 14, 15], rather than to create a setup that favors synthetic data. Domain generalization is an essential requirement in many real-world continual learning scenarios. For example, a self-driving model continually adapted after deployment in a new region must still recognize a newly learned vehicle type in foggy or nighttime conditions, even if that type was learned from sunny daytime data, to ensure safety. Our evaluation is therefore designed to measure domain generalization, going beyond standard schemes that assess performance only in ID domains (i.e., domains identical to the training data).
>
> Moreover, we argue that evaluation on domain generalization (DG) benchmarks may in fact favor web-scraping baselines rather than synthetic approaches. This is because many DG benchmarks (e.g., DomainNet) consist of web-scraped data and thus share the same source domain as web-scraping baselines such as C2C and IE, potentially giving these baselines an inherent advantage over generative ones. Indeed, as shown in Table 2, the web-scraping baseline C2C outperforms all generative baselines (except our proposed GenCL and manually annotated (MA) data) on DomainNet in both ID and OOD domains. However, our proposed GenCL not only surpasses C2C, but also outperforms MA data in the OOD domain, demonstrating its ability to generate diverse, high-quality data that effectively enhances domain generalizability.
>
> >> **Q3.2.** Evaluation focuses on standard visual concepts
>
> $\to$ We respectfully note that our evaluation goes beyond standard visual concepts, incorporating fine-grained concepts and complex object–action combinations that reflect new concepts emerging in real-world continual learning scenarios. Specifically, beyond the common visual concepts in DomainNet and PACS, we evaluate on Birds-31, which classifies 31 fine-grained bird species, and Bongard-HOI, which goes beyond simple noun-based concepts to distinguish visual concepts composed of object–action pairs (e.g., lie on the bed, wash a motorcycle). If there are additional benchmarks you believe would further strengthen our evaluation, we are happy to include results on those as well.
>
> > **Q4.** Scalability issues:
>
> >> **Q4.1.** Requires substantial computational resources
>
> $\to$ Great point! We respectfully emphasize that, thanks to the efficiency of the recent generative models, it does not require substantial computational resources, even though it requires more GPU resources compared to web scraping or manual annotation. Specifically, as detailed in Sec. 5.5, image generation can be performed even on RTX 4090 GPUs, without the need for high-end GPUs like the A100 and H100. For example, using only four RTX 4090 GPU nodes, we can generate the full DomainNet dataset in about **3 hours**, while manual annotation of a comparable dataset has been reported to require **50,000 working hours**, and web scraping takes over **5 hours**.
>
> Moreover, while generative baselines generally require more GPU resources than web-scraping approaches, web-scraping baselines still incur notable GPU resources for computing CLIP scores on all candidate images in order to filter out noisy samples [16], as well as network connections for crawling. Finally, we believe that integrating our proposed GenCL framework with emerging, more computationally and memory-efficient sparse generative models [17, 18] would further reduce GPU resource requirements and enhance scalability.
>
>
> >> **Q4.2.** Limited analysis of deployment costs.
>
> $\to$ We believe our detailed analysis and comparison of computational and memory costs in Sec. 5.5 can measure deployment costs. If there are specific aspects you feel are missing, we would be happy to provide additional analysis.

---

> ### Author Response · Authors · 2025-08-18
> **Answers to the questions of Reviewer wpN7 (3/5)**
>
> > **Q5.** HIRPG is well-motivated but fundamentally limited by underlying LLM capabilities.
>
> $\to$ Thank you for the great comment! We agree that HIRPG relies on the capabilities of the underlying LLM; however, it can be easily replaced with a stronger LLM, and we believe its performance will improve in step with the continual advancement of LLM capabilities, as also noted in our response to Q1. To assess this flexibility, beyond GPT-4o (our main experimental backbone), we additionally evaluated HIRPG with LLaMA and Qwen, and summarize the results in Sec. A.25 of the revision. As shown, HIRPG outperforms all baselines on both LLMs, demonstrating that its effectiveness is consistent regardless of the underlying LLM, even though absolute performance may vary. In summary, while HIRPG relies on the underlying LLM, its flexibility ensures that it can readily benefit from future advances in LLMs, enabling stronger performance as more capable models become available.
>
> > **Q6.** CONAN's RMD-based selection is reasonable but lacks comparison with sophisticated active learning approaches
>
> $\to$ Thank you for the insightful comment. As you suggested, we have added additional comparisons with active learning approaches [22, 23], since off-the-shelf active learning algorithms can also be applied to data subset selection, as noted in prior work [22, 23]. We added the results in Table 4. As shown, CONAN consistently outperforms active learning baselines as well.
>
> > **Q7.** Missing failure case analysis and human evaluation of generated image quality
>
> $\to$ Thank you for the suggestion. We will include a detailed failure case analysis in the next revision (in a few days).
>
> > **Q8.** Add Critical Missing Baselines
>
>
> >> **Q8.1.** Few-shot learning: Compare with 1, 3, 5, 10 real examples per concept (most obvious practical alternative).
>
> $\to$ Thank you for the suggestion. As you suggested, we compare GenCL with few-shot learning baselines [24, 25, 26] that exploit several real examples (e.g., 1, 3, 5, 10 real examples). As shown in Fig. 16 in Sec. A.26 of the revision, increasing the number of real samples substantially improves ID accuracy; however, for OOD performance, accuracy converges after around 3 examples and even slightly degrades in several baselines when more real samples are added. This occurs because fine-tuning generative models leads to image generation that closely matches the style and background of the provided real examples, thereby enhancing ID accuracy. However, this simultaneously reduces dataset diversity.
>
>
> Importantly, this limitation stems from the differing goals of few-shot learning baselines versus name-only generative baselines, which is why we did not originally include such methods. Few-shot learning baselines are primarily designed for personalized image generation (e.g., generating images of a specific person or a user’s own car), whereas our focus is on general concept learning. In personalized continual learning settings, where the goal is to continually acquire user-specific concepts, few-shot example-based generation can indeed serve as an effective data acquisition strategy. In contrast, our name-only continual learning aims to learn concepts in a general sense rather than personalized instances. For instance, in a traffic-violation detection model, if riding a motorcycle on sidewalks is considered a violation, the model must detect all types of motorcycles, not just one specific style or instance. For such general concept learning scenarios, training the model with diverse data is essential. Therefore, we select baselines that generate diverse images given a concept via the stochasticity of off-the-shelf generative models and prompt diversification.
>
> Finally, we argue that GenCL can also effectively employ few-shot real examples. As shown in Fig. 16 (Sec. A.26 of the revision), fine-tuning with real examples further improves its performance. Thus, GenCL is not restricted to the name-only setup but can flexibly benefit from real examples when available, achieving even stronger results.

---

> ### Author Response · Authors · 2025-08-18
> **Answers to the questions of Reviewer wpN7 (4/5)**
>
> >> **Q8.2.** Modern continual learning methods: Include recent prompt-based and parameter-efficient approaches.
>
> $\to$ Thank you for the insightful suggestion. We have conducted additional experiments incorporating recent prompt-based and parameter-efficient continual learning methods (Cprompt [27], MISA [28]).
>
> As our work focuses on data acquisition for continual learning rather than developing a new continual learning method, our primary comparisons used a widely adopted continual learning baseline (i.e., ER) for a fair comparison. Beyond this, and as you suggested, we further evaluate GenCL with these recent methods to demonstrate its applicability regardless of the underlying continual learning algorithm, and summarize the results in Sec. A.24 of the revision. As shown, continual learning on data acquired by GenCL consistently outperforms baselines, demonstrating that GenCL can be effectively combined with future state-of-the-art continual learning algorithms.
>
> >> **Q8.3.** Stronger web-scraping: Use more sophisticated data collection beyond the basic 3-search-engine approach.
>
> $\to$ Thank you for the suggestion. We will include a comparison with more sophisticated web-scraping baselines in the next revision (in a few days).
>
>
> > **Q9.** Inadequate Broader Impact Statement. The authors have included a broader impact section; however, it does not address the concern of potentially present biases in generative and/or LLMs. Furthermore, generative models can potentially generate harmful and/or biased content, which requires careful removal from the dataset. The authors do not clarify how to mitigate such an issue.
> In continual learning scenarios, biased synthetic data could lead to progressive bias accumulation as new concepts are learned, potentially amplifying discrimination over time.
>
>
> $\to$ Thank you for your valuable comment. We first clarify that, as shown in Fig. 5 and Fig. 6 in Sec. 5.3, GenCL’s diverse image generation strategies (i.e., HIRPG and CONAN)  implicitly lead to producing less-biased data compared not only to other generative baselines but also to manually annotated data.
>
> Nevertheless, we fully agree with the reviewer that explicit bias-prevention strategies are essential, as continual learning on biased data can progressively amplify model biases over time, posing critical risks for real-world deployment. To further address this issue, combining GenCL with bias-reducing techniques, such as editing bias-inducing parameters in generative models [19, 20], as well as fine-tuning generative models with small, unbiased datasets [21], can effectively prevent bias propagation in GenCL and reduce the risks associated with deploying biased models. We have added the discussion on potential risks of generated data and possible mitigation strategies in Sec. A.35 of the revised manuscript.

---

> > ### Comment · Reviewer_wpN7 · 2025-09-01
> > **Reply to the Rebuttal**
> >
> > Thank you for the rebuttal and revision. It mostly addresses my concerns.

---

> > > ### Author Response · Authors · 2025-09-02
> > > **Thank you for your response**
> > >
> > > Dear Reviewer wpN7,
> > >
> > > Thank you for your comments and for taking the time to review our manuscript. We are happy that our additional clarification and experiments addressed your concerns! If you have any further comments or suggestions, please let us know. We are committed to improving the quality of our work, and we value your feedback.
> > >
> > > Thank you very much,
> > >
> > > Authors

---

> ### Author Response · Authors · 2025-08-18
> **Answers to the questions of Reviewer wpN7 (5/5)**
>
> [1] He et al., Is synthetic data from generative models ready for image recognition?, ICLR 2023
>
> [2] Zhou et al., Uncertainty-Aware Curriculum Learning for Neural Machine Translation, ACL 2020
>
> [3] Kim et al., Denoising Task Difficulty-based Curriculum for Training Diffusion Models, ICLR 2025
>
> [4] Sosea et al., MarginMatch: Improving Semi-Supervised Learning with Pseudo-Margins, CVPR 2023
>
> [5] Son et al., Difficulty-aware Balancing Margin Loss for Long-tailed Recognition, arXiv 2024.12
>
> [6] Lalor et al., Dynamic Data Selection for Curriculum Learning via Ability Estimation, EMNLP 2020
>
> [7] Jiang et al., Hard negative sampling via regularized optimal transport for contrastive representation learning, IJCNN 2023
>
> [8] Xia et al., Moderate Coreset: A Universal Method of Data Selection for Real-world Data-efficient Deep Learning, ICLR 2023
>
> [9] Sun et al., PEARL: Prompting Large Language Models to Plan and Execute Actions Over Long Documents, EACL 2024
>
> [10] Erdogan et al., Plan-and-Act: Improving Planning of Agents for Long-Horizon Tasks, ICML 2025
>
> [11] Jin et al., Hierarchical Document Refinement for Long-context Retrieval-augmented Generation, ACL 2025
>
> [12] Zhou et al., LLM×MapReduce: Simplified Long-Sequence Processing using Large Language Models, ACL 2025
>
> [13] Simon et al., On Generalizing Beyond Domains in Cross-Domain Continual Learning, CVPR 2022
>
> [14] Liu et al., DEJA VU: Continual Model Generalization For Unseen Domains, ICLR 2023
>
> [15] Cui et al., Generalized Few-Shot Continual Learning with Contrastive Mixture of Adapters, arXiv 2023
>
> [16] Prabhu et al., From Categories to Classifiers: Name-Only Continual Learning by Exploring the Web, CoLLAs 2024
>
> [17] Cai et al., HiDream-I1: A High-Efficient Image Generative Foundation Model with Sparse Diffusion Transformer, arXiv 2025.05
>
> [18] Wang et al., SparseDM: Toward Sparse Efficient Diffusion Models, ICME 2025
>
> [19] Wang et al., etoxifying large language models via knowledge editing, ACL 2024
>
> [20] Xu et al., iasedit: Debiasing stereotyped language models via model editing, ACL 2025 TrustNLP Workshop
>
> [21] Chen et al., Unified view of differentially private deep generative modeling, arXiv 2023.09
>
> [22] Cho et al., Querying Easily Flip-flopped Samples for Deep Active Learning, ICLR 2024
>
> [23] Park et al., Active Learning is a Strong Baseline for Data Subset Selection, NeurIPSW 2022
>
> [24] Aiello et al., DreamCache: Finetuning-Free Lightweight Personalized Image Generation via Feature Caching, CVPR 2025
>
> [25] Purushwalkam et al., Bootpig: Bootstrapping zero-shot personalized image generation capabilities in pretrained diffusion models, ECCV 2024
>
> [26] Ruiz et al., DreamBooth: Fine Tuning Text-to-Image Diffusion Models for Subject-Driven Generation, CVPR 2023
>
> [27] Gao et al., Consistent Prompting for Rehearsal-Free Continual Learning, CVPR 2024
>
> [28] Kang et al., Advancing Prompt-Based Methods for Replay-Independent General Continual Learning, ICLR 2025

---

> ### Author Response · Authors · 2025-08-23
> **Additional Answers to Reviewer wpN7 in the Second Revision**
>
> > **Q7.** Missing failure case analysis and human evaluation of generated image quality
>
> $\to$ Thank you for this valuable suggestion. In response, we have added a detailed failure case analysis of GenCL in Sec.A.36. As shown, GenCL occasionally produces unnatural images, such as implausible arm and leg arrangements. Nevertheless, we argue that training with both natural and imperfect samples promotes robustness and improves generalizability, which in turn contributes to the strong out-of-domain performance reported in Tables 1, 2, and 3.
>
> > **Q8.3.** Stronger web-scraping: Use more sophisticated data collection beyond the basic 3-search-engine approach.
>
> $\to$ Thank you for the suggestion. We respectfully clarify that the web-scraping baselines we compared with GenCL, i.e., C2C [1] and IE [2], are not “basic” strategies relative to other prior work. In particular, while most web-scraping baselines rely on a single search engine, C2C employs multiple engines (e.g., Google, Baidu, and Flickr) to increase the diversity of collected data. Furthermore, many recent baselines that use web-scraped datasets assume that large-scale web-scraped datasets for the target concept are already available, and primarily focus on how to combine them for training [3, 4, 5, 6, 7, 8]. As a result, they can only be applied when such preexisting datasets exist. In contrast, our name-only setup does not assume that concept-related data are already collected; thus, it begins from a concept name and directly constructs the training dataset via scraping. Consequently, most recent baselines that rely on pre-collected web-scraped data are unsuitable for our name-only setup.
>
> That said, we acknowledge the reviewer’s point and have extended our comparison to additional web-scraping baselines, Seafaring [9] and Tiara [10]. These methods acquire images for active learning by selecting the most informative samples using acquisition functions, and they do not rely on a fixed search engine. As shown in Table ??, our proposed GenCL consistently outperforms both methods, demonstrating that GenCL is more effective at acquiring data with higher quality and diversity compared to existing web-scraping approaches.
>
> [1] Prabhu et al., From Categories to Classifiers: Name-Only Continual Learning by Exploring the Web, CoLLAs 2024
>
> [2] Li et al., Internet explorer: Targeted representation learning on the open web, ICML 2023
>
> [3] Zhu et al., Multimodal C4: An Open, Billion-scale Corpus of Images Interleaved with Text, NeurIPS 2023 Dataset and Benchmark Track
>
> [4] Laurençon et al., OBELICS: An Open Web-Scale Filtered Dataset of Interleaved Image-Text Documents, NeurIPS 2023 Dataset and Benchmark Track
>
> [5] Li et al., OmniCorpus: A Unified Multimodal Corpus of 10 Billion-Level Images Interleaved with Text, ICLR 2025
>
> [6] Singla et al., From Pixels to Prose: A Large Dataset of Dense Image Captions, arXiv 2024.06
>
> [7] Kumar et al., VisCon-100K: Leveraging Contextual Web Data for Fine-tuning Vision Language Models, PAKDD 2025
>
> [8] Cahyawijaya et al., Crowdsource, Crawl, or Generate? Creating SEA-VL, a Multicultural Vision-Language Dataset for Southeast Asia, ACL 2025
>
> [9] Sato et al., Active Learning from the Web, WWW 2023
>
> [10] Ryoma et al., Retrieving Black-box Optimal Images from External Databases, WSDM 2022

---

### Author Response · Authors · 2025-08-18
**General response**

We thank the reviewers for their helpful feedback and encouraging comments, including strong empirical results (**ALL reviewers**), comprehensive evaluation (**ALL reviewers**), innovative and effective method (**Reviewer 4rEq, XSeX**), tackling a practical and important problem (**Reviewer wpN7, 4rEq**), well-documented details (**Reviewer XSeX**), and detailed bias and fairness analysis (**Reviewer 4rEq**).

We have uploaded the first revision of the manuscript (changes highlighted in red), with the following key updates:
- Additional experiments on recent parameter-efficient CL methods, e.g., MISA, CPrompt (Sec. A.24)
- Additional experiments using LLaMA-3-8B and Qwen-3-8B as HIRPG’s LLM (Sec. A.25)
- Additional comparisons with baselines using a few real samples (Sec. A.26)
- ImageNet-1K experiments: added baselines and moved results to the main paper (Table 1)
- Additional comparison of CONAN with active learning approaches (Table 4)
- Add bias concerns and corresponding mitigation strategies in the Broader Impact Statement (Sec. A.35).

---

### Author Response · Authors · 2025-08-23
**Second revision**

We have uploaded the second revision of the manuscript. This version incorporates additional comparisons and more detailed analysis, addressing the suggestions provided by Reviewer wpN7.

Summary of Changes
- Added comparisons with sophisticated web-scraping baselines, such as Seafaring and Tiara (Tables 2 and 3).
- Included a detailed failure case analysis of GenCL (Sec. A.36).

---

### Decision · Action_Editor_ViMS · 2025-09-24

**Recommendation:** Accept as is

**Audience:**

Yes

**Audience Explanation:**

Indeed. The paper would interest researchers from several areas such as those working on continual learning, data-efficient training, and generative models. This paper’s findings show how to scale continual learning to make it practical and applicable in real-world settings. Even if some might see the contribution as incremental, the results provide useful empirical insights and design ideas that would interest researchers working in these areas.

**Claims And Evidence:**

Yes

**Claims Explanation:**

This paper addresses the "name-only" continual learning problem by introducing GenCL, which combines hierarchical prompt generation with a difficulty-based ensemble and selection strategy.

The paper claims that GenCL enables generating diverse and high-quality training data without manual labels or web scraping, and leads to the gains in diversity, robustness, and generalization. In line with these claims, the method is shown to outperform both web-scraped and even manually annotated data, particularly on out-of-distribution domains, while also reducing measured demographic bias. The method is further claimed to be flexible across different foundation models and computationally feasible compared to annotation or scraping pipelines.

There is general consensus among reviewers that the problem is important, the method is effective, and the empirical validation is strong across diverse benchmarks. There were some concerns initially regarding some missing baselines, the method's reliance on foundation models, and limited novelty. All of these aspects were substantially addressed by revising the paper. The authors added comparisons with few-shot learning, parameter-efficient CL, stronger web-scraping and active-learning baselines, alternative LLMs (LLaMA, Qwen), failure case analyses, and detailed cost/bias studies.

All the reviewers were in general satisfied with the author response and the paper revision and are in favor of acceptance.

While the work’s conceptual novelty is incremental and some claims (e.g., rare/unseen concepts) should be scoped more cautiously, the core claims are well supported by convincing evidence, in line with the TMLR acceptance criteria.

Overall, in view of the reviewers' as well as my own assessment of the submission, I recommend acceptance.